# $20^{th}$-century changes in carbon isotopes and water-use efficiency: Tree-ring based evaluation of the CLM4.5 and LPX-Bern models

Kathrin M. Keller[1,2], Sebastian Lienert[1,2], Anil Bozbiyik[1,2], Thomas F. Stocker[1,2], Olga V. Churakova (Sidorova)[3,4], David C. Frank[2,5,6], Stefan Klesse[2,5,6], Charles D. Koven[7], Markus Leuenberger[1,2], William J. Riley[7], Matthias Saurer[5,8], Rolf Siegwolf[5,8], Rosemarie B. Weigt[8], and Fortunat Joos[1,2]

[1]Climate and Environmental Physics, Physics Institute, University of Bern, Switzerland
[2]Oeschger Centre for Climate Change Research, University of Bern, Switzerland
[3]Institute for Environmental Sciences, University of Geneva, Switzerland
[4]Siberian Federal University, Krasnoyarsk, Russia
[5]Swiss Federal Research Institute WSL, Birmensdorf, Switzerland
[6]Laboratory of Tree Ring Research, the University of Arizona, Tucson, USA
[7]Lawrence Berkeley National Lab (LBNL), Berkeley, CA, USA
[8]Paul Scherrer Institute, Villigen, Switzerland

*Correspondence to:* K.M. Keller (keller@climate.unibe.ch)

**Abstract.** Measurements of the stable carbon isotope ratio ($\delta^{13}$C) on annual tree rings offer new opportunities to evaluate mechanisms of variations in photosynthesis and stomatal conductance under changing $CO_2$ and climate, especially in conjunction with process-based biogeochemical model simulations. The isotopic discrimination is indicative of the ratio between the $CO_2$ partial pressure in the intercellular cavities and the atmosphere ($c_i/c_a$) and of the ratio of assimilation to stomatal conductance, termed intrinsic water-use efficiency (iWUE). We performed isotope-enabled simulations over the industrial period with the land biosphere module (CLM4.5) of the Community Earth System Model and the LPX-Bern dynamic global vegetation model. Results for C3 tree species show good agreement with a global compilation of $\delta^{13}$C measurements on leaves, though modeled $^{13}$C discrimination by C3 trees is smaller in arid regions than measured. A compilation of seventy-six tree-ring records, mainly from Europe, boreal Asia, and western North America, suggest on average small $20^{th}$-century changes in isotopic discrimination and in $c_i/c_a$ and an increase in iWUE of about 27% since 1900. LPX-Bern results match these century-scale reconstructions, supporting the idea that the physiology of stomata has evolved to optimize trade-offs between carbon gain by assimilation and water loss by transpiration. In contrast, CLM4.5 simulates an increase in discrimination and in turn a change in iWUE that is almost twice as large as revealed by the tree-ring data. Factorial simulations show that these changes are mainly in response to rising atmospheric $CO_2$. The results suggest that the down-regulation of $c_i/c_a$ and of photosynthesis by nitrogen limitation is possibly too strong in the standard setup of CLM4.5 or there may be problems associated with the implementation of conductance, assimilation, and related adjustment processes to long-term environmental changes.

# 1 Introduction

Measurements of the stable isotope $^{13}C:^{12}C$ ratio ($\delta^{13}C$) on samples from air and natural archives hold information on the carbon cycling in the Earth System. A particularly important area of isotopic research is to clarify mechanisms controlling $\delta^{13}C$ carbon assimilation and water transpiration by land plants (Farquhar et al., 1989; Saurer et al., 2014; Voelker et al., 2016) and their role for the global terrestrial carbon sink (Ciais et al., 2013). There are numerous other interesting applications of $\delta^{13}C$ in the context of Earth System science. The modern decrease in atmospheric $\delta^{13}CO_2$ is an unequivocal testimony to the input of isotopically-light fossil and terrestrial carbon by human activities (Keeling et al., 1979; Francey et al., 1999). $\delta^{13}C$ data representing atmospheric air are used to quantify the global ocean and land carbon sources and sinks (Keeling et al., 1989; Joos and Bruno, 1998; Trudinger et al., 2002; Bauska et al., 2015). Furthermore, $\delta^{13}C$ observations allow identification of the imprint of fossil fuel carbon in atmospheric air to quantify regional-to-local-scale land carbon sources and sinks (Torn et al., 2011; Vardag et al., 2016), or to evaluate air-sea transfer velocity parameterizations (Krakauer et al., 2006). $\delta^{13}C$ data from the modern ocean are applied to infer the oceanic uptake of anthropogenic carbon (Heimann and Maier-Reimer, 1996; Gruber et al., 1999; Sonnerup and Quay, 2012; Becker et al., 2016), while paleo proxy $\delta^{13}C$ data from ocean sediments and ice cores permit inference of land carbon changes between the last glacial maximum and the current warm period (Shackleton, 1977; Ciais et al., 2012; Peterson et al., 2014). Paleo $\delta^{13}C$ data are also used to trace water mass, circulation and biological productivity changes on glacial-interglacial time scales and during past abrupt events (Menviel et al., 2012; Schmittner and Somes, 2016), to disentangle processes of past glacial-interglacial carbon cycle changes (Menviel and Joos, 2012; Schneider et al., 2013; Eggleston et al., 2016), and of ancient climate events (Kennett and Stott, 1991; Korte and Kozur, 2010).

Box models, ocean and land only models, and Earth System Models of Intermediate Complexity (Siegenthaler and Joos, 1992; Aranibar et al., 2006; Lai et al., 2006; Tschumi et al., 2011; Holden et al., 2013; Schmittner et al., 2013) have been traditionally evaluated by $\delta^{13}C$ data and used for the interpretation of $\delta^{13}C$ observations. Yet, despite such potential, $^{13}C$ has only been implemented recently in comprehensive Earth System Models and its subcomponents (Tagliabue and Bopp, 2008; Oleson et al., 2013; Jahn et al., 2015). Now carbon isotopes have been implemented in the ocean component of the Community Earth System Model (CESM) (Jahn et al., 2015). In this manuscript we present the implementation of $\delta^{13}C$ in the CESM land module.

Isotopic discrimination of plants following the C3 photosynthesis pathway depends on $CO_2$ assimilation and stomatal conductance (Farquhar et al., 1982; Lloyd and Farquhar, 1994) which themselves depend on the availability of nitrogen, water, and light, as well as species-specific leaf traits. Discrimination and variations thereof are thus indicative of the extent to which carbon assimilation by plants, fueling plant growth, is limited by factors such as drought and nitrogen limitation. In other words, quantification of isotopic discrimination changes over time and permits the evaluation of responses of stomatal conductance and assimilation to environmental variation and extremes. Environmental changes on the policy-relevant time scale of global warming include increasing atmospheric $CO_2$, climate change and increasing nitrogen deposition. Tree ring $\delta^{13}C$ records capture the influence of local climate variability with measured $\delta^{13}C$ variations used to reconstruct, for example, fluctuations in

temperature (Treydte et al., 2009; Sidorova et al., 2013), precipitation (Schubert and Jahren, 2011), or cloud cover (Gagen et al., 2011; Young et al., 2012).

The intrinsic water-use efficiency (iWUE), defined as the ratio between assimilation and stomatal conductance, is closely related to $^{13}$C discrimination. Rising atmospheric $CO_2$ concentrations can have a fertilizing effect on plants, which in turn potentially increases iWUE (Keenan et al., 2013; Saurer et al., 2014) – at least as long as plant growth is not limited by other factors such as nitrogen or phosphorus supply (Reich et al., 2014; Yang et al., 2016) or water stress (Walker et al., 2015).

There is a rich literature on changes in $^{13}$C isotopic discrimination and iWUE and on observational evidence from $\delta^{13}$C tree-ring records (Tans et al., 1979; McCarroll and Loader, 2004; Fichtler et al., 2010; Leonardi et al., 2012; Churakova (Sidorova) et al., 2014, 2016a; Lévesque et al., 2014; Liu et al., 2014b; Saurer et al., 2014; Hartl-Meier et al., 2015; Voelker et al., 2016), FACE-type experiments (Battipaglia et al., 2013; Klein et al., 2016), $\delta^{13}$C site measurements (Pataki et al., 2003; Bowling et al., 2014) and $\delta^{13}$C paleo data (Voelker et al., 2016). These data generally suggest small to moderate decadal-to-century-scale changes in discrimination that correspond to a $20^{th}$-century increase in iWUE in the order of 20% and physiological control towards a constant ratio of the partial pressure of $CO_2$ within the leaf's substomatal cavities to the $CO_2$ pressure outside the leaf ($c_i/c_a$; Saurer et al., 2004; Leonardi et al., 2012; Frank et al., 2015) and, more generalized, a pattern of stomatal optimization towards minimizing water loss per unit carbon assimilated (Voelker et al., 2016).

$\delta^{13}$C data are used to evaluate global and local models of plant growth, carbon cycling, and of land-atmosphere isotopic fluxes for atmospheric carbon balancing (Scholze et al., 2003, 2008; Suits et al., 2005; Danis et al., 2012; van der Velde et al., 2014). $\delta^{13}$C data from leaf material (Cornwell et al., 2016) are used by Prentice et al. (2014) and Wang et al. (2016) to develop the representation of assimilation in land biosphere models following an optimization principle to balance carbon gain by assimilation and water loss.

The goal of this study is to present the implementation of $\delta^{13}$C in the land component, CLM4.5, of CESM and LPX-Bern and to discuss the model performance for $\delta^{13}$C on the global scale. This is a step towards fully coupled isotope-enabled CESM applications and complements recent advances in simulating marine carbon isotopes. We compare the CLM4.5 and LPX-Bern results to a data set of $\delta^{13}$C measurements on modern leaf material (Cornwell et al., 2016), a comprehensive compilation of century-scale $\delta^{13}$C tree-ring records, as well as to results from the isotope-enabled LPX-Bern dynamic global vegetation model. We discuss spatial and century-scale trends in isotopic discrimination and iWUE of the two models in light of observational evidence, and the models contrasting implementations of stomatal conductance and the balance between carbon assimilation and water loss.

## 2 Methods

### 2.1 CLM4.5

We use the Community Land Model version 4.5 (CLM4.5; Oleson et al., 2013), the land component of the Community Earth System Model version 1.2.0 (CESM1.2; Hurrell et al., 2013). The implementation of $^{13}$C is outlined below (Section 2.1.1). A comprehensive description of the implementation of carbon isotopes in CLM4.5 is given in the Technical Description (Oleson

et al., 2013), further details elsewhere for [13]C (Raczka et al., 2016; Duarte et al., 2016) and [14]C (Koven et al., 2013). In addition to the land, carbon isotopes are also implemented in the ocean model of CESM1.2 (Jahn et al., 2015).

CLM4.5 features fully prognostic terrestrial carbon and nitrogen cycling which comprises all vegetation, litter, and soil organic matter pools (Oleson et al., 2013). Each grid cell is composed of multiple, independently represented land use classes. Each class has its own set of plant functional types (PFTs), snow and soil columns.

Vegetation comprises fifteen different PFTs, which are classified into three different phenological groups: evergreen, seasonal-deciduous, and stress-deciduous. Fourteen of these PFTs follow the C3 photosynthetic pathway (11 tree, 2 grasses and crops) and one the C4 path (warm grasses). Altogether, 20 carbon and 19 nitrogen pools per PFT represent carbon (C) and nitrogen (N) in vegetation. C and N are tracked for leaf, live stem, dead stem, live coarse root, dead coarse root, and fine root pools and corresponding storage pools representing, respectively, short-term and long-term storage of non-structural carbohydrates and labile nitrogen.

Decomposition of fresh litter material (including C and N) into progressively more recalcitrant forms of soil organic matter is represented as a cascade of transformations between decomposing coarse woody debris, litter, and soil organic matter pools. Depending on the C:N ratios of involved pools and the amount of carbon lost by respiration, each transformation can generate either a source or a sink of new mineral nitrogen.

Steps that result in an uptake of mineral nitrogen (e.g., immobilization fluxes) are subject to rate limitation, depending on the availability of mineral nitrogen and the sum of nitrogen demands from immobilization, photosynthesis, nitrification, and denitrification. If mineral N is less than the sum of these demands, fluxes are downregulated in order to match N supply. We note that the "Relative Demand" downregulation in CLM4.5 has recently been shown to be inaccurate in several tropical (Zhu et al., 2016b), tundra (Zhu et al., 2016a), and grassland (Zhu et al., 2017) systems. In addition to the cycling of nitrogen within the plant – litter – soil organic matter system, CLM represents external sources, including atmospheric deposition and biological nitrogen fixation. CLM also represents other N sinks not included in this budgeting, including leaching and losses in fire.

Photosynthesis in C3 and C4 plants is based on Farquhar et al. (1980) and Collatz et al. (1992), respectively. The maximum rate of carboxylation at 25°C varies with an assumed static foliage nitrogen concentration and specific leaf area and is a PFT specific parameter. It is assumed that leaf nitrogen and sunlight decrease exponentially with cumulative leaf area index from the canopy top to bottom. Accordingly, the carboxylation rate and other photosynthesis parameter decrease exponentially within the canopy. Leaf level photosynthesis is scaled to the canopy level by integration over the total leaf area. This is done separately for sunlit and shaded leaves and by considering the exponential scaling.

The allocation of carbon and nitrogen is determined in the following steps: First, gross primary productivity (GPP) is calculated under the assumption of unlimited nitrogen supply. Then, from GPP the maintenance respiration demand is subtracted. Following this, the actual nitrogen supply is compared against the nitrogen demand and GPP, if necessary, downregulated. Finally, the available carbon is either utilized for plant growth and growth respiration or stored for growth in the subsequent years. Ghimire et al. (2016) recently presented an improved scheme which avoids this instantaneous downregulation by N limitation.

### 2.1.1 Carbon isotope discrimination during photosynthesis in CLM4.5

Isotopic ratios are usually reported as deviation from a standard material:

$$\delta^{13}C = \left( \frac{R_{sample}}{R_{std}} - 1 \right) \cdot 1000 \tag{1}$$

where $R_{sample}$ and $R_{std}$ denote the $^{13}C/^{12}C$ molar ratios of the sample and the standard material. Isotopic fractionation

factors, $\alpha$, are here defined as the ratio of the carbon isotope ratios in reactant to product (Farquhar et al., 1989) (We note that $\alpha$ is also defined by the ratio of product ratio to reactant ratio in the literature). The fractionation factor for photosynthetic assimilation of $CO_2$ from canopy air is then:

$$\alpha_{psn} = R_{air}/R_{GPP} \tag{2}$$

where $R_{air}$ and $R_{GPP}$ denote the $^{13}C/^{12}C$ molar ratios in the canopy air and in the resulting gross primary productivity flux

incorporated in the plant material. $\alpha$ larger than unity results in a discrimination against the heavier isotope and therefore to a depletion of $^{13}C$ in GPP and plant material compared to air. Discrimination is also expressed by $\Delta_i$, the deviation of $\alpha$ from unity and here multiplied by 1000 for conformity with the $\delta^{13}C$ notation ($\Delta_i = (\alpha - 1) \cdot 1000$).

Photosynthesis in CLM4.5 and, embedded in this process, photosynthetic discrimination, are implemented in two steps following Farquhar et al. (1989). Step 1), given that no enzymatic fractionation is of relevance, the diffusion of $CO_2$ across

the leaf boundary layer and into the stomata, is associated with a kinetic isotope effect of $a$=4.4. During step 2), enzymatic fixation, the effect on C3 plants is $b$=27. These two steps are additive and result in the leaf-level fractionation factor; however, note that in the case of C4 plants only step 1) is of relevance. The CAM photosynthetic pathway is not considered in the model. The leaf-level fractionation factors ($\alpha_{psn}$) of C3 and C4 plants are defined as:

C4 plants:

$$\alpha_{psn} = 1 + \frac{a}{1000} \tag{3}$$

C3 plants:

$$\alpha_{psn} = 1 + \frac{a + (b-a)\frac{c_i^*}{c_a}}{1000} \tag{4}$$

where $c_i^*$ and $c_a$ represent the intercellular and atmospheric concentration of $CO_2$ (mol/mol)), respectively. This results in an isotopic discrimination between assimilated plant material ($\delta^{13}C_{\text{plant}}$) and atmospheric $CO_2$ ($\delta^{13}c_a$) expressed in permil units

(Farquhar et al., 1989) for C3 plants as:

$$\Delta_i = (\delta^{13}c_a - \delta^{13}C_{\text{plant}})/(1 + \delta^{13}C_{\text{plant}}/1000) \tag{5}$$

and can be approximated by $\Delta_i = a + (b-a)c_i^*/c_a$ (Farquhar et al., 1989). Fractionation factors for all other land biosphere fluxes are set to unity and, thus, no further discrimination occurs in the land model.

The kinetic isotope effect during $CO_2$ fixation is constrained by $c_i^*$ which, in turn, depends on the net carbon assimilation during photosynthesis ($a_n$; $\mu$mol m$^{-2}$ s$^{-1}$). The asterisk in $c_i^*$ denotes the consideration of nitrogen down-regulation in the photosynthesis calculation (for details, see Oleson et al., 2013) which is implemented as follows:

$$c_i^* = c_a - a_n(1 - f_{dreg})\frac{(1.4g_s) + (1.6g_b)}{g_b g_s} \tag{6}$$

where $f_{dreg}$ is the downscaling factor due to nitrogen limitation, $g_b$ is leaf boundary layer conductance, and $g_s$ is leaf stomatal conductance for water ($\mu$mol m$^{-2}$ s$^{-1}$).

In CLM4.5, assimilation is calculated before nitrogen limitation is considered. Thus, $c_i$ in the photosynthesis module is different from $c_i^*$. $c_i$ follows from Eq. 6 with $f_{dreg} = 0$. In other words, plants photosynthesize carbon at a potential, nitrogen unlimited rate. Photosynthesis $a_n$ and, in turn, stomatal conductance $g_s$ are not directly affected by nitrogen limitation. The flow of water and carbon through the stomata is simulated to occur by molecular diffusion and a ratio of 1.6 applies between the diffusivity of water and $CO_2$, while a diffusivity ratio of 1.4 is assumed for the leaf boundary layer (Oleson et al., 2013). Stomatal conductance, $g_s$, itself is linearly related to the product of leaf net photosynthesis ($a_n$) and the relative humidity at the leaf surface, $R_h$, and the inverse of the $CO_2$ concentration at the leaf surface, $c_s$ (Ball et al., 1987; Collatz et al., 1991; Oleson et al., 2013):

$$g_s = m \cdot \frac{R_h}{c_s} \cdot a_n + b \tag{7}$$

where $b$ is a minimal conductance as a function of soil water availability, and $m$ a constant parameter, defining the slope between $a_n$ and $g_s$. $m$ is set to 9 for C$_3$ plants and to 4 for C$_4$ plants (Oleson et al., 2013) Both assimilation and stomatal conductance are downregulated by a drought stress factor which depends on soil moisture availability and can vary between 0 and 1. Nitrogen limitation and water stress can therefore result in a downregulation of $c_i^*/c_a$ and thus to a reduction in the carbon isotope discrimination of C3 plants in CLM4.5.

Equation 6 represents the constraint that $CO_2$ consumption by photosynthesis ($A = a_n \cdot (1 - f_{dreg})$) equals the transport of $CO_2$ through the leaf boundary layer and the stomata into the stomatal cavity. The latter is the product of the concentration gradient between the atmosphere and the stomatal cavity ($c_a - c_i^*$) and an overall conductance for $CO_2$ ($g_{CO_2} = (g_b g_s)/(1.4g_s + 1.6g_b)$) ($\mu$mol m$^{-2}$ s$^{-1}$). Thus, we can also write:

$$A = g_{CO_2} \cdot (c_a - c_i^*) = g_{CO_2} \cdot c_a \cdot (1 - c_i^*/c_a) \tag{8}$$

In the following sections, we will, for simplicity and for consistency with the isotopic literature and the LPX description, omit the superscript of $c_i^*$ and denote $c_i^*$ also by $c_i$.

## 2.2 LPX-Bern 1.3

The Land surface Processes and eXchanges (LPX-Bern 1.3) model (Spahni et al., 2013; Stocker et al., 2013; Saurer et al., 2014; Churakova (Sidorova) et al., 2016b; Ruosch et al., 2016; Keel et al., 2016) is based on the Lund-Potsdam-Jena (LPJ) dynamic global vegetation model (Sitch et al., 2003, see also Joos et al. 2001; Gerber et al. 2004; Strassmann et al. 2008). LPX combines process-based, large-scale representations of terrestrial vegetation dynamics, the dynamics of terrestrial carbon and nitrogen stocks and fluxes, and land-atmosphere exchanges of water, carbon dioxide, methane, nitrous oxide, and carbon and water isotopes in a modular framework.

Following the assimilation of multiple experimental constraints, such as net primary productivity (NPP) (Olson et al., 2013) and the seasonal fraction of absorbed photosynthetically active radiation (Gobron et al., 2006), several key model parameters were updated. Note that no $\delta^{13}C$ observational data, e.g., from tree-rings or atmospheric samples, was used as a constraint in the assimilation. A list of the updated parameters and their values can be found in Tab. 1.

Each grid cell in LPX is subdivided into different land use classes (areas with natural mineral soils, peatland, other wetlands, cropland, pasture, urban). LPX simulates the distribution of PFTs based on bioclimatic limits for plant growth and regeneration, and plant-specific parameters that govern plant competition for light, water (Sitch et al., 2003), and nitrogen (Xu-Ri and Prentice, 2008; Xu-Ri et al., 2012). Here, seven generic tree PFTs all following the C3 photosynthetic pathway together with C3 grasses/forbs, and C4 grasses are considered on natural non-peatland areas. The two PFTs on cropland and pastures have the same properties as the C3 and C4 grasses on natural land and grow depending upon climatic conditions. On peatland, two PFTs (C3 graminoids and Sphagnum moss) grow. Seven carbon and nitrogen pools per PFT represent leaves, sapwood, heartwood, fine roots, aboveground leaf litter, aboveground woody litter, and belowground litter. Separate soil organic carbon and nitrogen pools receive input from litter of all PFTs. N input by biological N fixation is implied by maintaining prescribed C to N ratios in litter and soil pools. Inorganic soil nitrogen pools receive input from litter and soil decomposition and through atmospheric N deposition and are subject to plant uptake, leakage, and gaseous losses (Xu-Ri and Prentice, 2008).

Photosynthesis and stomatal control in LPX is described by Haxeltine and Prentice (1996b) as summarized elsewhere (Keel et al., 2016; Sitch et al., 2003). The equations for water supply from soil and transpiration, assimilation, and canopy conductance are solved simultaneously by varying the ratio $c_i/c_a$, also termed $\lambda$, to yield self-consistent values for these properties.

Total day time net photosynthesis, $A_{dt}$, is modeled following Collatz et al. (1991, 1992) which is a Farquhar model (Farquhar et al., 1980) generalized for global modeling purposes (for details, see Haxeltine and Prentice, 1996a). $A_{dt}$ is a function of daily radiation, temperature, day length, and atmospheric $CO_2$ partial pressure and $\lambda$. $A_{dt}$ is computed from a formulation which

gives a gradual transition between light-limited and Rubisco-limited rate of photosynthesis. The amount of photosynthetically active radiation absorbed by the entire canopy increases with the modelled leaf area index following Beer's law and is used to compute the light-limited photosynthesis rate. The N content and Rubisco activity of leaves are assumed to vary seasonally and with canopy position in a way to maximize $A_{dt}$.

Canopy conductance, $g_c$, is linked to $A_{dt}$ through

$$g_c = g_{min} + \frac{1.6 A_{dt}}{c_a(1-\lambda)} \tag{9}$$

where $g_{,min}$ is a PFT specific minimum canopy conductance.

Daily transpiration of water is calculated for each PFT as the minimum of a plant- and soil limited supply function ($E_{supply}$) and the demand for transpiration ($E_{demand}$). $E_{supply}$ is the product of root-weighted soil moisture availability and a maximum water supply rate that is equal for all PFTs. $E_{demand}$ is calculated following the empirical relation between evaporation efficiency and surface conductance of Monteith (1995):

$$E_{demand} = E_{eq}\alpha_m \left[1 - \exp\left(\frac{-g_c\phi}{g_m}\right)\right] \tag{10}$$

where $E_{eq}$ is the equilibrium evaporation rate, dependent on temperature and radiation, $g_m$ and $\alpha_m$ are empirical parameters, $g_c$ is the canopy conductance, and $\phi$ the fraction of present foliage area to ground area (i.e., projected leaf area). Eq. 10 is solved for $E_{demand}$ using the non-water-stressed potential canopy conductance which is calculated using Eq. 9 and a fixed ratio $\lambda$ to compute $A_{dt}$. $\lambda$ is set equal to 0.8 following Sitch et al. (2003) to approximate non-water-stressed conditions and as a starting value for the iterative computation of carbon assimilation and transpiration under water shortage. In case of water-stressed conditions when $E_{demand}$ exceeds $E_{supply}$, canopy conductance and photosynthesis are down-regulated; $E_{demand}$ is set to $E_{supply}$ and Eq. 10 is solved for $g_c$. Knowing $g_c$ and $c_a$, $\lambda$ is varied in the photosynthesis module until the following relationship is satisfied:

$$A_{dt}(\lambda) = \frac{g_c - g_{min}}{1.6} c_a(1-\lambda) \tag{11}$$

NPP is downregulated on the daily model time step if N-demand exceeds N-availability from the inorganic soil nitrogen pools. Daily NPP is integrated over a year and allocated annually to vegetation. Thus, in contrast to CESM there is no immediate feedback of nitrogen limitation on isotopic discrimination on a daily time scale, but there is a long-term feedback by annual changes in vegetation structure and, in turn, photosynthesis and carbon assimilation.

### 2.2.1 Carbon isotope discrimination during photosynthesis in LPX

The isotopic discrimination during assimilation is calculated on a daily time step following Farquhar et al. (1989). Thus, the same discrimination formulations as in CESM (Eq. 3 and 4) are used in LPX. We recall, however, that the intercellular $CO_2$ concentration is, unlike in CLM4.5, not downregulated by nitrogen limitation in LPX. For sphagnum mosses, discrimination

during assimilation is fixed to -30‰. As in CESM, no further discrimination is assumed in LPX. In earlier LPX applications, $\delta^{13}C$ was implemented following Scholze et al. (2003) and discrimination modeled following Lloyd and Farquhar (1994). Here, we adjusted the discrimination formulations and used instead the simpler formulations of Farquhar et al. (1989) for consistency with CLM4.5 and the computation of iWUE as described below. We note that our conclusions do not depend on this choice. LPX yields similar $20^{th}$-century changes in discrimination and iWUE for both formulations. However, simulated

$\delta^{13}C$ of carbon assimilated by C3 trees is generally less negative (by about 2‰) when applying the Lloyd and Farquhar (1994) formulation and agreement with leaf $\delta^{13}C$ is less favorable.

### 2.3 Spin-up and transient simulations and model forcings

**CLM:** Starting with empty pools, the CLM model is brought into equilibrium with the following steps: 1) accelerated decomposition of soil organic matter with perpetual 1850 Common Era (CE) forcing (1000 model years), 2) normal decomposition

with perpetual 1850 CE forcing (500 model years), 3) transition to 1900 CE conditions (100 model years) and 4) transient simulation over the $20^{th}$ century (1900-2005). During steps 1) and 2), both default model options, atmospheric $\delta^{13}C$ is held on a constant value of -6‰. The first step, based on the accelerated decomposition technique by Thornton and Rosenbloom (2005), accelerates the equilibration of the soil carbon pools. During step 3) the model is adjusted to 1900 CE conditions by prescribing transient atmospheric $pCO_2$ and $\delta^{13}C$ (years 1800-1900) together with CRU-NCEP climate forcing data (repeated

years 1901-1920; Viovy, 2011). Step 4), the $20^{th}$-century simulation, is run with both transient atmospheric $pCO_2$ and $\delta^{13}C$ and climatic forcing data (years 1901-2005; Viovy, 2011). In both steps 3) and 4), atmospheric $CO_2$ is prescribed following Law Dome data (Etheridge et al., 1996; MacFarling Meure et al., 2006) and atmospheric samples; prescribed atmospheric $\delta^{13}C$ is a combination of data from Rubino et al. (2013) and, from 1993 on, White et al. (2015) (Fig. 1). Land use area and change is prescribed following Hurtt et al. (2006).

**LPX:** The LPX model is forced with CRU TS3.23 climate data (Harris et al., 2014) and the same atmospheric $CO_2$ and $\delta^{13}C$ data is prescribed as in CLM4.5 . Land use maps are prescribed following Hurtt et al. (2006). After the spin-up procedure, the model is adjusted to 1900 CE conditions by applying transient atmospheric $pCO_2$ and $\delta^{13}C$ (years 1765-1900) together with recycled climate data (years 1901-1930). Then, the $20^{th}$ century is simulated with transient atmospheric $CO_2$ and $\delta^{13}C$ and climatic forcing data (years 1901-2007).

**Factorial runs** were performed in addition to the standard simulation. Each setup is the same as for the $20^{th}$-century simulations, however with one driving factor held constant: (i) *cCLIM*, constant climate (repeated years 1901-1920 for CLM4.5 and 1901-1930 for LPX). (ii) *cCO2*, constant atmospheric $CO_2$ concentrations, only done with LPX. (iii) *cNDEP*, constant atmospheric nitrogen deposition, only done with LPX. (iv) *cLU*, constant land use, only done with LPX. The difference between

the factorial simulation and standard simulation is attributed to the driving factor that was held constant in the factorial run. An interaction term is computed from the difference between the change simulated in the standard simulations and the sum of all attributed changes.

The two models are forced with two slightly different climate data sets as CLM4.5 is run on a sub-hourly time step, while a daily time step is used in LPX. CRU NCEP, used to force CLM4.5, is a combination of the CRU TS3.2 monthly climatology (resolution of $0.5^o$x$0.5^o$) and the NCEP reanalysis product with a time step of six hours ($2.5^o$x$2.5^o$). The results of the factorial runs, presented later, suggest generally a small influence of this difference in forcings on simulated discrimination and iWUE. We further investigated the potential impact of using two different climate data products. The CRU NCEP data are interpolated on the $1^o$x$1^o$ LPX grid and integrated to monthly values for use in LPX. LPX is run with the modified CRU NCEP and, as usual, with CRU-TS3.22 in the standard model setup where climate is changing transiently and the factorial setup with "constant climate" (cCLIM). Difference in simulated changes in discrimination and iWUE are small for both cases. In conclusion, the large model differences, presented in the result section, are not caused by differences in climate input data.

## 2.4 Observations

Two different observational datasets are used to evaluate the model simulations. The first dataset is a compilation of $\delta^{13}$C time series measured on tree rings (for an overview, see Tab. A1). The time series cover at least the years 1900–1985 (if possible 2005) and comprise a wide range of different tree species and locations. Since absolute values are highly variable, depending on factors such as, e.g., species, tree age or location (McCarroll and Loader, 2004), we refrain from a direct comparison of absolute values and focus on the changes over the course of the century.

The second dataset is a compilation of $\delta^{13}$C measurements on leaves (Cornwell et al., 2016). The dataset comprises data of C3, C4 as well as CAM plants, however, to ensure comparability between models and observations only data of C3 plants are used.

The observational $\delta^{13}$C data are compared with ten-year averages of available model output to remove interannual variability from the model data. For CESM, data are compared to the simulated grid-cell average of the $\delta^{13}$C signature of the "live stem" pool. This pool has a very fast turnover time (0.7/year; Oleson et al., 2013) and no post photosynthetic fractionation occurs in the model during the allocation of assimilated carbon. Modeled $\delta^{13}$C in the live stem pool represents thus on average the $\delta^{13}$C signature of the leaves of C3 trees. For LPX the $\delta^{13}$C signature of daily GPP, $GPP_{(t,PFT)}$, is averaged for all tree PFTs within a grid cell and the land use class "mineral soil":

$$\delta^{13}C_{av} = \sum_t \sum_{PFT} \frac{GPP_{(t,PFT)} \times \delta^{13}C_{(t,PFT)}}{\sum_t \sum_{PFT} GPP_{(t,PFT)}} \tag{12}$$

$t$ is time and the sum is over 10 years. $PFT$ is the index for PFTs and the sum is over all tree PFTs (without including grasses).

The evolution of $\delta^{13}$C in (C3) trees is a consequence of both the changes in atmospheric $\delta^{13}$C as well as changes in fractionation, related to changes in $c_i/c_a$ (Eq. 4) and thus to a combination of physiological adjustments of trees and ecosystems to $CO_2$, climate, and other environmental changes. The atmospheric $\delta^{13}CO_2$ record shows a century-scale decrease in $\delta^{13}$C caused by the input of isotopically-depleted $CO_2$ from deforestation and fossil fuel burning. This century-scale decrease, known as the Suess Effect (e.g., Tans et al., 1979), is a precisely known external forcing. To account for this effect, we focus on the isotopic discrimination $\Delta_i$ between assimilated plant material and atmospheric $CO_2$ (Eq. 5).

Changes in discrimination of C3 plants are related to the ratio of photosynthesis to the conductivity of carbon through the stomata, two relevant tree physiological parameters regulating carbon and water fluxes. The ratio of net daytime photosynthesis to stomatal conductance for water vapor $(A/g_{H_2O})$ is also known as iWUE and discussed in the literature (e.g., Scheidegger et al., 2000; Saurer et al., 2004, 2014). Using Eq. 5 and 6 for CLM4.5 and Eq. 11 for LPX, the definition of iWUE, and a ratio of 1.6 between the conductance of water and $CO_2$ we obtain:

$$\text{iWUE} = \frac{A(t)}{g_{H_2O}(t)} = c_a(t) \frac{b - \left[ \frac{\delta^{13}c_a(t) - \delta^{13}C_{\text{plant}}(t)}{1 + \frac{\delta^{13}C_{\text{plant}}(t)}{1000}} \right]}{1.6(b-a)} \tag{13}$$

In the following, the symbol $\Delta$ is used to denote a temporal change (not to be confused with the symbol for discrimination $\Delta_i$). For example, $\Delta$iWUE is the difference between iWUE at time $t_2$ and time $t_1$. We approximate $\Delta$iWUE to better understand how $\Delta$iWUE depends on discrimination, changes in discrimination and atmospheric $CO_2$. The $CO_2$ concentration $c_a(t_2)$ is expressed as the sum of $c_a(t_1)$ and the change in $CO_2$, $\Delta c_a$. Similarly for the difference between the atmospheric and plant $\delta^{13}$C ($d = \delta^{13}c_a - \delta^{13}C_{\text{plant}}$), we write $d(t_2) = d(t_1) + \Delta d$. The term $1 + \frac{\delta^{13}C_{\text{plant}}}{1000}$ is denoted as $k$. Variations in $k$ are of order 1‰ and have a small influence on $\Delta$iWUE. Then, it follows for the difference in iWUE, $\Delta$iWUE:

$$\Delta\text{iWUE} \cong +\frac{\Delta c_a \cdot b}{q} - \frac{c_a(t_2)}{q} \cdot \frac{\Delta d}{k(t_2)} - \frac{\Delta c_a}{q} \cdot \frac{d(t_1)}{k(t_2)}$$
$$\cong 0.7467 \cdot \Delta c_a - \frac{1}{36.16} \cdot [c_a(t_2) \cdot \Delta\Delta_i + \Delta c_a \cdot \Delta_i(t_1)] \tag{14}$$

$q$ denotes the term $1.6(b-a)$ and is with b=27 and a=4.4 equal to 36.16. The approximation given in Eq. 14 holds well within 0.1%.

Equation 14 shows that $\Delta$iWUE is linearly related to the isotopic discrimination at time $t_1$, $\Delta_i(t_1)$ and the change in discrimination between $t_1$ and $t_2$, $\Delta\Delta_i$. In other words, the change in iWUE depends both on the change in discrimination as well as on the initial magnitude of the discrimination. The sensitivity in $\Delta$iWUE is about 7 times larger for a unit change in $\Delta\Delta_i$ compared to a unit change in $\Delta_i(t_1)$ when assuming a change in $CO_2$ from 300 to 350 ppm as reconstructed for the period 1900 to 1980. Under increasing $CO_2$, discrimination and thus iWUE may change. The larger the decrease in discrimination the larger the increase in iWUE and vice versa.

The relative change in iWUE may be expressed, again very accurately, as follows:

$$\frac{\Delta \text{iWUE}}{\text{iWUE}(t_1)} \simeq \frac{\Delta c_a}{c_a(t_1)} - \frac{c_a(t_2)}{c_a(t_1)} \times \frac{\Delta \Delta_i}{(b - \Delta_i(t_1))} \tag{15}$$

The relative change in iWUE scales in proportion with the relative change in atmospheric $CO_2$ in the absence of a change in discrimination. It also scales with changes in discrimination. For typical values, a unit change in discrimination changes the relative change in iWUE by 50 to 100%. Further, the relative change in iWUE depends on the difference between b (=27) and the isotopic discrimination $\Delta_i(t_1)$ ($\approx$20). This difference is typically 7 and so a unit change in $\Delta_i(t_1)$ affects the second term in equation 15 by about 14% and the relative change in iWUE typically between 0 and 10%. We note that equation 15 is non-linear and so the numerical values given here are illustrative.

Equation 13 is evaluated for LPX and CESM by using the annual-mean $\delta^{13}$C signature of C3 plants for CESM (live stem) and LPX (GPP) within each pixel. We note that in CLM4.5 leaf boundary conductance is also explicitly modeled, and $g_{H_2O}$ in Eq. 13 reflects thus the ratio of photosynthesis to the total conductivity through the leaf boundary layer and the stomata of water, whereas in LPX $g_{H_2O}$ reflects the resistance by the stomata only. In CLM4.5, the ratio between the conductivity for $H_2O$ and $CO_2$ may vary between 1.4 ($g_s \gg g_b$) to 1.6 ($g_b \gg g_s$) (Eq. 6). Here, we have assumed for simplicity a factor of 1.6 for CLM4.5 , LPX and observational data. This simplification introduces an uncertainty of up to 11% in computed trends for CLM4.5 .

Turning to the tree-ring data, there are isotopic offsets between assimilated material and cellulose. The leaf-level model for fractionation is more representative for the bulk matter rather than a specific chemical compound such as cellulose. We correct for this isotope offset between cellulose and total organic matter ($\delta^{13}$C$_{\text{plant-corrected}}$ = $\delta^{13}$C$_{\text{plant cellulose}}$ - offset) following Saurer et al. (2014) and using an offset of 1.2‰ for all tree species. This correction has a relatively small influence: as discussed above, a change in $\delta^{13}$C$_{\text{plant}}$ by 1‰ affects the relative change in iWUE by about 10%.

In the following, we will present and discuss annual or multi-annual averages of the related variables $\delta^{13}$C, $c_i/c_a$, discrimination $\Delta_i$, iWUE, and $A/g$. These values represent, similarly to Eq. 12, weighted-averages. The time-varying carbon assimilation rate, or more generally, the assimilation rate multiplied by the fraction of assimilates allocated to the carbon reservoir under consideration (e.g. leaves, tree ring) act as the weighting factors.

# 3  Results

## 3.1  Primary productivity and carbon pools

We first compare GPP, transpiration, vegetation and soil carbon stocks of the two models (Tab. 2, Fig. 2), as well as model results to observation-based estimates of vegetation carbon (Fig. 3; Carvalhais et al., 2014). Both models show highest GPP per unit land area in the tropics and low GPP in arid regions. Overall, the GPP patterns are similar between the two models.

However, the tropical maxima in GPP are about a third larger in CLM4.5 than in LPX. Simulated transpiration is similar for the two models.

Estimates of total vegetation carbon, including below and aboveground biomass, were derived by Carvalhais et al. (2014) from a collection of estimates for pan-tropical regions and for northern and temperate forests based on radar remote-sensing retrievals. The global carbon inventory of vegetation is overestimated by a factor of two by CLM4.5 and slightly underestimated by LPX compared to the estimate of Carvalhais et al. (2014) (Tab. 2). CLM4.5 simulates too much vegetation carbon in northern mid-to-high latitudes and overestimates vegetation carbon stock by more than a factor of two in tropical forests (Fig. 3) (Negrón-Juárez et al., 2015; Koven et al., 2015a). Correspondingly, we expect a negative bias in the global-mean $\delta^{13}$C signature of vegetation in CLM4.5. LPX simulates vegetation carbon stocks in relatively good agreement with observation-based estimates (Fig. 3). Both models underestimate vegetation carbon in parts of southeastern Asia. Overall, the spatial correlation ($r$) between modeled and observation-based vegetation carbon is 0.83 for CLM4.5 and 0.85 for LPX.

The global carbon inventory in soils of LPX is 1700 GtC, while CLM4.5 simulates a twice larger soil C inventory of 3900 GtC (Tab. 2). This larger inventory mainly stems from the large carbon stocks simulated by CLM4.5 in peatland and permafrost regions (Koven et al., 2013, 2015b) and in northern mid-latitude regions (Fig. 2). CLM4.5 includes formulation for carbon storage in deep soils. The large soil carbon inventory of CLM4.5 is in contrast to that of its predecessor, CLM4 (Oleson et al., 2010), which severely underestimated the global soil carbon reservoir by 50% and more (Todd-Brown et al., 2013; Anav et al., 2013; Tian et al., 2015), especially at the Northern high latitudes (Foereid et al., 2014). LPX, in the version applied here, does not include formulations for carbon storage below 2 m. An evaluation of peat and permafrost carbon stocks simulated by LPX is provided by Spahni et al. (2013). Data on soil carbon have particularly high uncertainties with notable discrepancies between different data sets (e.g.; Carvalhais et al., 2014; Hugelius et al., 2014).

Simulated GPP and carbon stocks change over the industrial period in response to increasing atmospheric $CO_2$, land use, climate change and nitrogen deposition. Both models indicate almost everywhere an increase in GPP over the course of the $20^{th}$ century (Fig. 2). Globally-averaged GPP increases from 148 GtC/yr in 1900 to 168 GtC/yr in 2000 for CLM4.5 and from 132 to 149 GtC for LPX. In the case of CLM4.5 the increase is especially pronounced in the tropics, particularly in Indonesia (Fig. 2), whereas LPX simulates a large increase in GPP in Australia.

Simulated $20^{th}$ century changes in annual-mean transpiration are generally small in both models, except in Australia for LPX and in parts of Latin America for CLM4.5. Generally an increase in transpiration is found in boreal and temperate forest regions in both models. Transpiration is slightly decreasing in LPX and slightly increasing in CLM4.5 in most tropical forest regions.

For vegetation carbon, both models show large areas with decreasing C storage in the mid latitudes (Europe, North America) and parts of the tropics, especially South-East Asia, in response to deforestation (Fig 2). Areas with a negative vegetation C balance are more widespread in LPX than CLM4.5 in tropical and subtropical Africa and South America. LPX simulates a relatively small increase in the vegetation stocks of remaining "natural" tropical forests, whereas CLM4.5 shows considerable increase in vegetation C in the remaining tropical forests of Africa, America, and Indonesia. This large increase in CLM4.5 is related to the strong stimulation of GPP and the long overturning time scale of vegetation carbon in tropical forests. As a result,

the decrease in globally-integrated vegetation C storage is more than two times larger for LPX than for CLM4.5 (-55 GtC to -21 GtC, Tab. 2, Fig. 1b).

For soil C (Fig. 2), CLM4.5 shows a general increase over the $20^{th}$ century which is especially pronounced in the tropics. For LPX the picture is more heterogeneous, with increases (decreases) in South America, Africa, Australia and the high latitudes (parts of the US and Eurasia, India). Globally, the models simulate an increase of 45 (CLM) and 19 (LPX) GtC, respectively (Tab. 2, Fig. 1). Overall, global carbon storage in the land biosphere increases by 34 GtC in CLM4.5 and decreases by 35 GtC in LPX over the $20^{th}$ century. Independent estimates of the difference between anthropogenic emissions and ocean uptake plus increase in atmospheric $CO_2$ suggest an increase in land carbon storage by roughly 20 GtC (Le Quéré et al., 2015) over the $20^{th}$ century.

## 3.2 Simulated $\delta^{13}$C of primary productivity, vegetation and soil carbon: spatial distributions and $20^{th}$-century trends

Global maps of the $\delta^{13}$C signature of GPP, vegetation and soil carbon in the year 2000 and related $20^{th}$-century changes are shown in Fig. 4. We note that isotopic fractionation, typically in the order of 20‰ for C3 plants, is only 4.4‰ for C4 plants, growing in high temperature, high light intensity regions. The grid-cell average $\delta^{13}$C signature of GPP therefore reflects to a large extent the occurrence or absence of C4 plants. Vegetation distribution is prescribed in CLM4.5 and simulated in LPX, CAM plants are not represented in both models. Both models show less negative isotopic signatures in arid and semi-arid low-latitude regions. In comparison to CLM4.5, LPX simulates in the year 2000 much larger areas with $\delta^{13}$C larger than -16‰. These areas include central and western parts of the US, Argentina, large parts of Southern Asia and parts of southern Australia. This is related to a larger share of C4 plants in these regions in LPX than in CLM4.5.

Atmospheric $\delta^{13}$C decreases by about 1.3‰ over the $20^{th}$ century (Fig. 1a). Thus, without a change in isotopic discrimination, we would expect an equal decrease in the GPP signature. Accordingly, CLM4.5 and LPX show a decrease in $\delta^{13}$C of GPP in most areas (Fig. 4).

In LPX, large $20^{th}$-century increases in the $\delta^{13}$C signature of GPP of up to 5‰ are simulated in many low and mid latitude areas indicating an expansion of C4 plants under warming and rising $CO_2$ as well as the influence of land use. Processes such as the conversion of forests into C4 pastures in the tropics can substantially affect the isotopic signature of total terrestrial vegetation at a location (Townsend et al., 2002; Kaplan et al., 2002). In CLM4.5, the influence of the negative atmospheric $\delta^{13}$C trend is offset mainly by changes in discrimination in the eastern US, Europe and East Asia, resulting in slightly positive $\delta^{13}$C changes in these areas (Fig. 4). Note that, in contrast to LPX, crops are treated as C3 plants in CLM4.5 (Oleson et al., 2013). This explains the land use-related, differing trends between models in regions such as North and South America, Africa and South-East Asia. Distribution and $20^{th}$-century change in $\delta^{13}$C of vegetation and soil are very similar to the distribution of $\delta^{13}$C of GPP. However, temporal trends are generally smaller in the soil carbon pools than for GPP.

The average $\delta^{13}$C signature of the globally-averaged pool of vegetation carbon is more negative than that of soil and leaf carbon (Fig. 1). This reflects the share of organic matter assimilated by the C4 path versus that by the C3 path in these globally-averaged pools. Generally, the share of C4-derived organic matter is much smaller than the contribution by C3-derived matter.

Correspondingly the average $\delta^{13}$C signature of globally-averaged C pools is strongly negative and closer to that of C3 plants. The contribution of C4-derived organic matter is very small for the globally-averaged vegetation pool and absent for the stem carbon pool, but noticeable for the leaf and soil carbon pool. Further, the globally-averaged leaf $\delta^{13}$C signature decreases in both models over the $20^{th}$ century, mainly in response to changes in the atmospheric $\delta^{13}$C source. Trends are less developed in the other global pools due to their longer residence times.

There is a substantial offset between the models for globally-averaged soil and vegetation pools, with CLM4.5 being around 5‰ more negative (Table 2). In addition to the already mentioned widespread, comparably less negative $\delta^{13}$C signatures simulated by LPX, this can be attributed to the spatial distribution of carbon pools in CLM4.5. As is evident from Fig. 3, CLM4.5 simulates higher vegetation and soil carbon stocks in the high latitudes and tropical forest regions than LPX. Since these regions are characterized by C3 plant cover and thus comparably negative signatures (see Fig. 4), the result is a shift of the global averages towards more negative $\delta^{13}$C in CLM4.5 compared to LPX.

## 3.3 Modelled versus measured leaf $\delta^{13}$C

In this section, we evaluate how well modelled $\delta^{13}$C, and thus isotopic discrimination, of C3 tree species compares with a global compilation of $\delta^{13}$C measurements on leaf material from C3 plants (Cornwell et al., 2016) (Figs 5, 6). Unlike the previous analyses, this comparison is not (or hardly) affected by land use as the comparison is for C3 trees only on unmanaged lands. However, a few caveats apply. We compare grid-cell averages (with GPP or mass as weights; see section 2.4) from all simulated C3 plants with site measurements for individual species. Difference in $\delta^{13}$C are reported for different species. In addition, differences within a species, even growing at the same site, can be as large or even larger as those between species (McCarroll and Loader, 2004; Leuenberger, 2007, and references therein). In addition, $\delta^{13}$C in the canopy air may deviate from the prescribed atmospheric $\delta^{13}$C, representative of tropospheric background air. Deviations may arise due to varying additions of isotopically-depleted respired carbon and of carbon from fossil sources to local air.

The bar plots in Fig. 5 and 6 indicate the mean $\delta^{13}$C of the leaf data and of modeled grid-cell averages for C3 plants at the measurement locations for observations, CLM4.5, and LPX, respectively. On average, $\delta^{13}$C of all 344 leaf samples is -27.54‰ compared to a corresponding average of -26.29‰ for CLM4.5 and -26.14‰ for LPX. This suggests a discrimination of 19.5‰, 18.3‰ and 18.1‰ relative to the atmospheric $\delta^{13}$C signature of approx. -8‰, respectively. According to eq. 4, these values correspond to a ratio between internal and atmospheric partial pressure, $c_i/c_a$, of 0.67 for the leaf data and of 0.63 and 0.61 for CLM4.5 and LPX, respectively. The bias of the models is particularly large in arid regions and larger for LPX than for CLM4.5, whereas good agreement between models and leaf data is found in the remaining regions, e.g., Europe or Alaska (Fig. 6).

The $\delta^{13}$C of C3 plants from CLM4.5 and LPX show similar spatial gradients. Both models reasonably capture the observations, though spatial correlations between data and models are low (CLM: $r$=0.36; LPX: $r$=0.34). Root-mean-square errors (RMSE) between models and data are 2.42‰ (CLM) and 3.22‰ (LPX). Largest discrimination and most negative $\delta^{13}$C is simulated in high northern latitude regions and in tropical moist forests and lowest discrimination in arid regions of inner Asia, southwestern North America, Patagonia, southern Africa, and parts of Australia. Low discrimination and high (less negative)

$\delta^{13}C$ correspond to low stomatal conductance, low leaf internal $CO_2$ partial pressure and therefore reduced assimilation. However, these high $\delta^{13}C$ values are not supported by the leaf measurements, when assuming that the leaf samples are regionally representative (Fig. 6). Maxima in $\delta^{13}C$ are a few permil higher for LPX than for CLM4.5 in these regions. This suggests a stronger downregulation of stomatal conductance by water stress in LPX than in CLM4.5 in these regions.

In summary, $\delta^{13}C$ and, by implication, discrimination of C3 trees is reasonably represented in both models, when considering the caveats discussed at the beginning of this section and the simplicity of the isotopic model approach. However, both models tend to overestimate $\delta^{13}C$ in comparison with the leaf data in many arid regions.

## 3.4   $^{13}C$ – century-scale trend

Next, simulated century-scale trends in the discrimination, $\Delta_i$ (Eq. 5, see also methods) of C3 tree species are discussed and
compared to the corresponding trends derived from 76 tree-ring $\delta^{13}C$ records (Fig. 7 to 10 and Tab. A1). Again grid-cell averages for C3 trees are compared to site data. Similar caveats as discussed for the comparison between leaf measurements and modeled $\delta^{13}C$ apply for this trend analysis. The term "average" refers again to an average over those grid cells with measurements only. 74 of the 76 sites are located in three regions (Fig. 9), with more than half of the sites (51) in Europe, and with only seven records in Asia and 16 records in western North America. For these two regions, the comparison of regional
model and data averages is therefore hampered by the scarcity of site data and the complex topography along the Rocky Mountains, while the "global" average is biased towards Europe.

    On average, discrimination inferred from the tree-ring records varies within a few tens of a permil in the first and second half of the $20^{th}$ century (Fig. 7). There is a transition to less discrimination in the 1940s, with no clear trend before and afterwards. Similar to the measurements, LPX-Bern yields no long-term trend in the discrimination of C3 trees. In contrast,
discrimination of C3 trees simulated by CLM4.5 shows a trend towards smaller values over the $20^{th}$ century. The discrepancy between tree-ring data and CLM4.5 results grows towards the present and is larger than one permil after year 2000.

    Next, we analyse spatial and regional changes in the discrimination of C3 trees by comparing the differences in $\Delta_i$ between the 1980s and 1900s decades. CLM4.5 simulates a decreasing trend in the discrimination $\Delta_i$ of C3 trees almost everywhere where C3 trees grow. In other words, the ratio of internal to atmospheric $CO_2$ partial pressure, $c_i/c_a$, decreases over the
$20^{th}$ century. The simulated decrease in $\Delta_i$ for CLM4.5 is larger than suggested by the tree-ring data. The average change in discrimination ((1980-1989)-(1900-1909)) is -0.36 in the tree-ring data and -0.94 in CLM4.5. The observations show on average a slight positive change in $\Delta_i$ (+0.18) in the North America region and a slight negative change (-0.13) in the Asian region and a relatively large decrease in $\Delta_i$ in Europe (-0.54) (Fig. 9). The three corresponding regional averages for CLM4.5 are between 0.52 and 0.79 more negative. LPX simulates generally small (positive or negative) changes in $\Delta_i$ of C3 trees over
the last century. Exceptions are areas at the margin of C3 tree covered regions such as the southern limit of the boreal tree belt that show a large positive change (Fig. 8). For grid-cells with measurements, the average change in $\Delta_i$ for LPX is with +0.06 very small (Fig. 10). For the three selected regions, LPX simulates changes in $\Delta_i$ that are between 0.45 to 0.54 more positive than suggested by the tree-ring records (Fig. 9). We note that the agreement between tree-ring data and LPX is better for previous and later decades (Fig. 7).

The correlations $r$ between models and observation-derived changes in $\Delta_i$ are relatively low, with values of 0.40 (CLM) and 0.27 (LPX). Here, possible explanations are the strong spatial heterogeneity of both measurements and model simulations combined with a limited number of observational data and the overall small changes in discrimination of e.g. only -0.36 for the average of tree-ring data or that the models do not properly represent photosynthesis and stomatal conductance. The RMSEs of changes in $\Delta_i$ are 0.93 (CLM) and 1.07 (LPX).

The factorial simulations with LPX reveal that average changes in $\Delta_i$ attributable to individual drivers are small. Thus, the relatively small changes in discrimination simulated by LPX are not the result of offsetting influences of different drivers. In detail, a somewhat complex interaction of the driving factors in shaping the change in $\Delta_i$ and non-negligible interaction between climate and $CO_2$ is found. Keeping nitrogen deposition or land use constant has generally a negligible influence on the change in $\Delta_i$ (Fig. 10). On average, a positive change in $\Delta_i$ is attributed to the $CO_2$ change for all sites (+0.03), the European (+0.05) and the Asian sites (+0.09), and a negative change for the North American sites (-0.18). Thus, the increase in $CO_2$ does not cause a general downregulation of $c_i/c_a$ in LPX. The influence of climate is small on average over all sites (+0.07) and for the European sites (-0.08). At the North American sites and the few Asian sites climate change tends to increase discrimination by about 0.7 and 0.4, respectively.

For CLM4.5, the change in $\Delta_i$ attributed to climate change is small, except for the North American sites. There, a positive influence (+0.55) similar as found for LPX is inferred. The other drivers together cause on average a change in discrimination between -0.79 and -1.16 in the three regions. This suggests that the simulated decrease in discrimination and in $c_i/c_a$ in the CLM4.5 runs is mainly linked to increasing $CO_2$ and a corresponding downregulation of $c_i/c_a$ (Eq. 6). The downregulation in CLM4.5 is larger than suggested by the tree-ring records.

Soil moisture and stomatal conductance are coupled and influence each other and discrimination. $20^{th}$ century changes in soil moisture are small in the factorial simulations where climate is kept constant, while $CO_2$ and other forcings vary. For the standard model setup, changes in soil moisture simulated by CLM and LPX are also small in large parts of Europe and Asia, where most of our tree-ring data are located. Despite these small changes in soil moisture, large differences in discrimination changes are found between the two models in these simulations and regions (Fig. 9). This suggests that the primary reason for the model-model difference in simulated $20^{th}$ century changes in discrimination is rooted in the different parameterizations of the photosynthesis-conductance coupling.

## 3.5  $^{13}$C – century-scale trend in iWUE

Of particular importance are changes in stomatal conductance and photosynthetic carbon assimilation since they are the primary controls of plant-atmosphere water and $CO_2$ exchange, carbon assimilation and, ultimately, tree growth (Lambers et al., 2008). Changes in the ratio of these two processes, also termed iWUE (see Eq. 13), thus reflect a change in assimilation, in conductance, or both (Scheidegger et al., 2000; Saurer et al., 2004, 2014).

Similar as in the previous sections, we compare relative changes in iWUE (Eq. 15) from the tree-ring records with model results (Figs. 10, 11, 12). Changes are evaluated between the 1990 decade and 1900 decade as in Saurer et al. (2014).

As shown by Eq. 15, the relative change in iWUE increases equal to the relative change in atmospheric $CO_2$ in the absence of changes in discrimination. This is equal to 23% for the change from (1900-1909) to (1990-1999). A negative change in discrimination, as simulated for CLM4.5, contributes to a further increase in iWUE.

In the standard setup (see also maps), both models show a substantial increase in iWUE over the century of approx. 52% (CLM) and 25% (LPX). The globally averaged iWUE derived from $\delta^{13}$C measurements show a value of 28%, which concurs with previous studies reporting increases in iWUE of 27.8% and approx. 30% for European forests (Saurer et al., 2014; Frank et al., 2015) and 17% for high latitude forests (Trahan and Schubert, 2016). While LPX is in good agreement with the available observations (both globally and regionally), CLM4.5 appears to overestimate $20^{th}$-century changes in iWUE by about a factor of two in Europe and the Asian region.

The drivers behind modeled changes in iWUE can be investigated based on the sensitivity experiments already discussed in the previous section for discrimination. As expected, all simulations show similar changes in iWUE in LPX. The small changes in discrimination simulated by LPX do not strongly affect iWUE and the relative change in iWUE of LPX (24.6%) is close to the relative change in atmospheric $CO_2$ (23%). Simulations with and without climate change with CLM4.5 yield similar results, pointing again to a downregulation of $c_i/c_a$ under raising $CO_2$ by the implemented nitrogen limitation.

## 4  Discussion

This study presents the implementation of $\delta^{13}$C in two global land biosphere models, CLM4.5 and LPX-Bern 1.3 and a global compilation of $20^{th}$ century tree ring $\delta^{13}$C records. Results from isotope-enabled simulations over the industrial period are investigated for regional carbon stocks and carbon isotope signatures (Figs. 1 to 4) and for the global carbon and $^{13}$C budgets (Table 2) of the land biosphere. The performance of the two models in terms of modern isotopic discrimination of C3 plants is evaluated by comparing model results with a global compilation of $\delta^{13}$C measurements on leaf material (Figs. 5, 6). The main focus of the study is on the analysis of $20^{th}$ century changes in isotopic discrimination, $\Delta_i$, in the ratio of $CO_2$ concentrations between the stomatal cavity and the atmosphere, $c_i/c_a$, and in the ratio between assimilation and stomatal conductance, $A/g$, termed intrinsic water use efficiency iWUE. The comparison of model results with observational estimates from the $\delta^{13}$C tree ring records permits us to evaluate the performance of the two models with respect to linkages between photosynthesis and stomatal conductance on the leaf level (Figs. 7 to 12).

The "minimal" discrimination model of Farquhar et al. (1989) was implemented in CLM4.5 and LPX-Bern. This model assumes fixed discrimination for C4 plants and that discrimination by C3 plants is directly proportional to $c_i/c_a$. Genetic species-specific variations in $\delta^{13}$C are not considered here, though they can be considerable (Marshall et al., 2008; Yang et al., 2015). The influence of rooting depth, water transport systems, root-to-leaf distance, leaf morphology, and irradiance (sunlit versus shaded) on discrimination is neglected. However, these factors may affect discrimination. Reported isotopic differences within a species, even growing at the same site, can be as large or even larger as those between species (McCarroll and Loader, 2004; Leuenberger, 2007). We also neglect fractionation between different pools within plants and soils (Wingate et al., 2010;

Brüggemann et al., 2011). Despite these limitations, we provide a comprehensive framework for a model-data comparison and derive new insights into patterns of physiological plant adaptation to $20^{th}$ century environmental changes.

## 4.1 Isotopic signatures, pools and fluxes of the global land biosphere

A high spatial correlation of simulated vegetation carbon with observational estimates is found for both models. However,
CLM4.5 overestimates carbon stocks in vegetation by a factor of two or more in many regions compared to the data of Carvalhais et al. (2014), while data-model agreement is reasonable for LPX (Fig. 3). Global soil carbon stock is 1700 GtC in LPX and much smaller than the 3900 GtC simulated by CLM4.5. Larger soil carbon inventories in CLM4.5 than LPX are mainly simulated in northern mid and high-latitudes (Fig. 2). Only the top two meters of soils are considered in LPX, while CLM4.5 includes deeper soil layers.

Both models feature most negative $\delta^{13}$C in high latitude and in tropical-forest regions (Fig. 4) and isotopically heavier signatures in semi-arid and arid regions, where typically C4 plants are abundant. The global average assimilation-weighted $\delta^{13}$C signature of GPP is -20.6‰ in LPX and -24.9‰ in CLM4.5 in the year 2000. The associated discrimination is 12.4‰ in LPX and 16.7‰ in CLM4.5, respectively. The CLM4.5 estimate is within the range (15.7 to 18.1‰) of previous studies as summarized in Suits et al. (2005) and Scholze et al. (2008). This may point to a too large abundance of C4 plants in LPX.
Differences in soil and vegetation carbon and in C3 and C4 plant abundance affect also the global isotopic budgets of the two models as the relative importance of C4-derived plant and soil material is less in CLM4.5 than in LPX. On global average, the mean $\delta^{13}$C signature of all land biosphere carbon is 4.6‰ more negative in CLM4.5 than in LPX (Table 2).

Turning to $20^{th}$ century changes, both models simulate a wide-spread and strong increase in GPP (Fig. 2), mainly in response to $CO_2$ fertilization under increasing atmospheric $CO_2$. On the other hand, $20^{th}$ century changes in transpiration are generally
small in both models (Fig. 2). The two models bracket independent estimates of the net change in global land biosphere carbon stocks over the last century (20 GtC), with CLM4.5 showing slightly larger uptake (35 GtC) and LPX-Bern 1.3 showing a release of carbon (35 GtC) instead of an uptake.

Land use is treated differently in the two models. In CLM4.5, C3 and C4 species are replaced by a C3 crop when land is converted to cropland. This results in a negative $20^{th}$-century change in $\delta^{13}$C of GPP of several permil (grid cell average) in
tropical and subtropical regions affected by land use (Fig. 4). In contrast, in LPX C3 and C4 crops have identical PFT parameters as C3 and C4 grasses and grow under competition on pasture and cropland. Thus, conversion of tropical and subtropical forests (C3 trees) to cropland or pasture results in an increase of C4 grasses and a decrease of C3 plants. Correspondingly, a positive $20^{th}$-century change in $\delta^{13}$C of up to five permil is simulated in regions undergoing land use changes (Fig. 4). The implication is that for atmospheric $\delta^{13}$C budget analyses (Keeling et al., 1989; Ciais et al., 1995; Joos and Bruno, 1998; Scholze
et al., 2008; van der Velde et al., 2013), the correct crop type (C3 vs C4) should be specified in the models.

## 4.2 $\delta^{13}$C in leaves of C3 plants: modern distribution

The mean and spatial gradients in discrimination of C3 trees simulated by CLM4.5 and LPX are evaluated using the leaf $\delta^{13}$C data of Cornwell et al. (2016). Both models reasonably represent the observation-based distribution in discrimination of C3

trees, though modeled discrimination is on average too small compared to the measurements, in particular in arid regions (Figs. 5, 6). The low discrimination in arid regions may be due to a mismatch in scale between local site conditions and grid-cell averages. Potentially, trees at sites with relatively good growing conditions were selected for the $\delta^{13}C$ analysis, while modeled trees experience grid-cell average soil water conditions.

In applications with the predecessor of LPX (Joos et al., 2004; Scholze et al., 2008) $\delta^{13}C$ was implemented following Scholze et al. (2003) with discrimination modeled following Lloyd and Farquhar (1994). Here, we adjusted the discrimination formulations and use instead the simpler formulations of Farquhar et al. (1989) for consistency with CLM4.5 and with the computation of iWUE from tree-ring $\delta^{13}C$ measurements. We note that our conclusions do not depend on this choice. LPX yields similar $20^{th}$-century changes in discrimination and iWUE for both formulations. However, simulated $\delta^{13}C$ of carbon

assimilated by C3 trees is generally less negative (by about 2‰) when applying the Lloyd and Farquhar (1994) formulation and agreement with leaf $\delta^{13}C$ is less favorable.

The difference between the two implementations arises from additional processes considered in the approach of Lloyd and Farquhar (1994) and different parameter choice. Here, fractionations associated with the diffusion of $CO_2$ from the stomatal cavity to the cell wall, entrance of $CO_2$ in solution at the cell wall, and transport within the cell, as well as photorespiration

are neglected. The discriminations by the first three processes are set to be constant in the earlier implementation following Scholze et al. (2003), neglecting a minor temperature dependency associated with the carboxylation by PEP-c (Lloyd and Farquhar, 1994). The discrimination associated with respiration varies with atmospheric $CO_2$ and the photocompensation point, but is small. In addition, $b$, the discrimination during photosynthetic fixation, is set to 27 in this study and to 27.5 in the earlier implementation. Overall, the consideration of the additional processes and the difference in b result in an approximately

constant offset between the two implementations. This highlights that absolute values of discrimination depend on uncertain model parameters. Agreement or disagreement between leaf and model data (or tree ring and model data) concerning absolute levels of $\delta^{13}C$ (Figs. 5, 6) may not be interpreted as an indication of the performance of the conductance/photosynthesis module.

## 4.3   $20^{th}$ century changes in carbon isotopes and water use efficiency of C3 plants

A set of seventy-six $20^{th}$-century $\delta^{13}C$ tree-ring chronologies, mainly from Europe, boreal Asia and western North America was compiled (Table A1). The $\delta^{13}C$ tree-ring data show on average no or little change in isotopic discrimination (Fig. 7). It remains unclear whether there is an overall small decrease in discrimination over the $20^{th}$ century, given the still limited number of records and the large variability of the averaged and individual $\delta^{13}C$ records. Small or no changes in discrimination of C3 plants imply that $c_i/c_a$ remained roughly unchanged over the $20^{th}$ century. It also implies that the change in iWUE

is approximately equal to the relative change in atmospheric $CO_2$ (Eq. 15). This change is about 25% over the $20^{th}$ century and 43% since preindustrial. Recall, however, that isotopic signatures and the related variables ($c_i/c_a$, $\Delta_i$, iWUE, $A/g$) as considered here represent annual or multi-annual averages that have been weighted by C assimilation or allocation.

Discrimination $\Delta_i$, $c_i/c_a$, and iWUE ($A/g$) as well as their changes are closely related. These variables hold, in the framework applied in this study (e.g., Eqs. 4, 5, 13, 15), basically the same information. For iWUE this information is transformed

by the known atmospheric $CO_2$ evolution. As shown by Eq. 15, iWUE increases in proportion with $CO_2$ in the absence of a change in discrimination. A substantial increase in discrimination of more than 2 permil would be required to offset this change. It is therefore no surprise that Silva and Horwath (2013) find an increase in iWUE when using randomized $\delta^{13}C$ records in a Monte Carlo analysis instead of actual tree-ring data. Their finding does, however, not invalidate the usefulness

of iWUE as a physiological meaningful interpretation of $\delta^{13}C$ tree-ring records particularly as many studies have shown that non-randomized $\delta^{13}C$ measurements contain well-known environmental signals ranging from the Suess effect to inter-annual to centennial changes in climate.

We applied the tree-ring $\delta^{13}C$ data to test whether the current CLM4.5 and LPX-Bern implementation for $\delta^{13}C$, stomatal conductance, photosynthesis and related $CO_2$ fertilization, nitrogen limitation, and water-use efficiency mechanisms are con-

sistent with the tree-ring records. Simulated changes by LPX agree on average excellently with the tree-ring data (Fig. 7). LPX shows little change in the discrimination by C3 trees and the evolution of the average change in iWUE closely matches the tree-ring reconstruction. In contrast to the data, CLM4.5 results show a steadily decreasing trend in discrimination by C3 trees, roughly in parallel with rising $CO_2$ concentrations (Figs. 1, 7). The average decrease in discrimination by C3 trees for the grid cells with data amounts to about 1.5‰ by the end of the simulation and the increase in iWUE is about two times the change

indicated by the tree-ring data. The decrease in discrimination corresponds to a decrease in $c_i/c_a$ (Eq. 4). In other words, the leaf internal partial pressure $c_i$ is downregulated too strongly in CLM4.5 at least in the regions with tree-ring data.

Factorial simulations demonstrate that these overestimated trends are dominated by the response of CLM4.5 to increasing atmospheric $CO_2$, and not by the response to changes in climate. Factorial and sensitivity simulations also show that differences in trends between CLM4.5 and LPX are not related to trends in soil moisture nor to (small) differences in the applied climate

forcing data. The downregulation in $c_i/c_a$ in CLM4.5 therefore does not reflect a response to drought or worsening climatic conditions for $CO_2$ assimilation. These findings suggest that the main reason for the model-model and model-tree ring data difference in the trends of discrimination and iWUE is rooted in the parameterization of the photosynthesis-conductance coupling.

It is difficult to trace in complex models such as CLM4.5 origins of data-model mismatch. There are two aspects in the

current implementation of photosynthesis and conductance in CLM4.5 that may be problematic (Bonan et al., 2014; Ghimire et al., 2016). First, nitrogen downregulation of photosynthesis and conductance occurs on the sub-hourly model time step, although it is not plausible that leaf structures adjust so quickly. Second, the relationship between assimilation of carbon and transpiration of water (Ball et al., 1987) is prescribed with time-invariant and globally constant parameters.

Regarding nitrogen limitation, photosynthesis for isotopic discrimination is downregulated immediately, on the subhourly

time step of the model, by limited nitrogen availability in CLM4.5 (Eq. 6). This can lead to a depression of assimilation in the isotope calculation during times of high assimilation. As explained in the method section, photosynthesis of carbon and stomatal conductance for water is computed, unlike for discrimination, without nitrogen limitation. Instead, the allocation of gross primary productivity to carbon pools is downregulated under nitrogen limitation. This downregulation is on annual average generally less than 20% in forested regions and $20^{th}$ century changes in downregulation are generally small (within

$\pm 2\%$) in areas with tree ring data.

Raczka et al. (2016) applied a recalibrated version of CLM4.5 at a single site (Niwot Ridge, US) in simulations with and without nitrogen limitation. These authors show that nitrogen downregulation strongly affects the change in average $c_i/c_a$ over the $20^{th}$ century. At their site, changes in $c_i/c_a$ and in discrimination were positive over the industrial period and the century-scale change in discrimination was smaller when nitrogen limitation was not active. The observation-estimated seasonal cycle in discrimination at Niwot Ridge is better reproduced without nitrogen limitation than with limitation. Indeed, Raczka et al. (2016) suggest that downregulation of assimilation and $c_i/c_a$ by nitrogen limitation (Eq. 6) may be too strong in CLM4.5. Ghimire et al. (2016) implemented an alternative formulation for nitrogen limitation in CLM4.5, where nitrogen availability affects the maximum rate of photosynthesis (Vcmax) on slow time scales through changes in the leaf carbon-to-nitrogen ratio. This reduces the global bias in GPP, leaf area index, and biomass and improved water-use efficiency predictions compared to the original CLM4.5 formulation.

Another possibility for the model-data mismatch is that the photosynthesis formulation in CLM4.5 may not adequately represent the relationship between stomatal conductance and assimilation and related adjustment processes to long-term environmental changes. CLM4.5 employs the experimentally well-verified and widely-used Ball-Berry equation (Eq. 7) applying globally uniform, time invariant slope parameters, $m$, for C3 and C4 plants. Thus, any potential adjustment of $m$ to changes in environmental conditions, including the century-scale increase in atmospheric $CO_2$ or changes in water stress are not considered. It is currently unclear whether such adjustment processes occur and the current understanding of the underlying physiological mechanisms of stomatal responses is incomplete (e.g. Miner et al., 2017). The value of $m$ is set to nine in CLM4.5 for all C3 plants world-wide. This value is within the observational range, but species differences (e.g., evergreen versus deciduous) are neglected by using a single value. The review of Miner et al. (2017) yields mean values for $m$ of 9.8, 8.7, and 6.8 for angiosperm evergreen, angiosperm deciduous and for gymnosperm trees, respectively (their figure 1). Aranibar et al. (2006) inferred values of $m$ around 10 to 12 using observed foliar $^{13}$C values in pine.

Eqs 7 and 6, defining the photosynthesis-conductance coupling are evaluated on the sub-hourly time step of the model and the variables $A$, $R_h$, $CO_2$, and $f_{dreg}$ vary daily and seasonally, while we investigate here decadal-to-century-scale trends. This hampers a simple interpretation of the two equations, in particular in terms of expected temporal changes in iWUE and whether expected century-scale changes in discrimination may be positive or negative. As shown in Fig. 8, simulated changes in discrimination are positive in small areas of South America and Southeast Australia, and thus opposite to the world-wide trend simulated by CLM4.5.

Sato et al. (2015) investigated the influence of the use of vapor pressure deficit (VPD) versus relative humidity (RH) in Ball-Berry-type stomatal conductance formulations. About half of the investigated models apply RH as a driving variable (Eq. 7), despite the fact that VPD is considered the more relevant controlling factor. The global warming simulations reveal an increase in VPD and little change in RH. Their results suggest that the increase in VPD under global warming leads to a stronger downregulation of stomatal conductance ($g_s$) and $c_i/c_a$ for the VPD formulations compared to the RH formulation. This implies that replacing RH by VPD in the Ball-Berry equation in CLM4.5 may, without further adjustments, even increase the disagreement between modeled and reconstructed changes in discrimination and in iWUE.

Duarte et al. (2016) and Raczka et al. (2016) recalibrated CLM4.5 to match site-specific conditions in a conifer forest in the Northwestern US and at Niwot Ridge, Colorado. These studies find a substantial $20^{th}$-century increase in discrimination for their site-calibrated versions. One does expect from Eq. 7 that a change in the slope parameter $m$ will result in a proportional change in the absolute magnitude of iWUE and a corresponding change in discrimination. Duarte et al. (2016) reduced $m$ by a third in their single site version. In turn, simulated discrimination was altered by about 2.5‰ roughly corresponding to the expected 33% change in iWUE. In addition, the $20^{th}$ century change in isotopic discrimination is reduced. In brief, model structure in CLM4.5 permits both positive and negative changes in discrimination under rising $CO_2$. The slope parameter $m$, potentially affecting trends in discrimination, is set time invariant and globally uniform for all C3 plants. The relationship between $A$ and $g_s$ is assumed to be linear for constant $CO_2$ and relative humidity.

In LPX, the shape of the relationship between $g$ and $A$ is not as tightly prescribed as in CLM4.5. Rather, $A$ and $c_i/c_a$ are optimized for given environmental conditions as described in the method section. This optimization leads to small $20^{th}$-century changes in simulated isotopic discrimination in agreement with the observational evidence. The parameterization used in LPX is for the daily model time step and may not be readily transferred to models with a sub-hourly time step.

Alternative conductance-photosynthesis formulations, may be preferable, compared to the Ball-Berry relation as used in CLM4.5. The Ball-Berry relation is viewed as consistent with an optimization (Cowan and Farquhar, 1977) of stomatal conductance towards maintaining a constant water use efficiency by optimizing carbon gain per unit water loss. However, a number of alternative formulations are found in the literature. These differ in potentially important ways. For example, Medlyn et al. (2011) suggest that the slope parameter increases with increasing temperature; further the inverse of the square root of vapor pressure deficit is used instead of relative humidity in their relationship. Prentice et al. (2014) and Wang et al. (2016) rely on an optimality hypothesis considering the costs of maintaining both water flow and photosynthetic capacity. These authors use information on the spatial gradients in $\delta^{13}$C from stable isotope measurements on leaf material (Cornwell et al., 2016) to develop their photosynthesis-conductance relation. The slope parameter proposed by Prentice et al. (2014) depends on a number of variables, e.g., the temperature-dependent Michaelis-Menten coefficient for Rubisco-limited photosynthesis, viscosity of water, or the water potential difference between soil and leaf or foliage height. Hence, the slope parameter varies with environmental conditions in this approach. Bonan et al. (2014) implemented different photosynthesis-conductance modules within a CLM4.5 model version featuring a multi-layer canopy and compared results with leaf analyses and eddy covariance fluxes at six forest sites. The continuous soil-plant-atmosphere (SPA) module, optimizing carbon gain per unit water loss and considering hydraulic safety, performs similar or better than the Ball-Berry formulations used in CLM4.5. A better performance is particularly achieved under soil moisture stressed conditions.

It is a task for the future to explore whether the implementation of such optimizing modules will lead to a more realistic simulation of the spatio-temporal changes in isotopic discrimination. Taken together, agreement between CLM-modeled and observational trends in discrimination may be improved in the future by adjusting model parameters such as the slope parameter $m$, replacing formulations for nitrogen limitation, e.g. following Ghimire et al. (2016), and by implementing photosynthesis-conductance routines that adhere to an optimization principle as proposed in the literature (Bonan et al., 2014; Prentice et al., 2014).

## 4.4 Consistency with previous studies

The small changes in discrimination, an increase in iWUE proportional to the atmospheric $CO_2$ increase, and approximately constant $c_i/c_a$ over the $20^{th}$ century as reconstructed by our tree-ring compilation and simulated by LPX-Bern is consistent with most, but not all studies. As reported by Voelker et al. (2016), studies that are consistent with the notion of constant $c_i/c_a$ include early work by Wong et al. (1979) measuring A and g of leaves in a chamber, and a range of tree ring $\delta^{13}C$ studies (Saurer et al., 2004; Ward et al., 2005; Bonal et al., 2011; Franks et al., 2013). as well as a meta-analysis of FACE experiments (Ainsworth and Long, 2005). The European $\delta^{13}C$ tree-ring records analyzed by Frank et al. (2015) and Saurer et al. (2014) also point to a moderate control towards a constant $c_i/c_a$ ratio. Leonardi et al. (2012) conclude that the temporal variation in $\delta^{13}C$ in their long-term isotope tree-ring chronologies for 53 sites worldwide supports the hypothesis of an active plant mechanism that maintains a constant ratio between intercellular and ambient $CO_2$ concentrations. Lévesque et al. (2014) reported, from their $\delta^{13}C$ tree ring data, an increase in iWUE over the last 50 yr in the range of 8 to 29% for xeric and mesic sites in the Alps and Switzerland. At their sites, drought-induced stomatal closure has reduced transpiration at the cost of reduced carbon uptake and growth. Churakova (Sidorova) et al. (2016a) report different iWUE strategies with almost constant $c_i/c_a$ for European larch since the 1990s and continuously increasing iWUE for mountain pine trees since the 1980s in the Swiss National Park. Liu et al. (2014a) find moderate changes in $c_i/c_a$ and an increase in iWUE by 36% in a riparian forest in Northwestern China from 1920 to 2012. Similarly, Peñuelas et al. (2011), analyzing changes in tree ring $\delta^{13}C$ and growth at 47 sites worldwide, inferred little change in discrimination, $c_i/c_a$, and an increase in iWUE of 20.5% from 1960 to 2000. 18 of the 35 studies with growth data show an increase in growth with time, while the others show no or negative growth changes. Voelker et al. (2016) analyzed studies of $\delta^{13}C$ and photosynthetic discrimination in woody angiosperms and gymnosperms that grew across a range of $CO_2$ spanning at least 100 ppm and combining paleo data, tree-ring records, and FACE-type experiments. They conclude that woody plants respond to increasing $CO_2$ by regulating leaf gas-exchange along a continuum of $c_a - c_i$ and $c_i/c_a$ that minimizes water loss for a given amount of C gain and therefore increasingly minimizes the likelihood of exposure to drought stress. Summarizing five years of results from the Basel FACE experiment, Klein et al. (2016) find that iWUE increased in their experiment by 38% at the needle level, as a result of higher assimilation at constant conductance. Interestingly, Klein et al. (2016) could not identify an increase in plant carbon stocks corresponding to the increase in assimilation and the fate of the additionally assimilated carbon remains unclear. The 38% increase in iWUE is comparable to the increase from ambient (400 ppm) to elevated (550 ppm) and in line with observations reported by De Kauwe et al. (2013) as well as a meta-analysis of FACE experiments (Ainsworth and Long, 2005). Streit et al. (2014) measured needle gas exchange and analyzed $\delta^{13}C$ of needles and tree rings of $Larix\ decidua$ and $Pinus\ mugo$ after nine years of free air $CO_2$ enrichment. Both species showed an increase in net photosynthesis and, in agreement with Keel et al. (2006), small or no changes in conductance under elevated $CO_2$ and a change in iWUE roughly in proportion to the increase in $CO_2$. Elevated $CO_2$ induced increased basal area growth in $L.\ decidua$, but not in $P.\ mugo$. Neither nitrogen limitation, nor end-product limitation, and, in agreement with other FACE studies (Bader et al., 2010; Liberloo et al., 2007), no downregulation of maximal photosynthetic rate was found.

In contrast, Battipaglia et al. (2013) report an increase in iWUE between 50 and 90% for the ORNL, DUKE, and POP-EUROFACE Face sites and that $c_i$ was likely maintained constant and $^{13}$C discrimination reduced under elevated $CO_2$. We note that this finding, based on $^{13}$C tree ring isotope measurements, differs remarkably from results by De Kauwe et al. (2013). De Kauwe et al. (2013) suggest a proportional increase in iWUE at the Duke and ORNL FACE sites based on in-situ foliage

gas exchange measurements (see their Fig. 6). This difference might be caused by the comparison of leaf-level gas-exchange and stem-level data. Integration times may differ between the two data streams and stem level data are influenced by growth rate and storage compounds. A scenario of constant $c_i$ under increasing $CO_2$ concentrations is also suggested by Keenan et al. (2013) who report a strong increase in water-use efficiency summarizing results from eddy-covariance measurements at about 20 temperate and boreal forest sites in the Northern Hemisphere. However, these studies may be affected by local

conditions not representative for larger scales and be influenced by temporal sampling biases as the record length of FACE and eddy-covariance measurements is limited. Indeed, the individual tree-ring records in Figure 7 show considerable site-to-site variability as well as decadal variability. In addition, factors other than $c_i/c_a$, e.g. vapor pressure deficit, may influence trends at eddy covariance sites (Knauer et al., 2017). The suggestion of a constant $c_i$ under increasing $CO_2$ is challenged by Knauer et al. (2017). These authors find that the ecosystem trends reported by Keenan et al. (2013) and a scenario of a constant

$c_i$, as also suggested by Battipaglia et al. (2013), are in conflict with observed large-scale trends in continental discharge, evapotranspiration, and the seasonal $CO_2$ exchange. Rather, the comparison of observational data and model outcome by Knauer et al. (2017) support the finding of a physiological regulation towards a constant $c_i/c_a$ under rising atmospheric $CO_2$.

## 5    Conclusions

We compiled $20^{th}$ century $\delta^{13}$C tree ring records for seventy-three sites (Table A1). The records are used to reconstruct changes

in the ratio of $CO_2$ concentration within the stomatal cavity of C3 trees and the atmosphere, $c_i/c_a$, and to reconstruct changes in the intrinsic water use efficiency, iWUE, denoting the ratio of photosynthesis to conductance (A/g). The tree-ring results suggest, on average, constant $c_i/c_a$ over the $20^{th}$ century and iWUE to have increased in proportion with atmospheric $CO_2$. The simulated changes in discrimination, $c_i/c_a$, and iWUE are consistent with the reconstructions for LPX-Bern 1.3, whereas CLM4.5 shows larger trends and overestimates $20th$ century changes in iWUE by almost a factor of two (Fig. 7). This suggest

that it is desirable to adjust the implementation of photosynthesis and conductance in CLM4.5 towards a better agreement with observation-derived century-scale trends in $^{13}$C discrimination and intrinsic water-use efficiency. In conclusion, this study demonstrates that existing information on the magnitude and trends of stable carbon isotopes permits the evaluation of global land carbon cycle models. In particular, $\delta^{13}$C data provide insight on the fundamental relationship between carbon assimilation and stomatal conductance controlling the flow of $CO_2$ and water. Tree ring records are useful as they cover retrospectively

decadal-to-century time scales not accessible by laboratory or field experiments and the relevant instrumental records, but a time scale that is directly relevant for the response of plants to the decadal-to-century-scale rise in $CO_2$ and to global warming.

*Acknowledgements.* We thank B. Otto-Bliesner and E. Brady for supporting this and the work of the CESM isotope group and for hosting FJ during visits at NCAR. We thank A. Ballantyne and an anonymous reviewer for constructive comments. The research leading to these results was supported through Swiss National Science Foundation (SNF) projects 200020_147174, 20020_159563, and CRSII3_136295 (Sinergia Project iTREE). OVC acknowledges support by Marie Curie IIF (EU-ISOTREC 235122), the Marie Heim-Vögtlin Program (MHV PMPDP2_145507), and Era.Net RUSplus project ELVECS (SNF project number: IZRPZ0_164735). WJR and CDK were supported by the Director, Office of Science, Office of Biological and Environmental Research of the US Department of Energy under Contract No. DE-AC02-05CH11231 as part of their Regional and Global Climate Modeling program. Simulations with CLM4.5 were carried out at the Swiss National Supercomputing Centre in Lugano, Switzerland.

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

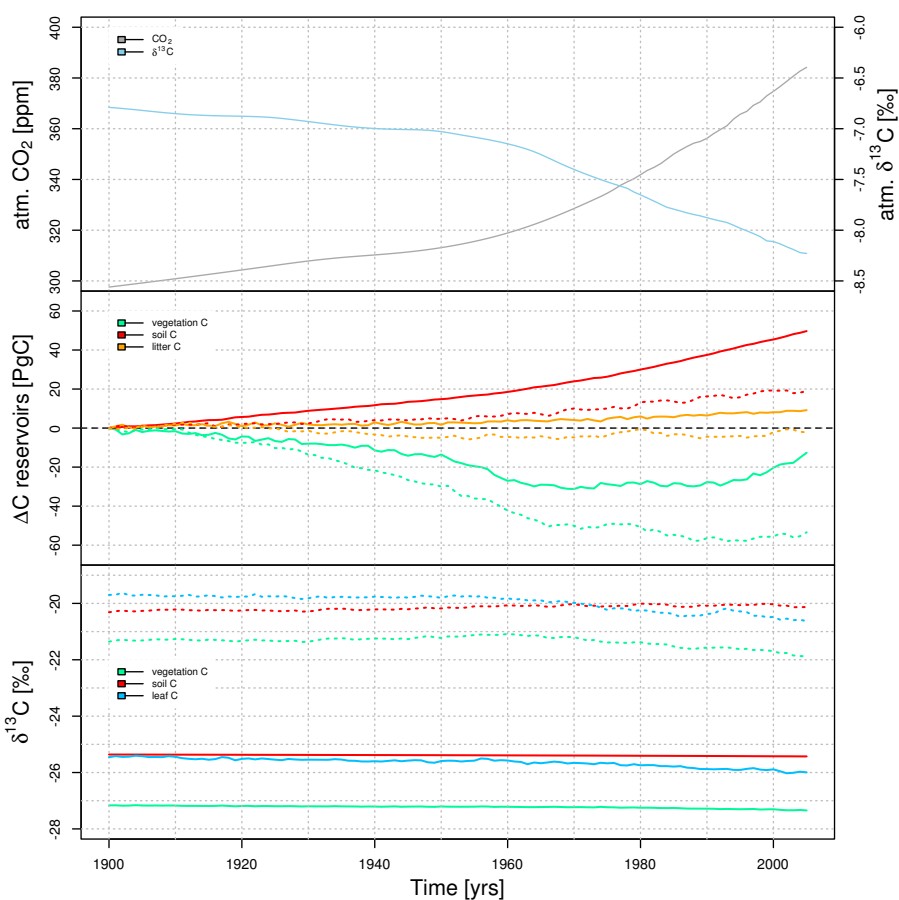

**Figure 1.** a) Prescribed evolution of atmospheric $CO_2$ and its $\delta^{13}C$ signature, b) simulated changes in the global inventory of carbon stored in vegetation (green), litter (orange), soils (red), c) Evolution of the global average (mass-weighted) $\delta^{13}C$ signature of carbon in leaves, vegetation, and soils (CLM4.5: solid, LPX: dotted).

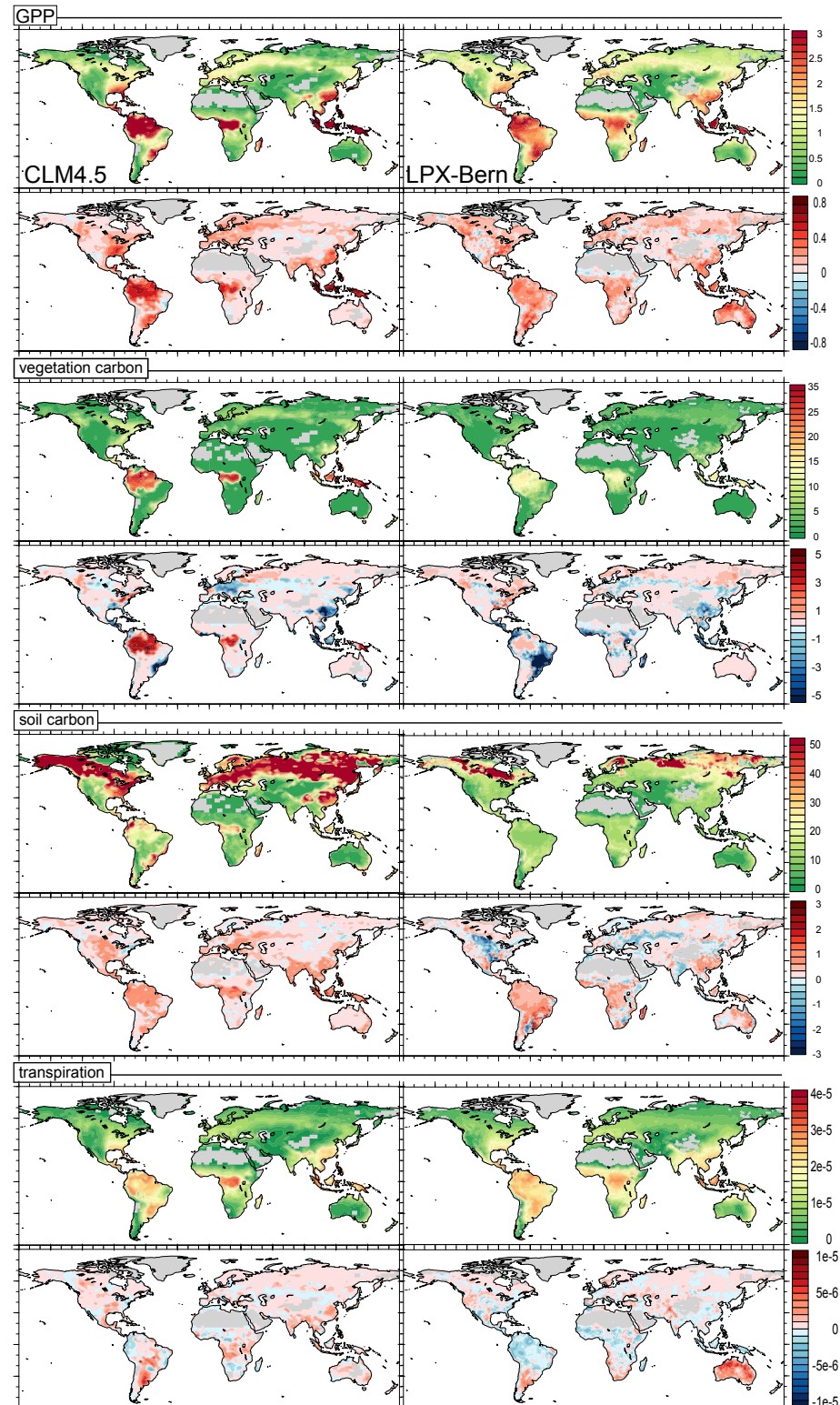

**Figure 2.** GPP (kgC/m$^2$/yr), vegetation carbon (kgC/m$^2$), soil carbon (kgC/m$^2$) and transpiration (mm/s) in the year 2000 (top) and the difference between 1900 and 2000 (below). The estimates are based on decadal means (1996-2005 and 1996-2005 minus 1896-1905, respectively).

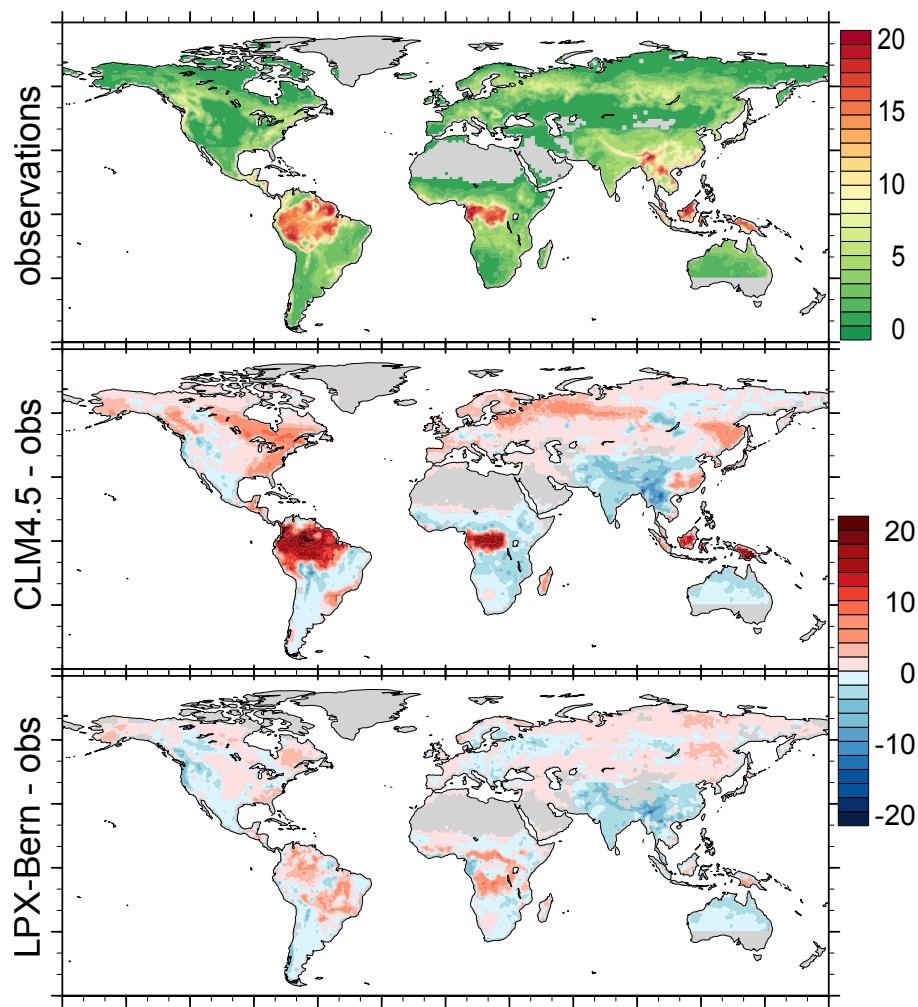

**Figure 3.** Estimated modern distribution of vegetation carbon derived from observations (Carvalhais et al., 2014) (top) and the differences of model minus observation-derived distributions for CLM4.5 (middle) and LPX (bottom). All results are in kgC/m$^2$ and averages for the decade 1996 to 2005 are used for the model distributions.

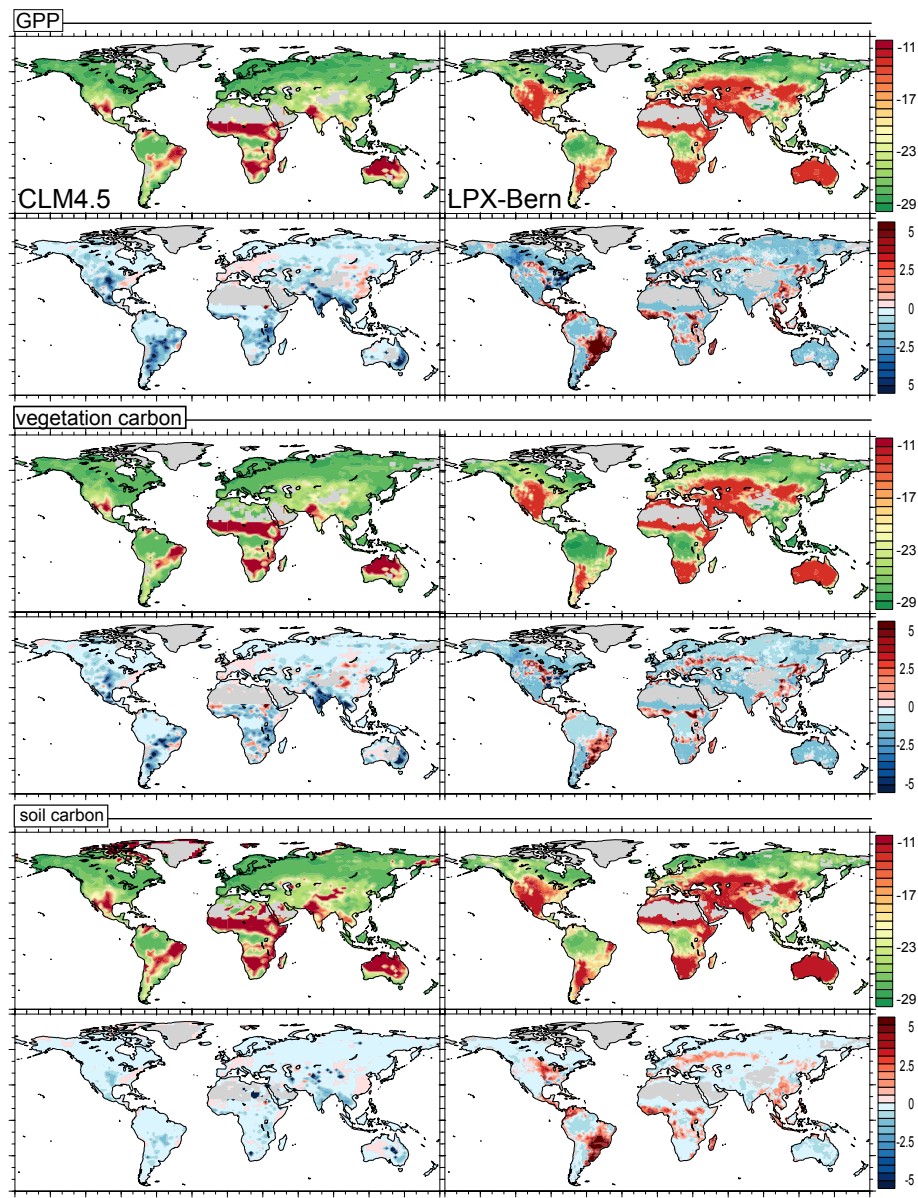

**Figure 4.** $\delta^{13}$C (‰) of GPP, vegetation carbon and soil carbon in the year 2000 (top) and the change from 1900 to 2000 (below). The estimates are based on decadal means (1896-1905 and 1996-2005, respectively).

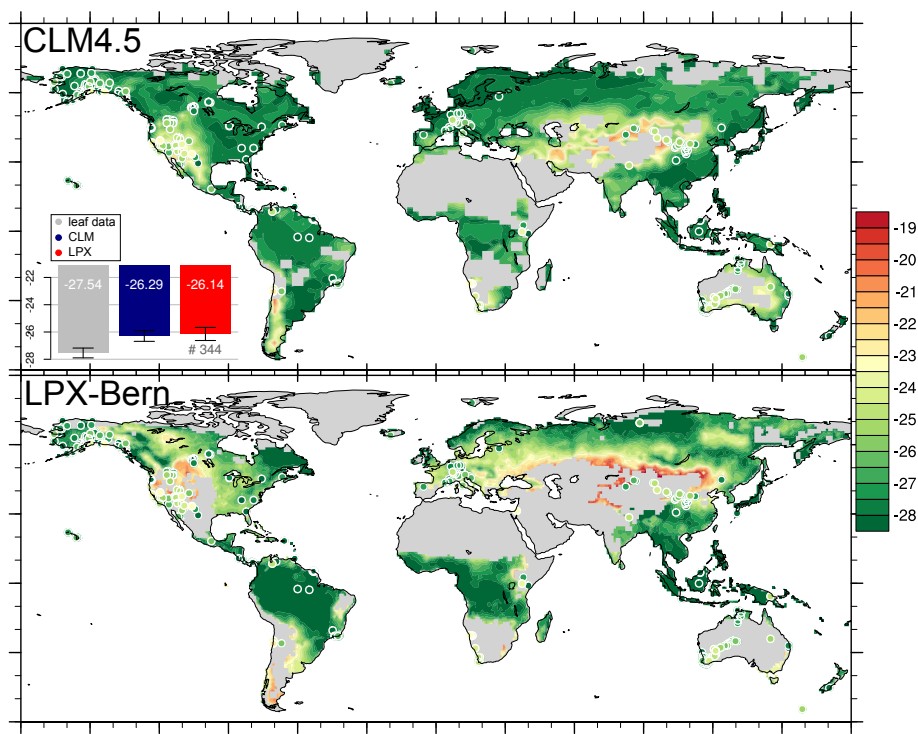

**Figure 5.** $\delta^{13}$C as measured on leaves of C3 trees (colored circles; Cornwell et al., 2016) and as simulated by a) CLM4.5 and b) LPX (shading) for the decade 1996 to 2005. Samples for the measurements were taken between 1975 and 2014. The bar plot shows the average over all measurements (gray) and the average signature of C3 trees of the corresponding grid cells for CLM4.5 (blue) and LPX (red). The number below the red bar indicates the number of available measurements.

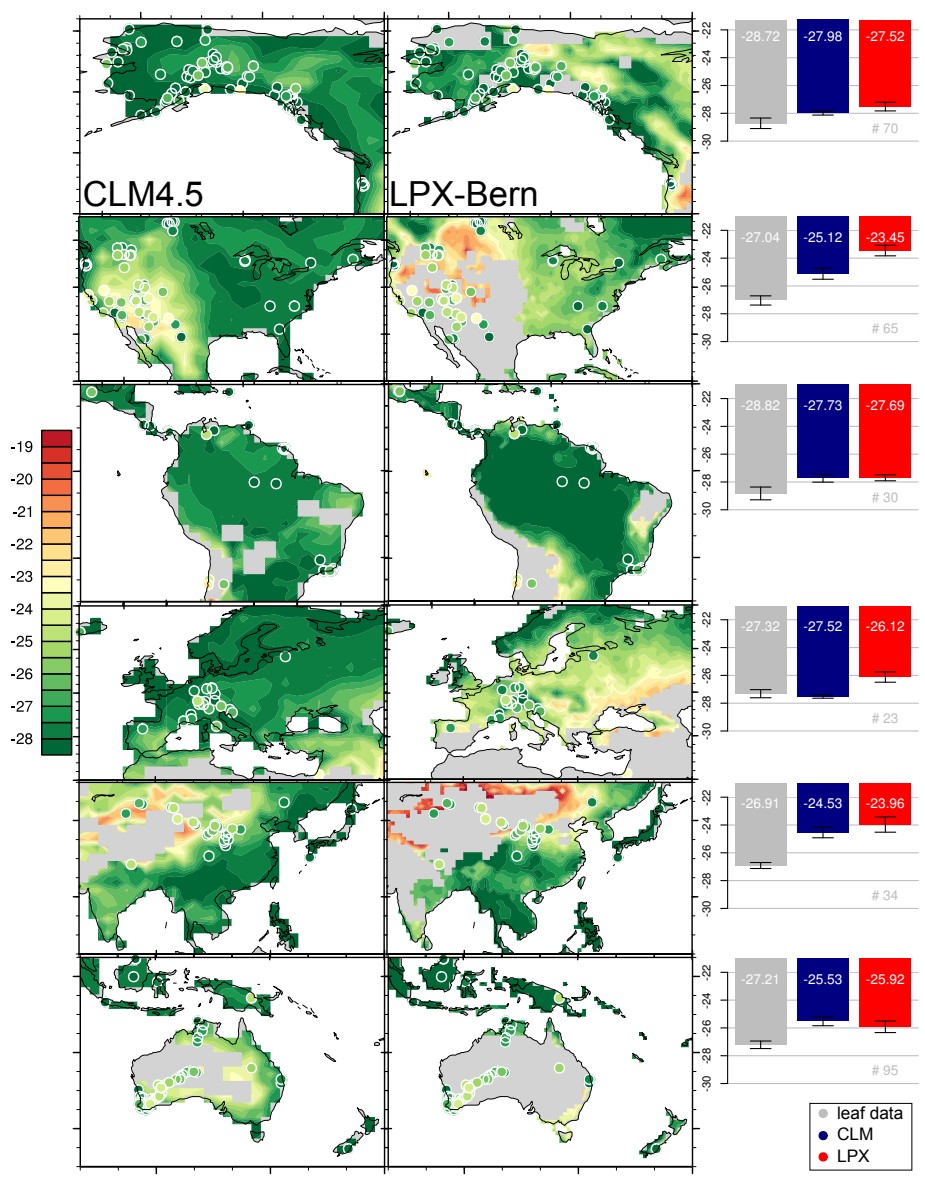

**Figure 6.** Same as Fig. 5 for selected regions.

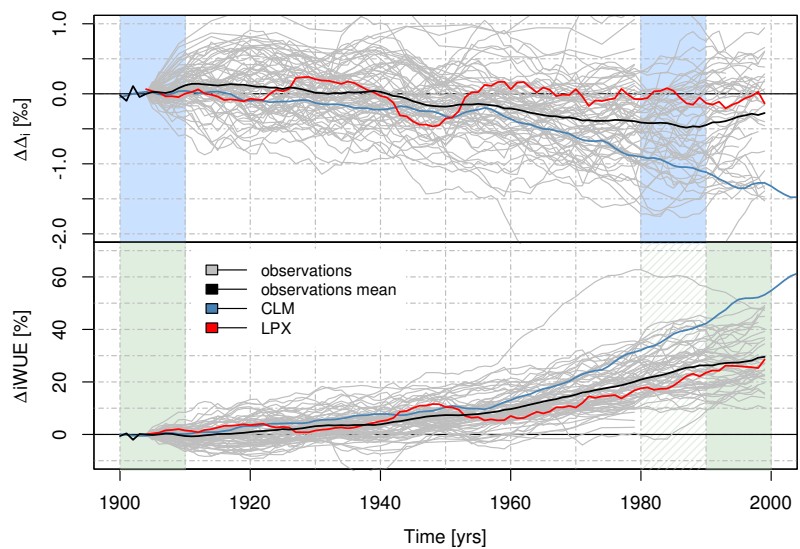

**Figure 7.** $20^{th}$-century changes in the discrimination $\Delta_i$ of C3 trees (($\delta^{13}c_a - \delta^{13}\mathrm{C}_{plant}/1 + \frac{\delta^{13}\mathrm{C}_{plant}}{1000}$)) and iWUE (%) as measured on tree rings (black) and modeled by CLM4.5 (blue) and LPX (red). Observations are represented by the average over all measurements (see Tab. A1), models by the average of the mean grid-cell signature of C3 trees and for all model grid cells where measurements are available. Values are given w.r.t. the decade 1900, the background colors indicate the respective decades investigated for changes in discrimination (Figs. 8, 9, 10) and iWUE (Figs. 10, 11, 12.) Absolute values of iWUE for the decade 1900-1909 are 78 for the observation and LPX-Bern and 50 for CLM4.5. Standard deviations calculated over all observations of century-scale changes (1980s minus 1900s) are 0.72 for $\Delta_i$ and 9.76 for iWUE, respectively.

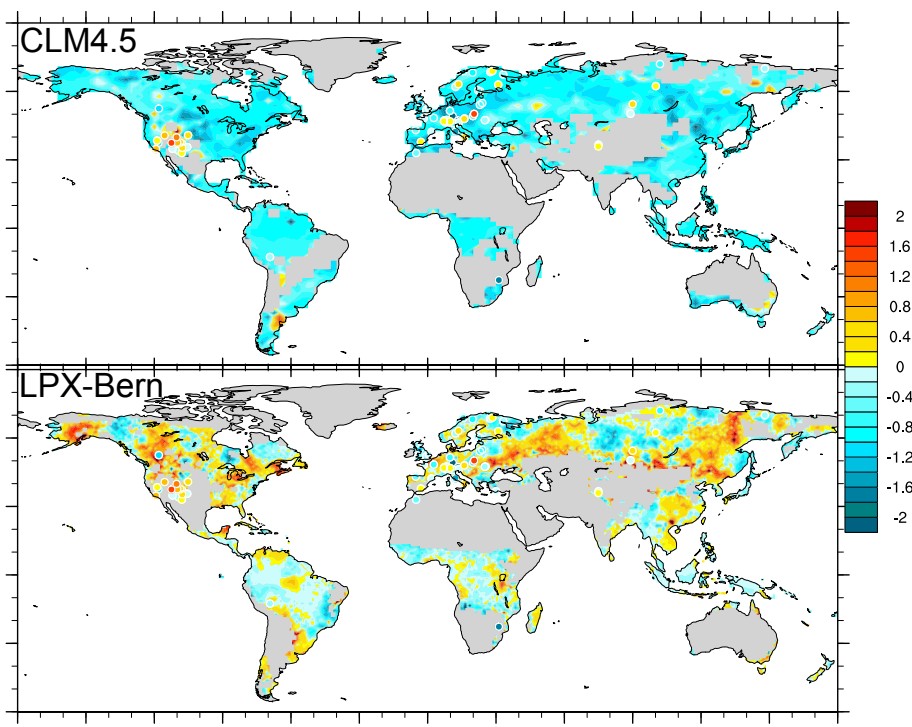

**Figure 8.** Century-scale change in the discrimination $\Delta_i$ of C3 trees. The discrimination is calculated from $\delta^{13}$C tree-ring data (colored circles) and (a) CLM4.5 and (b) LPX model results using equation 5. The changes are based on decadal means (1980s minus 1900s).

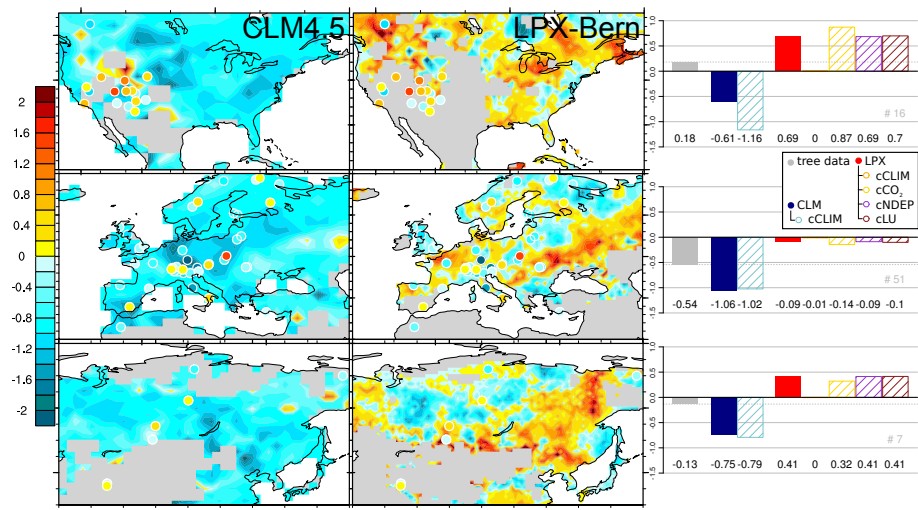

**Figure 9.** Same as Fig. 8 but for three selected regions. The bar plots show the regional average change in discrimination of C3 trees as calculated from tree-ring $\delta^{13}$C data (gray) and from results of the standard simulations of CLM4.5 (filled blue) and LPX (filled red) and from factorial runs (pattern). Individual driving factors (climate, $CO_2$, N-deposition, and land use) were kept constant in the factorial runs as explained in the main text and indicated by the legend. Model averages are computed from the mean grid-cell discrimination of C3 trees and for all model grid cells where tree-ring estimates are available. The number of available tree-ring records is indicated.

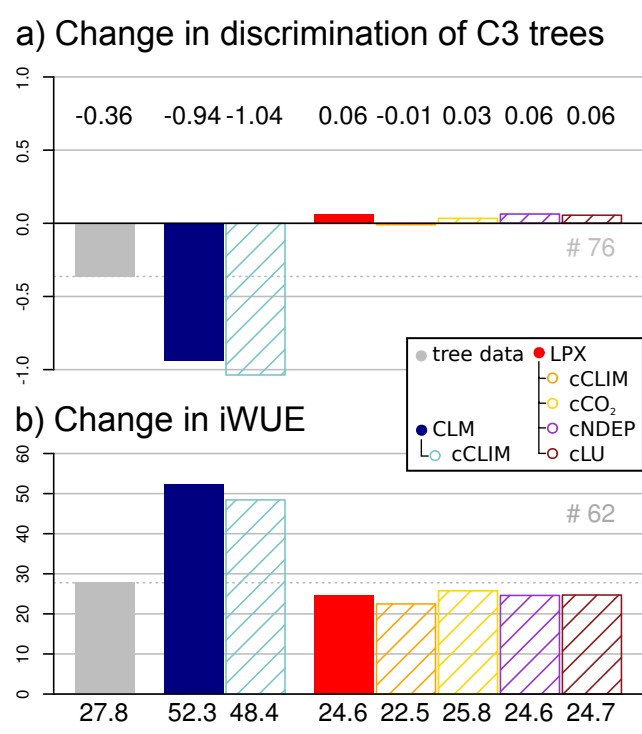

**Figure 10.** Century-scale changes in a) discrimination $\Delta_i$ and b) iWUE (%) of C3 trees as calculated from tree-ring $\delta^{13}$C data (gray) and from results of the standard simulations of CLM4.5 (filled blue) and LPX (filled red) and from factorial runs (pattern). Individual driving factors (climate, $CO_2$, N-deposition, and land use) were kept constant in the factorial runs as explained in the main text and indicated by the legend. The number of available records is indicated in each subpanel. All estimates are base on decadal means (a): 1980s minus 1900s; b) 1990s minus 1900s.

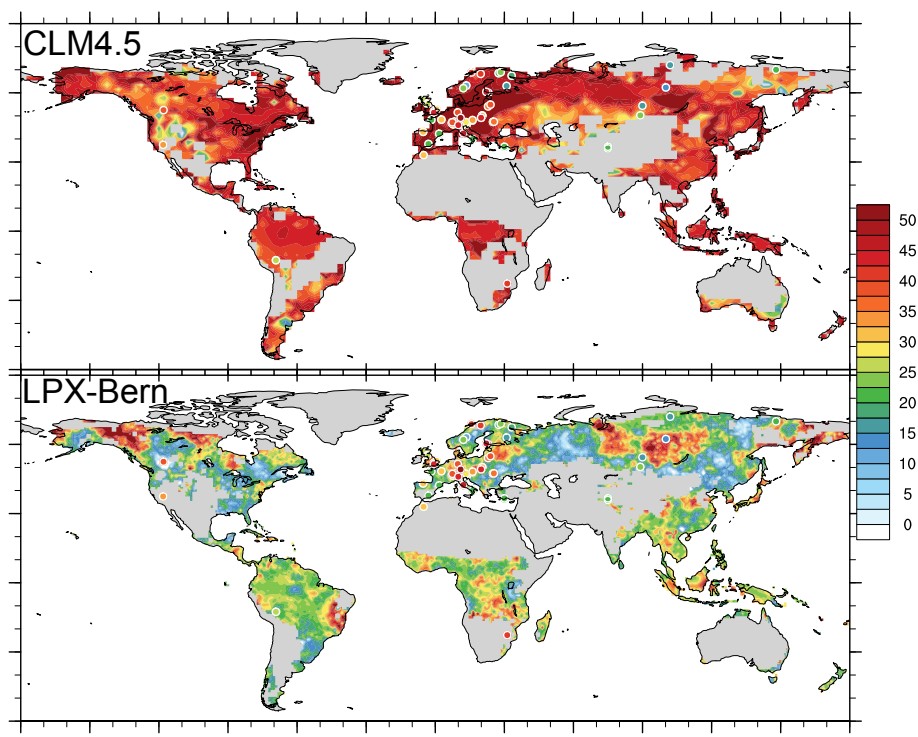

**Figure 11.** Century-scale changes in iWUE (%) of C3 trees as calculated from $\delta^{13}$C tree-ring data (colored circles) and (a) CLM4.5 and (b) LPX model results. The changes are based on decadal means (1990s minus 1900s).

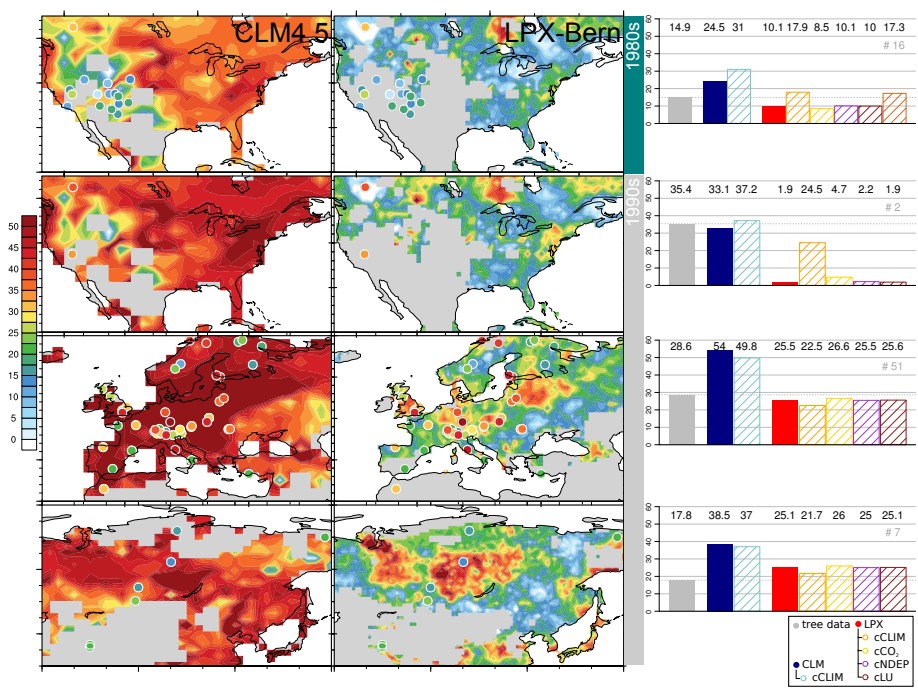

**Figure 12.** Same as Fig. 11 but for three selected regions. The bar plots show the regional average change in iWUE (%) over all tree-ring measurements (gray) and the average for CLM4.5 (blue) and LPX (red) of the mean grid-cell iWUE of C3 trees and for all model grid cells where tree-ring estimates are available. Individual driving factors (climate, $CO_2$, N-deposition, and land use) were kept constant in the factorial runs (patterns) as explained in the main text and indicated by the legend in the bar plots. The number of available records is indicated in each subpanel. All estimates are based on decadal means (a): 1980s minus 1900s; (b), (c), (d): 1990s minus 1900s.

**Table 1.** Parameter values of LPX-Bern that were modified from the previous model version (LPX v1.2) and as used in this study (LPX v1.3). The new values were determined by maximizing agreement with a set of observational data.

| Parameter | | LPX v1.2 | LPX v1.3 |
|---|---|---|---|
| $\alpha_a$ | Fraction of PAR assimilated at ecosystem level relative to leaf level | 0.5 | 0.620 |
| $\alpha_{C3}$ | Intrinsic quantum efficiency of $CO_2$ uptake in C3 plants | 0.07 | 0.0857 |
| $\theta$ | Co-limitation shape parameter | 0.7 | 0.694 |
| $g_m$ | Canopy conductance scaling parameter for water demand calculation | 3.24 | 2.95 |
| $\alpha_m$ | Priestley-Taylor coefficient in water demand calculation | 1.4 | 1.764 |
| $\tau_{sapwood}$ | Sapwood to heartwood turnover [yr] | 20 | 11.1 |
| $k_{la:sa}$ | Allometric scaling parameter: leaf area to sapwood area | 1.0 | 1.3 |
| $k_{mort}$ | Coefficient of growth efficiency in mortality equation | 0.01 | 0.0118 |
| $E_{0,hr}$ | Temperature sensitivity of heterotrophic respiration [K] | 308.56 | 222.0 |
| $f_{atm}$ | Fraction of litter entering atmosphere directly | 0.6 | 0.66 |
| $f_{slow}$ | Fraction of litter entering slow soil pool | 0.015 | 0.0106 |
| $k_{soil,tune}$ | Tuning factor for soil decay | 0.7 | 0.840 |
| $\text{nitr}_{max}$ | Maximum nitrification rate | 0.1 | 0.0923 |
| $\text{f}_{imob,soil}$ | Nitrogen immobilization in soil | 0.0 | 0.249 |
| $\text{ox}_{past}$ | Fraction of direct oxidation of leaf turnover on pastures | 0.4 | 0.298 |
| $\text{ox}_{crop}$ | Fraction of direct oxidation of leaf turnover on cropland | 0.9 | 0.9094 |

**Table 2.** Integrals and mean isotopic signatures of global carbon pools: steady state in the year 2000 (mean 1996-2005) and change over the $20^{th}$ century for both models and an observational dataset (Carvalhais et al., 2014).

| C reservoir [GtC] | CLM | LPX | Observations |
|---|---|---|---|
| *in the year 2000* | | | |
| GPP | 168 | 149 | - |
| vegetation | 874 | 425 | 442 |
| soil | 3916 | 1710 | 2397 |
| litter | 82 | 260 | - |
| leaf | 21 | 24 | - |
| **total C** | 5099 | 2405 | - |
| *change over $20^{th}$ century* | | | |
| GPP | 20 | 17 | - |
| vegetation | -21 | -55 | - |
| soil | 45 | 19 | - |
| litter | 8 | -3 | - |
| leaf | 2 | 5 | - |
| **total C** | 32 | -35 | - |

| C reservoir [$\delta^{13}$C; ‰] | CLM | LPX | Observations |
|---|---|---|---|
| *in the year 2000* | | | |
| GPP | -24.9 | -20.7 | - |
| total C | -27.6 | -23 | - |
| total C [GtC ‰] | -140660 | -55227 | - |
| *change over $20^{th}$ century* | | | |
| GPP | -0.74 | -0.45 | - |
| total C | -0.00 | 0.21 | - |
| total C [GtC ‰] | -904 | 1329 | - |

Table A1: $\delta^{13}$C tree-ring time series compiled for this study.

| # | Site | Country | Period | Lat | Lon | Reference |
|---|------|---------|--------|-----|-----|-----------|
| 1 | Dachstein | Austria | 1900-1996 | 47.28 | 13.36 | Saurer et al. (2014) |
| 2 | Lainzer Tiergarten | Austria | 1900-2003 | 48.18 | 16.20 | Treydte et al. (2007) |
| 3 | Poellau | Austria | 1900-2002 | 47.95 | 16.06 | Treydte et al. (2007) |
| 4 | Columbia | Canada | 1900-1990 | 52.50 | -118.00 | Edwards et al. (2008) |
| 5 | Bromarv | Finland | 1901-2002 | 60.00 | 23.08 | Treydte et al. (2007) |
| 6 | Ilomantsi | Finland | 1900-2002 | 62.98 | 30.98 | Treydte et al. (2007) |
| 7 | Kessi/Inari | Finland | 1900-2002 | 68.93 | 28.42 | Treydte et al. (2007) |
| 8 | Laanila | Finland | 1900-2002 | 68.50 | 27.50 | Gagen et al. (2011) |
| 9 | Turku | Finland | 1900-1994 | 60.41 | 22.17 | Robertson et al. (1997) |
| 10 | Fontainebleau | France | 1900-2000 | 48.38 | 2.67 | Treydte et al. (2007) |
| 11 | Rennes | France | 1900-1999 | 48.25 | -1.70 | Treydte et al. (2007) |
| 12 | Dransfeld | Germany | 1900-2002 | 51.50 | 9.78 | Treydte et al. (2007) |
| 13 | Franconia | Germany | 1900-2005 | 49.15 | 11.00 | Saurer et al. (2014) |
| 14 | Monte Cimino | Italy | 1900-2005 | 42.41 | 12.20 | Klesse et al. in prep |
| 15 | Serra di Crispo | Italy | 1900-2003 | 39.93 | 16.20 | Treydte et al. (2007) |
| 16 | Trento | Italy | 1900-2004 | 45.98 | 11.66 | Saurer et al. (2014) |
| 17 | Panemunes Silas | Lithuiana | 1900-2002 | 54.88 | 23.97 | Treydte et al. (2007) |
| 18 | Col Du Zad | Morocco | 1900-2000 | 32.97 | -5.07 | Treydte et al. (2007) |
| 19 | Forfjorddalen | Norway | 1900-2001 | 68.79 | 15.72 | Young et al. (2012) |
| 20 | Gutuli | Norway | 1900-2003 | 62.00 | 12.18 | Treydte et al. (2007) |
| 21 | Bagrot | Pakistan | 1900-1998 | 35.90 | 74.93 | Treydte et al. (2006) |
| 22 | Boibar | Pakistan | 1900-1998 | 36.62 | 74.98 | Treydte et al. (2006) |
| 23 | Ballantyne | Peru | 1900-2004 | -12.60 | -69.20 | Ballantyne et al. (2011) |
| 24 | Niopolomice Gibiel | Poland | 1900-2003 | 50.12 | 20.38 | Treydte et al. (2007) |
| 25 | Niopolomice Gibiel | Poland | 1900-2003 | 50.12 | 20.38 | Treydte et al. (2007) |
| 26 | Suwalki | Poland | 1900-2003 | 54.10 | 22.93 | Treydte et al. (2007) |
| 27 | Cucuraena | Romania | 1900-2005 | 47.40 | 25.08 | Klesse et al. in prep |
| 28 | Giumalau | Romania | 1900-2005 | 47.45 | 25.45 | Klesse et al. in prep |
| 29 | Altai | Russia | 1900-2005 | 50.23 | 89.04 | Sidorova et al. (2012) |
| 30 | Indigirka | Russia | 1900-2004 | 70.00 | 148.00 | Sidorova et al. (2008) |
| 31 | Khakasia | Russia | 1900-2005 | 54.41 | 89.96 | Knorre et al. (2010) |
| 32 | Khibiny | Russia | 1900-2005 | 67.41 | 33.15 | Saurer et al. (2014) |
| 33 | Taimyr | Russia | 1900-2005 | 72.00 | 102.00 | Sidorova et al. (2010) |
| 34 | Tura | Russia | 1900-2005 | 62.32 | 100.14 | Sidorova et al. (2009) |
| 35 | Hlinna Dolina1 | Slovakia | 1900-2005 | 49.19 | 19.90 | Weigt et al. in prep |
| 36 | Hlinna Dolina2 | Slovakia | 1900-2005 | 49.19 | 19.90 | Weigt et al. in prep |
| 37 | Veza | Slovenia | 1907-2005 | 46.37 | 13.69 | Saurer et al. (2014) |
| 38 | Pafuri | South Africa | 1900-2005 | -22.70 | 31.25 | Woodborne et al. (2015) |
| 39 | Cazorla | Spain | 1900-2005 | 37.81 | -2.96 | Treydte et al. (2007) |
| 40 | Pinar de Lillo | Spain | 1900-2002 | 43.07 | -5.25 | Treydte et al. (2007) |

| #  | Site               | Country     | Period    | Lat   | Lon     | Reference                          |
|----|--------------------|-------------|-----------|-------|---------|------------------------------------|
| 41 | Pedraforca         | Spain       | 1900-2005 | 42.24 | 1.70    | Treydte et al. (2007)              |
| 42 | Furuberget         | Sweden      | 1900-2005 | 63.16 | 13.50   | Saurer et al. (2014)               |
| 43 | Torneträsk         | Sweden      | 1900-2005 | 68.22 | 19.72   | Loader et al. (2013)               |
| 44 | Bettlachstock      | Switzerland | 1900-1995 | 47.22 | 7.42    | Saurer et al. (2000, 2012)         |
| 45 | Cavergno           | Switzerland | 1900-2003 | 46.35 | 8.60    | Treydte et al. (2007)              |
| 46 | Davos north        | Switzerland | 1900-2005 | 46.82 | 9.86    | Klesse et al. in prep              |
| 47 | Laegern B          | Switzerland | 1900-2005 | 47.48 | 8.36    | Klesse et al. in prep              |
| 48 | Loetschental       | Switzerland | 1900-2004 | 46.43 | 7.80    | Saurer et al. (2014)               |
| 49 | Loetschental N19   | Switzerland | 1900-2005 | 46.39 | 7.77    | Klesse et al. in prep              |
| 50 | Salvenach          | Switzerland | 1900-2005 | 46.91 | 7.15    | Kimak and Leuenberger (2015)       |
| 51 | Swiss National Park | Switzerland | 1900-2005 | 46.00 | 10.00   | Churakova (Sidorova) et al. (2016a) |
| 52 | Vigera             | Switzerland | 1900-2003 | 46.50 | 8.77    | Treydte et al. (2007)              |
| 53 | Elmali             | Turkey      | 1900-2005 | 36.60 | 30.02   | Saurer et al. (2014)               |
| 54 | Lochwood           | UK          | 1900-2003 | 55.27 | -3.43   | Saurer et al. (2014)               |
| 55 | Sandringham        | UK          | 1900-1994 | 52.83 | 0.50    | Saurer et al. (2014)               |
| 56 | Southern Upl.      | UK          | 1907-2005 | 57.10 | -5.43   | Saurer et al. (2014)               |
| 57 | Windsor            | UK          | 1900-2003 | 51.41 | -0.59   | Treydte et al. (2007)              |
| 58 | Woburn             | UK          | 1900-2003 | 51.98 | -0.59   | Treydte et al. (2007)              |
| 59 | Alton              | USA         | 1900-1984 | 37.44 | -112.49 | Leavitt et al. (2007)              |
| 60 | Aztec              | USA         | 1900-1984 | 37.00 | -107.82 | Leavitt et al. (2007)              |
| 61 | Blanco             | USA         | 1900-2005 | 37.45 | -118.17 | Bale et al. (2011)                 |
| 62 | Cerro Colorado     | USA         | 1900-1984 | 35.28 | -107.72 | Leavitt et al. (2007)              |
| 63 | Dry Canyon         | USA         | 1900-1984 | 37.58 | -108.55 | Leavitt et al. (2007)              |
| 64 | Gate Canyon        | USA         | 1900-1984 | 39.88 | -110.23 | Leavitt et al. (2007)              |
| 65 | Hawthorne          | USA         | 1900-1984 | 38.43 | -118.75 | Leavitt et al. (2007)              |
| 66 | Kane Springs       | USA         | 1900-1984 | 37.52 | -109.90 | Leavitt et al. (2007)              |
| 67 | Lamoille           | USA         | 1900-1984 | 40.69 | -115.47 | Leavitt et al. (2007)              |
| 68 | Lower Colonias     | USA         | 1900-1984 | 35.56 | -105.55 | Leavitt et al. (2007)              |
| 69 | Mimbres            | USA         | 1900-1984 | 33.00 | -107.93 | Leavitt et al. (2007)              |
| 70 | NE AZ              | USA         | 1900-1984 | 34.08 | -109.35 | Leavitt et al. (2007)              |
| 71 | NC AZ              | USA         | 1900-1984 | 34.83 | -111.98 | Leavitt et al. (2007)              |
| 72 | Owl Canyon         | USA         | 1900-1984 | 40.79 | -105.18 | Leavitt et al. (2007)              |
| 73 | Ozena              | USA         | 1900-1984 | 34.72 | -119.24 | Leavitt et al. (2007)              |