# Peer review of "20th-century changes in carbon isotopes and water-use efficiency: Tree-ring based evaluation of the CLM4.5 and LPX-Bern models"

_Biogeosciences, 2016_

## Referee Comment (RC1) · Anonymous Referee #1 · 23 Dec 2016

In the manuscript "20th – century changes in carbon isotopes and water-use efficiency: Tree-ring based evaluation of the CLM4.5 and LPX-Bern models" Keller et al. present the implementation of a carbon isotope scheme in two global models as well as their performance with respect to simulated spatial patterns and decadal trends. The model results are compared to two different datasets, tree-ring records and bulk leaf delta 13C data. This study is a valuable contribution to ongoing efforts on the implementation of carbon isotopes in global vegetation models. The overall approach as well as the results are presented in an adequate and clear manner. My main criticism is on the conclusions that are drawn from these results. Some parts of the discussion will be subject to revision. More detailed comments are listed below.

Abstract The abstract is a nice summary and contains all important aspects of the paper. However, I disagree with the last sentence. Suggesting "fundamental problems associated with the prescribed relationship between conductance and assimilation" is rather provocative and not supported by the results of this study. This relationship is strongly supported by observations (see e.g. Wong 1979 and also De Kauwe et al. 2013, Global Change Biology; papers cited in this study) and consequently used in most global models. I.e. it would indeed be a fundamental problem in our understanding of plant physiology. If this thought is brought up at such a prominent position in the paper, it needs to be better discussed and corroborated later in the manuscript, see below.

Introduction The introduction starts with a nice overview on the application of isotopes in the Earth System, which is a good motivation for this study. Following this part, the connection between carbon isotopes and plant physiological behavior is pointed out. The introduction closes with a very clear outline of the goals of this study.

Methods Page 4: the listing of all PFTs seems a bit unnecessary to me. It is enough to mention the classifiers (phenology, photosynthetic type, etc.). Alternatively, one could provide a table showing the different PFTs and their main attributes in the appendix, but I don't think this is necessary for this manuscript. I would appreciate some more information on the carbon and nitrogen pools mentioned on the same page, in brief. How do they communicate? On what time scales? There is an abrupt jump from leaf level photosynthesis (the models by Farquhar and Collatz) to the canopy level (GPP). Please add a short sentence explaining how photosynthesis is scaled to the canopy level. However, I think it would make more sense to explain this (p.4, lines 14-19) after equation 6, and not in the general description of the model. In addition, rather than describing the stomatal model in words on page 6 and showing the Equation in the discussion (Eq. 14), I would show the equation at this point. Please make sure that its original source is cited and that the notation is consistent: here you use "ca" for atmospheric $CO_2$, later in Equation 14 "CO2" is used. Equation 6: Could this equation

be double-checked? In my understanding the last term of this equation should be the overall resistance, i.e. 1/(1.6*gs) + 1/(1.4*gb), which differs from the term here. Please also check the equation on page 6 l.13, including the unit for conductance. The information on the LPX-Bern model is quite detailed and in some parts unnecessary. Again, the information on the PFTs can be shortened. E.g. for this paper it is not relevant what PFTs grow on peatland. Descriptive text elements such as "The CO2 flux from the atmosphere to the stomatal cavity is proportional to the CO2 difference between the atmosphere and the stomatal cavity (ca - ci)" are not needed and can be seen from the Equations (e.g. Eq.8) or are physical principles. The stomatal control as simulated in LPX is poorly described. It is stated that ci/ca is set to 0.8 for non-water stressed conditions. That reads as if ci/ca is constant whenever there is enough water, even under low light, high VPD etc. Further, it is not really clear how this optimization works. Is it an optimization in the sense of Cowan & Farquhar 1977? If yes, the original reference should be cited. If not, it would be good to either elaborate this aspect or cite another study at this point where it is explained in more detail. Are the two models forced with two different meteorological datasets? CRUNCEP and CRU TS3.23? Is there a reason for this? And could that affect the results in some way?

Just for clarification: when referring to delta 13C forcing (e.g. p.8, l.21) it would be clearer to write atmospheric delta 13C.

The sentence "An empirical convective boundary layer parameterization (Monteith, 1995) couples the carbon and water cycle" does not make sense to me. Please explain why the convective boundary layer couples the water and carbon cycle.

To me it would be more helpful to read how the leaf boundary layer is treated in the model, as it directly affects your calculations (see Eq. 6). In general, when describing the models I recommend putting more emphasis on the calculation of variables that are directly used for later calculations or referred to in the Results section (e.g. calculation of the leaf boundary layer, are there differences in how soil water stress affects gs or An?). This will certainly be of greater interest to the reader than a list of PFTs that

occurs in every land surface model in a similar form.

Page 8: not everyone is familiar with the discrimination model by Lloyd & Farquhar 1994. Please mention the key differences between the two models here (Lloyd & Farquhar 1994 and Farquhar 1989). The fact that the two formulations give similar trends is an interesting aspect but it is a little bit hidden in the Methods section. Lines 14 – 20 are better moved to the discussion and can be extended. I think it would be good to be more precise here: what processes are not considered in the discrimination formulation and what does that change or not change? For instance, why is the agreement with leaf delta 13C worse when the more complex model is used? Why does it not change the trend? Discussing such aspects may not be the focus of this study but it would be a valuable contribution to the discussion on how isotopes are (or should be) considered in global models.

Results Overall, this section is nicely written and clearly structured. Model results are compared to a study by Carvalhais et al. 2014. It would be good to provide a bit more detail here. Do you mean aboveground and belowground vegetation carbon? How was vegetation carbon estimated in the study by Carvalhais et al. 2014? Section 3.3: Results and Discussion are mixed here. It would be better to focus on the Results and discuss uncertainties in section 4. Just a thought: Why not taking PFT-specific model output? One could only take the corresponding PFT of the simulations that matches the PFT of the measured species. Up to the authors. The authors suggest that there is a stronger downregulation of stomatal conductance by water stress in LPX than in CLM4.5 in some regions. Here, it would be helpful to provide some possible explanations. Is it because water stress in LPX is stronger due to the climate forcing, the way soil moisture is simulated, or due to a stronger stomatal response to water stress?

Discussion This section contains many interesting thoughts, but its structure is not very clear. If it was divided in several subsections as it is the case for the results section, it would be easier to find certain aspects the reader is interested in. p. 17, l.14-18: This

paragraph can be expanded. As mentioned before, the differences between the discrimination model used here, and a more complex one (e.g. Lloyd & Farquhar, 1994), as well as possible implications for the simulated absolute values of discrimination and its trends can be discussed in more detail. p. 18: The question that the reader will have is: why does CLM4.5 simulate such a strong trend in iWUE? The authors provide two possible explanations: 1) the downregulation of photosynthesis by nitrogen, and 2) an inadequate relationship between simulated stomatal conductance and assimilation. The first one is described well and is supported by other recent studies. In this context it would be helpful to know how the fdreg factor in Equation 6 changes over time, and whether it affects the relationship between An and gs. Concerning the second explanation, I don't understand what the key message should be. Is it the general form of the Ball-Berry model and the prescribed relationship between gs and An? In this case it should be mentioned that this model or similar models are used in most land surface models (see e.g. Sato et al. 2015, JGR Biogeosciences). If the reason for the strong iWUE trend is due to an inadequate relationship between gs and An, we should see a similar behavior in other land surface models. A comparison with other models is missing here. It is then argued that the trend may partly be attributed to changes in relative humidity, but no data are shown that would support this statement. What does the CRUNCEP climate forcing dataset suggest? Is there a trend in relative humidity that could explain the strong trend to some extent? Do areas that show a decrease in discrimination also show a decrease in relative humidity? The role of relative humidity (and possibly other climate variables) is an interesting aspect to discuss at this point, but it should be supported by data and discussed in context of the factorial simulations that were made. It also mentioned that the value of the stomatal slope parameter m might be too high. It would be good to provide some more information, here or in the method section. What is the value of m? Is it constant across PFTs? I agree that m is probably too high for coniferous forests, but not necessarily for other vegetation types. If the value of m is to be discussed here, the authors should at least cite Lin et al. 2015, Nature Climate Change, who looked more generally at patterns of m across

[Figure]

PFTs. They used a slightly different model, but that shouldn't affect the patterns of m, see also Miner et al. 2016 Plant, Cell & Environment. Changing m would certainly affect the absolute values of iWUE and discrimination, but would it make a difference to the simulated percentage trend in iWUE as shown in Figure 7? If the value of m is taken as a reason for the overestimated trend in iWUE by CLM4.5 this needs to be shown somehow. In my opinion, a change in m would primarily change the spatial patterns of the simulated discrimination. The formulation implemented in the LPX is better able to capture the observed iWUE trend. But is that really because of the optimization? I would argue that also the Ball-Berry model (Eq. 14) predicts a constant ci/ca and thus a trend in iWUE that is proportional to ca, provided that rH and m do not change over time. In my eyes this is indicative of changes in rH, or more likely, problems with the nitrogen downregulation, as discussed earlier. Why not testing this? The CLM4.5 model could be run with a version that does not include the nitrogen downregulation. The comparison of this alternative version with the version used in this study could be used to answer the question whether the problem lies in the nitrogen downregulation or in the stomatal conductance scheme (Eq. 14 ). If the alternative model version still shows a stronger iWUE trend than expected, this would be a stronger indication that an optimization based approach indeed works better. Maybe new global runs are not necessary, and a simple analysis based on Eq. 6 would suffice. However, without testing this, the statement (p.20, l.27f) remains speculative and should not be mentioned in the conclusion of the paper. In general, this part of the discussion needs to be revised according to the comments above.

p. 19: The behavior of iWUE and ci/ca as reconstructed from the tree-ring measurements and modeled by LPX-Bern is compared to other studies. The nice thing on this paragraph is that it is very comprehensive. But it could be clearer with respect to the method used in the cited studies. Rather than just listing the studies you could sort them by method, i.e. mention other isotope-based studies first, then other methods. At the moment studies using the same methods (e.g. FACE) are mentioned in different parts of this section (Ainsworth & Long, 2005 and De Kauwe et al. 2013) which

seems a bit fuzzy. I think this aspect is important as different methods are associated with different uncertainties (which, however, do not have to be discussed here). With respect to the eddy covariance records it may be interesting to mention that a recent study (Knauer et al. 2016, New Phytologist) found that large-scale carbon and water fluxes are not in agreement with a constant $c_i$, but rather with a constant $c_i/c_a$.

p. 20: The effects of land use change and representation issues between the datasets/model simulations are adequately addressed. It may be helpful for the reader to mention the Figures again where the described aspects can be seen.

Figures In some figures (e.g. Fig.2), the color code and the associated numbers are very small and hard to read. It would be ok to have fewer color classes as they are hard to distinguish. Fig. 2: For the difference maps, please state what is subtracted from what, at least in the legend. Fig. 5: representing the differences in mean delta 13C as barplots is not appropriate here. I recommend to remove the bars and show the error bars only, also in Fig. 6. From Fig. 5 onwards: Some of the points on the map are hard to see. It would be helpful if their representation could be changed.

---

## Referee Comment (RC2) · A.P. Ballantyne (Referee) · 7 Jan 2017

Here the authors compare a compilation of tree ring and leaf isotope data from around the world with isotopic simulations from two common land surface models. While several studies have compared isotopic estimates of iWUE with model simulations of iWUE, especially at regional scales. This study is novel in that it is one of the few to actually investigate isotopic tracer simulations within models as a critical diagnostic for how accurate models are at simulating the global C cycle. In principle, this approach allows us to evaluate to what extent the terrestrial biosphere is being fertilized by increased atmospheric CO2; however, I think that the authors could further partition the response of iWUE into its component processes of assimilation and transpiration (at least in the models). This may also help reconcile why the models appear to show differing degrees of iWUE response.

General Comments:

I suspect that the two models investigated here differ considerably in how stomatal conductance is simulated and this is having a big impact ultimately on the isotopic tracers. While these models may be responding similarly to increases in atmospheric $CO_2$ they may be responding to different metrics of atmospheric water vapor. As the authors point out, assimilation in CLM is modeled as a function of RH and CO2, while it is my understanding that in LPJ stomatal conductance is modeled as a function of VPD. While RH and VPD may be inversely related in some environments, this is not always the case and their relationship might vary over the 20th century. It would be nice to see how assimilation and transpiration have responded over the 20th century independently in the two models. This may also help explain why LPX and CLM show different responses of iWUE over the 20th century.

Specific Comments:

P1L12 'water loss by transpiration.'

P2L4 'and water transpiration'

P2L22 Graven article is on 14C not 13C as cited. Check reference as they may have also included 13C in their simulations.

P2L27 conductance can be of CO2 or H2O, could be more specific here and say 'transpiration' as the process and H2O as the mass.

P3L7  While the authors mention many 13C tree ring records, they fail to mention the pioneering work by Tans et al. which is found in the references.

P3L22 'to complement recent advances in simulating marine carbon isotopes'

P4L24 'reactant to product'

P5L10  I believe that diffusion is only relevant for fractionation in non vascular plants such as bryophytes as well.

P6L6 more realistically related to the gradient between internal water pressure and atmospheric water pressure (approximated as vapor pressure deficit or VPD).

P7L7 del 13C signature of what?  Atmosphere? Please clarify

Eqn 8  Isn't this the same as Eqn 7?  But not quite sure c* is specified in Eqn 7.

P9L9 del 13 C is estimated as the 'weighted flux' of component GPP fluxes from PFTs from within grid cell. Omit 'GPP is used as a weight'.

Eqn 11. I don't think that this equation is necessary (especially given the number of equations already included) and this can simply be explained.

P10L14 While this approximation of 36.16 holds well within 0.1% isotopic ratio differences are per 1000, so is this enough significant figures?

P11L1 Not sure that you need to correct for the offset if you are only focusing on the trends and normalizing them across sites, regardless this should not affect your analysis.

P12L18 Were the CLM and LPX simulations conducted with or without land use change and does this have any impact on the global isotopic budget.

P13L13 'changes in the atmospheric del13C source'

P13L15 'globally-averaged' what? Soil, atmosphere?

P13L34 2.42 and 3.22 per mil these should have units

P14L8 Maybe these global mean estimates should be reported first before noting all the regional differences and more nuanced results.

P14L12 'bias of the models'

P14L28 Were any trend statistics (e.g. Mann-Kendall) conducted on the observations or models?

P15L10 These aren't really 'spatial' correlations

P15L15to21 This paragraph seems to fit better in the methods

P16L24 Model simulations with increased CO2 and constant climate change could be compared at least quantitatively to FACE data

P16L34 'Recall, however, that …represent annual or multi-annual averages that have been weighted by C assimilation or alloclation'

P17L28 This paragraph is rather short think about combining.

Eqn. 14 Would also be interesting to compare how conductance is simulated in LPX. While assimilation in both models is clearly responding to increasing atmospheric CO2, I suspect that transpiration may be responding differentially in the models due to different stomatal response to atmospheric water demand.

P18L27 For the CLM response you should look at the relative changes in CO2 and relative humidity over time (this should be a prognostic variable in the model). Also see work by Isaac Held on the response of the hydrologic cycle to atmospheric warming. Essentially, at the global scale RH does not change in response to warming; however, this might not be true over land. So it would be interesting to see in CLM how RH has changed at the tree ring sites.

P20L5 Similar work by Penuelas et al (2011) has shown an increase in water use efficiency but not necessarily an increase in annual ring width. However, a true test would be the relationship between WUE and Biomass- not sure if Klein looked at biomass in this study.

P20L9 It seems that both of these FACE studies report a consistent increase in WUE, but of slightly different magnitudes. It is interesting that the responses are so different between European forests and the N. American forests. Unfortunately, most of the FACE studies have been conducted in the Eastern US, where there are no tree ring isotope records.

Figure 3. not so sure that the Carvalhais estimates are 'observations', maybe 'derived from' or 'constrained by' observations.

Figures 5 and 6. I am not sure that you need both of these figures as they illustrate the same data. Perhaps move one to supplemental.

Figure 7. Can you include all of the tree ring records as thin grey traces in this figure? Would be nice to see some distribution of the observations to see if all the obs are bound by the model simulations.

Figure 8. Not sure that you need the discrimination equation, which should be defined in the text.

Figure 9. The right hand panels where certain variables have been kept constant is not explained in the caption

Figures 11 and 12. Once again these figures are both great but they illustrate redundant information maybe move one to the supplemental information.

In summary, with tree ring isotope data we are only able to approximate iWUE and cannot partition this response between assimilation and transpiration. However, in the models you can partition these processes, so it would be interesting to see how transpiration and assimilation are responding in the models, which may help identify processes that can reconcile these model simulations.

---

## Author Comment (AC1) · 23 Feb 2017

**Reply to Review Comments**

We thank the anonymous referee and Ashley Ballantyne for their thoughtful comments and for their time and effort to review this manuscript. Original review comments are given in black, our answer in red, and new or revised text added to the manuscript in blue fonts.

**Anonymous Referee #1**

In the manuscript "20th – century changes in carbon isotopes and water-use efficiency: Tree-ring based evaluation of the CLM4.5 and LPX-Bern models" Keller et al. present the implementation of a carbon isotope scheme in two global models as well as their performance with respect to simulated spatial patterns and decadal trends. The model results are compared to two different datasets, tree-ring records and bulk leaf delta 13C data. This study is a valuable contribution to ongoing efforts on the implementation of carbon isotopes in global vegetation models. The overall approach as well as the results are presented in an adequate and clear manner.

Thank you.

My main criticism is on the conclusions that are drawn from these results. Some parts of the discussion will be subject to revision. More detailed comments are listed below.

We will address your points below.

Abstract The abstract is a nice summary and contains all important aspects of the paper. However, I disagree with the last sentence. Suggesting "fundamental problems associated with the prescribed relationship between conductance and assimilation" is rather provocative and not supported by the results of this study. This relationship is strongly supported by observations (see e.g. Wong 1979 and also De Kauwe et al. 2013, Global Change Biology; papers cited in this study) and consequently used in most global models. I.e. it would indeed be a fundamental problem in our understanding of plant physiology. If this thought is brought up at such a prominent position in the paper, it needs to be better discussed and corroborated later in the manuscript, see below.

The sentence will be revised to read: "The results suggest that the down-regulation of $c_i/c_a$ and of photosynthesis by nitrogen limitation is possibly too strong in the standard setup of CLM4.5 or there may be problems associated with the implementation of conductance, assimilation, and related adjustment processes to long-term environmental changes."

We agree that the relationship is experimentally well-established. Yet, it remains controversial whether and how the relationship between conductance and assimilation changes under changing environmental conditions, particularly those now addressed in the model experiments such as the monotonic increase in atmospheric $CO_2$ over the past 170 years (Miner et al., 2016).

Introduction The introduction starts with a nice overview on the application of isotopes in the Earth System, which is a good motivation for this study. Following this part, the connection between carbon isotopes and plant physiological behavior is pointed out. The introduction closes with a very clear outline of the goals of this study.

Thank you.

Methods Page 4: the listing of all PFTs seems a bit unnecessary to me. It is enough to mention the classifiers (phenology, photosynthetic type, etc.). Alternatively, one could provide a table showing

the different PFTs and their main attributes in the appendix, but I don't think this is necessary for this manuscript.

Done. PFT listing is deleted.

I would appreciate some more information on the carbon and nitrogen pools mentioned on the same page, in brief. How do they communicate? On what time scales?

Noted. Text will be added as requested.

Most text in the following explanations is taken directly from the technical description (Oleson et al., 2013). Separate state variables for C and N are tracked for leaf, live stem, dead stem, live coarse root, dead coarse root, and fine root pools. Each of these pools has two corresponding storage pools representing, respectively, short-term and long-term storage of non-structural carbohydrates and labile nitrogen. There are two additional carbon pools, one for the storage of growth respiration reserves, and another used to meet excess demand for maintenance respiration during periods with low photosynthesis. One additional nitrogen pool tracks retranslocated nitrogen, mobilized from leaf tissue prior to abscission and litterfall. Altogether there are 20 state variables for vegetation carbon, and 19 for vegetation nitrogen.

Decomposition of fresh litter material (including C and N) into progressively more recalcitrant forms of soil organic matter is represented as a cascade of transformations between decomposing coarse woody debris (CWD), litter, and soil organic matter (SOM) pools. The decomposition flux is the product of a decomposition rate and source pool size.

Depending on the C:N ratios of the upstream and downstream pools and the amount of carbon lost in the transformation due to respiration, the execution of this potential carbon flux can generate either a source or a sink of new mineral nitrogen. Steps that result in an uptake of mineral nitrogen (immobilization fluxes) are subject to rate limitation, depending on the availability of mineral nitrogen, the total immobilization demand, and the total demand for soil mineral nitrogen to support new plant growth.

If mineral N is less than the demand to support new plant growth and immobilization, both plant growth and immobilization are downregulated by the same fraction in order to match N supply.

In addition to the cycling of nitrogen within the plant – litter – soil organic matter system, CLM represents external sources, including atmospheric deposition and biological nitrogen fixation. CLM also represents sinks, including nitrification, denitrification, leaching, and losses in fire.

There is an abrupt jump from leaf level photosynthesis (the models by Farquhar and Collatz) to the canopy level (GPP). Please add a short sentence explaining how photosynthesis is scaled to the canopy level. However, I think it would make more sense to explain this (p.4, lines 14-19) after equation 6, and not in the general description of the model.

Noted. Text will be added as requested.

The maximum rate of carboxylation at 25 °C varies with foliage nitrogen concentration and specific leaf area (Thornton and Zimmermann (2007)) and is a PFT specific parameter. It is assumed that leaf nitrogen and sunlight decrease exponentially with cumulative leaf area index from the canopy top to

bottom. Accordingly, the carboxylation rate and other photosynthesis parameter decrease exponentially within the canopy. Leaf level photosynthesis is scaled to the canopy level by integration over all leaf area. This is done separately for sunlit and shaded leaves and by considering the exponential scaling.

In addition, rather than describing the stomatal model in words on page 6 and showing the Equation in the discussion (Eq. 14), I would show the equation at this point. Please make sure that its original source is cited and that the notation is consistent: here you use "ca" for atmospheric CO2, later in Equation 14 "CO2" is used.

Done. Equation 14 moved to method section as requested. Reference to (Ball et al., 1987)will be added.

Equation 6: Could this equation be double-checked? In my understanding the last term of this equation should be the overall resistance, i.e. 1/(1.6*gs) + 1/(1.4*gb), which differs from the term here.

Done. Equation is correct and given as implemented in CLM.
Conductance, $g$, and resistance $r$, are inversely related. The boundary layer, $g_b$, and stomatal, $g_s$, conductance of $H_2O$ is related to the boundary layer, $r_b$, and stomatal, $r_s$, resistance of $H_2O$ by:

$$g_b=1/r_b \text{ and } g_s=1/r_s.$$

Total resistance is the sum of individual resistances. For $H_2O$ the total resistance is:
$$r= r_b+ r_s$$

For $CO_2$, diffusive transport is slower than for $H_2O$ as the molecular mass of $CO_2$ is higher than of $H_2O$. Thus, resistance is larger for $CO_2$ than for $H_2O$ (by a factor of 1.6 for stomatal, molecular diffusion and by a factor 1.4 assumed for the boundary layer). It holds for the total resistance of $CO_2$:

$$r_{CO2} = 1.6 \ r_s+1.4 \ r_b = 1.6/g_s+1.4/ \ g_b$$

*Rearranging yields for the conductance:*

$$g_{CO2} = 1/ \ r_{CO2} \ = \ 1 / (1.6/g_s+1.4/ \ g_b) = g_s \ g_b / (1.6 \ g_b + 1.4 \ g_s )$$

Please also check the equation on page 6 l.13, including the unit for conductance.

Done. Equation and units correct. Conductance is usually related to the mixing ratio and given in units of mol m$^{-2}$ s$^{-1}$. Here, conductance is related to the partial pressure of the gas and given in units mol m$^{-2}$ s$^{-1}$ Pa$^{-1}$). Typesetting has been corrected and "Pa-1" is now correctly typeset as "Pa$^{-1}$"

The information on the LPX-Bern model is quite detailed and in some parts unnecessary. Again, the information on the PFTs can be shortened. E.g. for this paper it is not relevant what PFTs grow on peatland. Descriptive text elements such as "The CO2 flux from the atmosphere to the stomatal cavity is proportional to the CO2 difference between the atmosphere and the stomatal cavity (ca - ci)" are not needed and can be seen from the Equations (e.g. Eq.8) or are physical principles.

Done. The list of PFTs is deleted for natural non-peatland areas. The PFTs for peatland are still mentioned as discrimination is set constant for Spaghnum Moss to 30 permil and is following the C3 path for C3 graminoids. This is of some relevance for the simulated d13C signature of vegetation and GPP as shown in Figure 4 and the global terrestrial isotopic budget as given in Figure 1 and Table 2. The sentence on the CO2 flux has been deleted as requested.

The stomatal control as simulated in LPX is poorly described. It is stated that ci/ca is set to 0.8 for non-water stressed conditions. That reads as if ci/ca is constant whenever there is enough water, even under low light, high VPD etc. Further, it is not really clear how this optimization works. Is it an optimization in the sense of Cowan & Farquhar 1977? If yes, the original reference should be cited. If not, it would be good to either elaborate this aspect or cite another study at this point where it is explained in more detail.

Noted. Text will be clarified as requested, and in the below we provide an answer to the above question and additional information.

Yes, ci/ca is set to 0.8 whenever there is enough water. Note that the photosynthesis-conductance routines are solved on a daily time step in LPX.

The stomatal control, as many other model details are described by Sitch et al. (2003). The following description is taken from Keel et al. (2016).

Daily evapotranspiration is calculated for each PFT as the minimum of a plant- and soil limited supply function ($E_{supply}$) and the demand for transpiration ($E_{demand}$). Esupply is the product of root-weighted soil moisture availability and a maximum water supply rate that is equal for all PFTs (Sitch et al., 2003). $E_{demand}$ is calculated following Monteith's (1995) empirical relation between evaporation efficiency and surface conductance:

$$E_{demand} = E_{eq}\alpha_m[1 - \exp(\frac{-g_c\phi}{g_m})] ,\qquad \text{(Eq. 1)}$$

where $E_{eq}$ is the equilibrium evaporation rate, $g_m$ and $\_\alpha_m$ are empirical parameters that are equal for all plant functional types, $g_c$ the canopy conductance, and $\phi$ the fraction of present foliage area to ground area (i.e., projected leaf area). Above equation is solved for $E_{demand}$ using the non-waterstressed potential canopy conductance as calculated by the photosynthesis routine for a fixed ratio $\lambda$ between the $CO_2$ mole fraction in the stomatal cavity and the ambient air. $\lambda$ is set equal to 0.8 following Sitch et al. (2003) to approximate non-water-stressed conditions and as a starting value for the iterative computation of carbon assimilation and transpiration. In case of water-stressed conditions when $E_{demand}$ exceeds $E_{supply}$, canopy conductance and photosynthesis are jointly and consistently down-regulated; $E_{demand}$ is set to $E_{supply}$ and Eq. (1) is solved for $g_c$.

Photosynthesis is modeled following Collatz et al. (1991, 1992), which is based on the formulations by Farquhar et al. (1980) and Farquhar and von Caemmerer (1982) generalized for global modeling purposes. The N content and Rubisco activity of leaves are assumed to vary seasonally and with canopy position in a way to maximize net assimilation at the leaf level. For C3 plants assimilation is a function of the daily integral of absorbed photosynthetically active radiation. For a detailed description see Haxeltine and Prentice (1996a,b).

Canopy conductance, $g_c$, is linked to daytime assimilation, $A_{dt}$, through

$$g_c = g_{min} + \frac{1.6 A_{dt}}{c_a(1-\lambda)}, \qquad \text{Eq. 2}$$

where $g_{min}$ is a PFT specific minimum canopy conductance, $c_a$ is the ambient mole fraction of $CO_2$, and $\lambda$ the ratio between the $CO_2$ mole fraction in the stomatal cavity and the ambient air. The equations for water supply and demand, assimilation, and canopy conductance are solved simultaneously by varying $\lambda$ to yield self-consistent values for $\lambda$, gc, assimilation, and transpiration.

Are the two models forced with two different meteorological datasets? CRUNCEP and CRU TS3.23? Is there a reason for this? And could that affect the results in some way?

Yes, the two models are forced by two different, though closely related products for technical reasons. CLM4.5 is run with a sub-hourly time step, while a daily time step is used in LPX.

This does not affect our conclusions and differences in these data products cannot explain differences in simulated discrimination and iWUE, as we explain below.

CRU NCEP (https://www.earthsystemgrid.org/dataset/ucar.cgd.ccsm4.CRUNCEP.v4.html) is used to force CLM4.5. CRU NCEP is a combination of two existing datasets (ftp://nacp.ornl.gov/synthesis/2009/frescati/model_driver/cru_ncep/analysis/readme.htm): The CRU TS.3.2 0.5°x0.5° monthly climatology covering the period 1901 to 2009 and the NCEP reanalysis 2.5°x2.5° 6 hours time step beginning in 1948.

The CRU TS3.2 climatology offers a good spatial resolution but only monthly mean fields are available which is a too low resolution for CLM4.5. On the other hand NCEP reanalysis has a temporal resolution of 6 hours and is compatible with the CLM4.5 time step. But the spatial resolution is low and precipitation of such reanalysis is known to be less reliable than CRU data based on station data.

As evident from Figure 9 and 10 of the manuscript, the influence on changes in climate on the change in discrimination and iWUE is small, except in semi-arid regions. But to more comprehensively address the reviewers' question, we further investigated the potential impact of using two different data products. The CRU NCEP data used to force CLM4.5 are interpolated on the 1x1 degree LPX grid and integrated to monthly values. This re-gridded CRU NCEP data are then used in LPX. LPX results for both climate input data, CRU NCEP and CRU-TS3.22, are compared. We investigated two cases, the standard model setup where climate is changing transiently and the factorial setup with "constant climate". Difference in simulated changes in discrimination and iWUE are small for both cases (Figure R1).

[Figure]

Figure R1: Century-scale changes in iWUE (%) of C3 trees as simulated by LPX forced by (a) the CRU-TS3.22 (top panels) and the CRU-NCEP climatology (bottom panel). The left panels show results obtained with the standard model setup including transient climate and $CO_2$ forcing. The right panels show results obtained by keeping climate constant. In both setups, the influence of the different climate input data on results is very small. Changes are based on decadal means (1990 minus 1900). Changes in iWUE as calculated from d13C tree-ring data are shown by colored circles. The upper left panel is shown as the bottom panel of Figure 11 in the originally submitted manuscript.

Just for clarification: when referring to delta 13C forcing (e.g. p.8, l.21) it would be clearer to write atmospheric delta 13C.

Done. The term forcing is not used anymore for prescribed atmospheric $CO_2$ and $^{13}C$ in the manuscript.

The sentence "An empirical convective boundary layer parameterization (Monteith, 1995) couples the carbon and water cycle" does not make sense to me. Please explain why the convective boundary layer couples the water and carbon cycle. To me it would be more helpful to read how the leaf boundary layer is treated in the model, as it directly affects your calculations (see Eq. 6). In general, when describing the models I recommend putting more emphasis on the calculation of variables that are directly used for later calculations or referred to in the Results section (e.g. calculation of the leaf boundary layer, are there differences in how soil water stress affects gs or An?). This will certainly be of greater interest to the reader than a list of PFTs that occurs in every land surface model in a similar form.

Noted. Text will be clarified as requested.

Please see our reply to the comments above on stomatal controls.

Page 8: not everyone is familiar with the discrimination model by Lloyd & Farquhar 1994. Please mention the key differences between the two models here (Lloyd & Farquhar 1994 and Farquhar 1989). The fact that the two formulations give similar trends is an interesting aspect but it is a little

bit hidden in the Methods section. Lines 14 – 20 are better moved to the discussion and can be extended. I think it would be good to be more precise here: what processes are not considered in the discrimination formulation and what does that change or not change? For instance, why is the agreement with leaf delta 13C worse when the more complex model is used? Why does it not change the trend? Discussing such aspects may not be the focus of this study but it would be a valuable contribution to the discussion on how isotopes are (or should be) considered in global models.

Noted. Text will be added and clarified as requested and we will consider moving lines 14 to 20 to the discussion.

Results Overall, this section is nicely written and clearly structured.

Thank you.

Model results are compared to a study by Carvalhais et al. 2014. It would be good to provide a bit more detail here. Do you mean aboveground and belowground vegetation carbon? How was vegetation carbon estimated in the study by Carvalhais et al. 2014?

Noted. Text will be clarified as requested.

Estimates of total vegetation carbon, including below and aboveground biomass, were derived by Carvalhais et al. 2014 from a collection of estimates for pan-tropical regions and for northern and temperate forests based on radar remote-sensing retrievals. Above- and below-ground biomass uncertainty for the tropical regions was propagated from errors in measurements, allometric relations, sampling and predictions. In the Northern Hemisphere, estimates accounted for uncertainties in the BIOMASAR GSVdata, wood density data and biomass compartment data. Herbaceous carbon is estimated from GPP data products. See the original publication for a more detailed description and references.

Section 3.3: Results and Discussion are mixed here. It would be better to focus on the Results and discuss uncertainties in section 4. Just a thought: Why not taking PFT-specific model output? One could only take the corresponding PFT of the simulations that matches the PFT of the measured species. Up to the authors.

We prefer to keep the text on the caveats and uncertainties related to the tree ring data-model comparison at the beginning of section 3.3.  This is important information for the reader and we feeö allows the reader to better put the results into perspective.

We prefer to use grid-cell average model output instead results for individual PFTs. As explained in section 3.3., differences in $\delta^{13}$C are reported for different species. In addition, differences within a species, even growing at the same site, can be as large or even larger as those between species. Differences in the rooting depth, water transport systems, root-to-leaf distance, leaf morphology, or in irradiance (sunlit versus shadow) affect discrimination. Further, the models as applied here use generic PFTs and not individual species.

The authors suggest that there is a stronger downregulation of stomatal conductance by water stress in LPX than in CLM4.5 in some regions. Here, it would be helpful to provide some possible explanations. Is it because water stress in LPX is stronger due to the climate forcing, the way soil moisture is simulated, or due to a stronger stomatal response to water stress?

Differences in the applied climate forcing are not responsible for the stronger downregulation of stomatal conductance in CLM compared to LPX (see Figure R1 and related discussion above).

Soil moisture and stomatal conductance are coupled and influence each other in both models in a non-linear way hampering somewhat a firm attribution of signals to these two processes. The available results suggest that the primary reason for the difference in modeled 20[th] century changes in discrimination and iWUE is rooted in the different parameterizations of the photosynthesis-conductance coupling in the two models.

[Figure]

Figure R2: Soil moisture (top) and 20[th] century changes (bottom) as simulated for CLM4.5 (left) and LPX (right) with the standard model setup. Changes are based on decadal means (1990 minus 1900). For CLM, soil liquid water content in kg m[-2] and its change (1990 minus 1900) is shown. For LPX, the water-filled fraction of the available water holding capacity in the top soil layer and its change is displayed.

[Figure]

Figure R3: As figure R2, but with a model setup where climate is kept constant.

A primary input to soil moisture is precipitation which is similar in both models. Patterns of soil moisture are similar in both models (Figure R2, top).

$20^{th}$ century changes in soil moisture are small in the factorial simulations where climate is kept constant (Figure R3, bottom). For the standard model setup, changes in soil moisture simulated by CLM and LPX are also small in large parts of Europe and Asia, where most of our tree-ring data are located. Despite these small changes in soil moisture, large differences in discrimination and iWUE changes are found between the two models in these simulations and regions (see Figure 9, 10, 11).

This is in line with the conclusion stated in the manuscript at the end of section 3.4 (p15, l33-35): "This suggests that the simulated decrease in discrimination and in c_i/c_a in the CLM4.5 runs is mainly linked to increasing $CO_2$ and a corresponding downregulation of c_i/c_a (Eq. 6). The downregulation in CLM4.5 is larger than suggested by the tree-ring records."

The following text is added to the manuscript at the end of section 3.4: "Soil moisture and stomatal conductance are coupled and influence each other and in turn $^{13}C$ discrimination. $20^{th}$ century changes in soil moisture are small in the factorial simulations where climate is kept constant. For the standard model setup, changes in soil moisture simulated by CLM and LPX are also small in large parts of Europe and Asia, where most of our tree-ring data are located. Despite these small changes in soil moisture, large differences in discrimination changes are found between the two models in these simulations and regions (Figure 9). This suggests that the primary reason for the model-model difference in simulated $20^{th}$ century changes in discrimination is rooted in the different parameterizations of the photosynthesis-conductance coupling."

Discussion This section contains many interesting thoughts, but its structure is not very clear. If it was divided in several subsections as it is the case for the results section, it would be easier to find certain aspects the reader is interested in.

Noted. We will aim to clarify the structure of the discussion section.

p. 17, l.14-18: This paragraph can be expanded. As mentioned before, the differences between the discrimination model used here, and a more complex one (e.g. Lloyd & Farquhar, 1994), as well as possible implications for the simulated absolute values of discrimination and its trends can be discussed in more detail.

Noted. Text will be added as requested.

p. 18: The question that the reader will have is: why does CLM4.5 simulate such a strong trend in iWUE? The authors provide two possible explanations: 1) the downregulation of photosynthesis by nitrogen, and 2) an inadequate relationship between simulated stomatal conductance and assimilation. The first one is described well and is supported by other recent studies. In this context it would be helpful to know how the fdreg factor in Equation 6 changes over time, and whether it affects the relationship between An and gs.

The downregulation by nitrogen does not directly affect An and gs for carbon. It affects the amount of GPP that is allocated to the carbon pools. We will clarify this when revising the model description in the method section.

[Figure]

Figure R4: An index for nitrogen limitation, FPG top) and its 20[th] century change (bottom) as simulated by CLM4.5. FPG is indicative of nitrogen limitation and varies between 1 (no nitrogen limitation) and 0.

Nitrogen demand is the amount of nitrogen required to transfer all available carbon from photosynthesis into plant carbon. The fraction FPG describes by how much nitrogen demand to support plant growth is downregulated to match nitrogen availability. A value of 1 corresponds to non-nitrogen limited conditions. As shown in figure R3, 20[th] century changes in annual and grid-cell mean FPG are relatively small.

The fraction $f_{dreg}$ used to calculate the intercellular $CO_2$ partial pressure and thus [13]C discrimination (Eq. 4 and 6 in the submitted manuscript) is closely related to FPG. We therefore also expect that annual mean values in $f_{dreg}$ changed little over the 20[th] century (Unfortunately, output for $f_{dreg}$ is not available).

It remains difficult to draw conclusions from annual or monthly values of FPG and $f_{dreg}$ as these variables are evaluated on the model time step and influence simulated fractionation and iWUE in a non-linear way (see e.g. Eq 4, 6, and 9 in the submitted manuscript).

Concerning the second explanation, I don't understand what the key message should be. Is it the general form of the Ball-Berry model and the prescribed relationship between gs and An? In this case it should be mentioned that this model or similar models are used in most land surface models (see e.g. Sato et al. 2015, JGR Biogeosciences). If the reason for the strong iWUE trend is due to an inadequate relationship between gs and An, we should see a similar behavior in other land surface models. A comparison with other models is missing here.

CLM4.5 employs the Ball-Berry equation using a constant time invariant slope value, m (Eq. 14). Thus, any potential adjustment of $m$ to changes in environmental conditions, including the century-scale increase in atmospheric $CO_2$ or changes in water stress are not considered. It is currently unclear whether such adjustment processes occur and our current understanding of the underlying physiological mechanisms of stomatal responses is less than complete (e.g. (Miner et al., 2016)).

Sato et al. (2015) investigate the influence of the use of vapor pressure deficit (VPD) or relative humidity (RH) in Ball-Berry-type stomatal conductance formulations. About half of the investigated models apply RH as a driving variable (Eq. 14), despite that VPD is considered the more relevant controlling factor. The global warming simulations reveal an increase in VPD and little change in RH. Their results suggest that the increase in VPD under global warming leads to a stronger downregulation of stomatal conductance ($g_s$) and ci/ca for the VPD formulations compared to the RH formulation. This implies that replacing RH by VPD in the Ball-Berry equation may, without further adjustments, even increase the disagreement between modeled and reconstructed changes in discrimination and in iWUE.

We will clarify the text in the revised manuscript according to the above two paragraphs.

It is then argued that the trend may partly be attributed to changes in relative humidity, but no data are shown that would support this statement. What does the CRUNCEP climate forcing dataset suggest? Is there a trend in relative humidity that could explain the strong trend to some extent? Do areas that show a decrease in discrimination also show a decrease in relative humidity? The role of relative humidity (and possibly other climate variables) is an interesting aspect to discuss at this point, but it should be supported by data and discussed in context of the factorial simulations that were made.

This was not our intention. We do not argue that the changes in discrimination are due to long-term changes in relative humidity. We rather suggest "that assimilation is shifted towards times with lower relatively humidity (page 18, line 28)".

This point is rather subtle and follows from Eq. 14. We will delete the text from lines 25 to 31 on page 18 of the submitted manuscript to avoid misunderstanding and for brevity.

It also mentioned that the value of the stomatal slope parameter m might be too high. It would be good to provide some more information, here or in the method section. What is the value of m? Is it constant across PFTs? I agree that m is probably too high for coniferous forests, but not necessarily for other vegetation types. If the value of m is to be discussed here, the authors should at least cite Lin et al. 2015, Nature Climate Change, who looked more generally at patterns of m across PFTs. They used a slightly different model, but that shouldn't affect the patterns of m, see also Miner et al. 2016 Plant, Cell & Environment.

Text will be adjusted as requested and references added. The value of $m$ is now mentioned in the methods section. Parameter values are $m$=9 for C3 plants and for $m$=4 C4 plants. These parameters are time invariant.

The review of Miner et al. (2016) yields mean values for $m$ of 9.8, 8.7, and 6.8 for angiosperm evergreen, angiosperm deciduous and for gymnosperm trees, respectively (Their figure 1).

Changing m would certainly affect the absolute values of iWUE and discrimination, but would it make a difference to the simulated percentage trend in iWUE as shown in Figure 7? If the value of m is taken as a reason for the overestimated trend in iWUE by CLM4.5 this needs to be shown somehow. In my opinion, a change in m would primarily change the spatial patterns of the simulated discrimination.

Duarte et al. (2016) present in their figure A1 results from sensitivity simulations where only the parameter *m* was changed in the model setup. The change in leaf $\delta^{13}$C is larger and the change in discrimination is smaller for the lower value of m (m equal 6 intead of 9).

The formulation implemented in the LPX is better able to capture the observed iWUE trend. But is that really because of the optimization? I would argue that also the Ball-Berry model (Eq. 14) predicts a constant ci/ca and thus a trend in iWUE that is proportional to ca, provided that rH and m do not change over time. In my eyes this is indicative of changes in rH, or more likely, problems with the nitrogen downregulation, as discussed earlier. Why not testing this? The CLM4.5 model could be run with a version that does not include the nitrogen downregulation. The comparison of this alternative version with the version used in this study could be used to answer the question whether the problem lies in the nitrogen downregulation or in the stomatal conductance scheme (Eq. 14 ). If the alternative model version still shows a stronger iWUE trend than expected, this would be a stronger indication that an optimization based approach indeed works better. Maybe new global runs are not necessary, and a simple analysis based on Eq. 6 would suffice. However, without testing this, the statement (p.20, l.27f) remains speculative and should not be mentioned in the conclusion of the paper. In general, this part of the discussion needs to be revised according to the comments above.

We will delete the statement ("and may be interpreted as supportive of a plant strategy towards optimizing assimilation and minimizing water loss under changing environmental conditions ") on p20, l27 as requested. The remaining text will read: "The results suggest constant ci/ca over the 20th century. The results suggest that it is desirable to adjust the implementation of photosynthesis and conductance in CLM4.5 towards a better agreement with observation-derived century-scale trends in $\delta^{13}$C discrimination and intrinsic water-use efficiency."

p. 19: The behavior of iWUE and ci/ca as reconstructed from the tree-ring measurements and modeled by LPX-Bern is compared to other studies. The nice thing on this paragraph is that it is very comprehensive. But it could be clearer with respect to the method used in the cited studies. Rather than just listing the studies you could sort them by method, i.e. mention other isotope-based studies first, then other methods. At the moment studies using the same methods (e.g. FACE) are mentioned in different parts of this section (Ainsworth & Long, 2005 and De Kauwe et al. 2013) which seems a bit fuzzy. I think this aspect is important as different methods are associated with different uncertainties (which, however, do not have to be discussed here).

Noted, we will consider this option during the revision of the discussion section.

With respect to the eddy covariance records it may be interesting to mention that a recent study (Knauer et al. 2016, New Phytologist) found that large-scale carbon and water fluxes are not in agreement with a constant ci, but rather with a constant ci/ca.

Noted. We will complement the text and add the citation.

p. 20: The effects of land use change and representation issues between the datasets/model simulations are adequately addressed. It may be helpful for the reader to mention the Figures again where the described aspects can be seen.

Noted. Figures will be mentioned where appropriate.

Figures In some figures (e.g. Fig.2), the color code and the associated numbers are very small and hard to read. It would be ok to have fewer color classes as they are hard to distinguish.

Noted. We will reduce the number of labels and increase their size in Figures 2 to 4. We will also change units from gC m$^{-2}$ to kgC m$^{-2}$ to reduce the number of digits.

Fig. 2: For the difference maps, please state what is subtracted from what, at least in the legend.

The differences are based on decadal means (1996-2005 minus 1896-1905). The caption will be adjusted.

Fig. 5: representing the differences in mean delta13C as barplots is not appropriate here. I recommend to remove the bars and show the error bars only, also in Fig. 6.

We prefer to keep the bars for a clear distinction between the difference and the error estimates.

From Fig. 5 onwards: Some of the points on the map are hard to see. It would be helpful if their representation could be changed.

Our choice represents a trade-off between the visibility of individual dots and the visibility of all dots.

**Review 2: A. Ballantyne**

Here the authors compare a compilation of tree ring and leaf isotope data from around the world with isotopic simulations from two common land surface models. While several studies have compared isotopic estimates of iWUE with model simulations of iWUE, especially at regional scales. This study is novel in that it is one of the few to actually investigate isotopic tracer simulations within models as a critical diagnostic for how accurate models are at simulating the global C cycle. In principle, this approach allows us to evaluate to what extent the terrestrial biosphere is being fertilized by increased atmospheric CO2; however, I think that the authors could further partition the response of iWUE into its component processes of assimilation and transpiration (at least in the models). This may also help reconcile why the models appear to show differing degrees of iWUE response.

General Comments:

I suspect that the two models investigated here differ considerably in how stomatal conductance is simulated and this is having a big impact ultimately on the isotopic tracers. While these models may be responding similarly to increases in atmospheric CO2 they may be responding to different metrics of atmospheric water vapor. As the authors point out, assimilation in CLM is modeled as a function of RH and CO2, while it is my understanding that in LPJ stomatal conductance is modeled as a function of VPD. While RH and VPD may be inversely related in some environments, this is not always the case and their relationship might vary over the 20th century. It would be nice to see how assimilation and transpiration have responded over the 20th century independently in the two models. This may also help explain why LPX and CLM show different responses of iWUE over the 20th century.

[Figure]

Figure R5: Simulated transpiration (in mm/s) for the decade around year 2000 (top) and 20[th] century changes (bottom; 1996-2005 minus 1896-1905).

The panels shown in Figure R5 will be added to figure 2, next to the panels showing GPP and GPP changes (Layout and labels will be adjusted).

Simulated 20[th] century changes in annual-mean transpiration are generally small in both models, except in Australia for LPX and in parts of Latin America for CLM4.5. Generally an increase in transpiration is found in boreal and temperate forest regions in both models. Transpiration is slightly decreasing in LPX and slightly increasing in CLM4.5 in most tropical forest regions. The overall increase in water use efficiency (not shown), the ratio of assimilated carbon to transpired water, is primarily driven by the increase in GPP in both models.

**Specific Comments:**

P1L12 'water loss by transpiration.'

Done. Text changed as requested.

P2L4 'and water transpiration'

Done. Text changed as requested.

P2L22 Graven article is on 14C not 13C as cited. Check reference as they may have also included 13C in their simulations.

Done. Reference removed.

P2L27 conductance can be of CO2 or H2O, could be more specific here and say 'transpiration' as the process and H2O as the mass.

Stomatal conductance is used here as a general term for $H_2O$ and $CO_2$. Transport of $^{12}CO_2$, $^{13}CO2$, and $H_2O$ through the stomata are by molecular diffusion. This process depends on molecule mass and leads to the "fractionation" between the $^{13}CO_2$ and $^{12}CO_2$ fluxes. Text is not changed.

P3L7 While the authors mention many 13C tree ring records, they fail to mention the pioneering work by Tans et al. which is found in the references.

Done. Reference added.

P3L22 'to complement recent advances in simulating marine carbon isotopes'

Done. Sentence changed to read "This is a step towards fully coupled isotope-enabled CESM applications and complements recent advances in simulating marine carbon isotopes".

P4L24 'reactant to product'

Done. Text changed as requested.

P5L10 I believe that diffusion is only relevant for fractionation in non vascular plants such as bryophytes as well.

Done. Text clarified by adding reference to Farquhar et al., 1989 at the beginning of the paragraph. Non-vascular plants are not included in the standard setup of CLM4.5

P6L6 more realistically related to the gradient between internal water pressure and atmospheric water pressure (approximated as vapor pressure deficit or VPD).

Done. Text clarified. The description of the relationship between conductance and assimilation in CLM4.5 is clarified and Eq. 14 (submitted MS) is moved here to page 6.

P7L7 del 13C signature of what? Atmosphere? Please clarify

Done. Text clarified to read: "Note that no $\delta^{13}$C observational data, e.g., from tree-rings or atmospheric samples, was used as a constraint in the assimilation."

Eqn 8 Isn't this the same as Eqn 7? But not quite sure c* is specified in Eqn 7.

Noted. Model description will be revised in reply to the comments by referee 1 and Eq. 8 may be deleted.

P9L9 del 13 C is estimated as the 'weighted flux' of component GPP fluxes from PFTs from within grid cell. Omit 'GPP is used as a weight'.

Done. Sentence deleted as requested.

Eqn 11. I don't think that this equation is necessary (especially given the number of equations already included) and this can simply be explained.

Done. Equation 11 deleted and text adjusted.

P10L14 While this approximation of 36.16 holds well within 0.1% isotopic ratio differences are per 1000, so is this enough significant figures?

Done. Text clarified. The approximation is for Eq. 12 (in the submitted MS) and not for 36.16. It holds exactly: q=1.6 (a-b) = 1.6 (27-4.4) = 36.16. Additionally, four significant digits are enough as units here correspond to "permil units" and are not "per 1000".Text clarified to read: "The approximation given in Eq. 12 holds well within 0.1%."

P11L1 Not sure that you need to correct for the offset if you are only focusing on the trends and normalizing them across sites, regardless this should not affect your analysis.

We agree with the reviewer that this correction hardly influences our analyses and does not alter conclusions. Nevertheless, the relative change in iWUE depends weakly on the magnitude of the discrimination ($\Delta_i(t_1)$) as evident from equation 13 and as clearly stated in the manuscript.

P12L18 Were the CLM and LPX simulations conducted with or without land use change and does this have any impact on the global isotopic budget.

Yes, anthropogenic land use is considered in the CLM and LPX simulations as stated on p6, line 26 for CLM and on page 8, line 22). Land use maps from Hurtt et al., 2006 are used in both models. Implications for the global isotopic budget are discussed on page 20, line 14 to 21 in the submitted manuscript.

P13L13 'changes in the atmospheric del13C source'

Done. Text adjusted as requested.

P13L15 'globally-averaged' what? Soil, atmosphere?

Done. Text clarified to read: "There is a substantial offset between the models for globally-averaged soil and vegetation pools, with CLM4.5 being around 5 per mil more negative."

P13L34 2.42 and 3.22 per mil these should have units

Done. Per mil units added.

P14L8 Maybe these global mean estimates should be reported first before noting all the regional differences and more nuanced results.

Done. Global mean estimates are now reported before the regional differences.

P14L12 'bias of the models'

Done. Typo corrected.

P14L28 Were any trend statistics (e.g. Mann-Kendall) conducted on the observations or models?

As suggested, we performed a Mann-Kendall analysis on the discrimination time series. The trend in discrimination of C3 trees is most monotonic in CLM-4.5 ($\tau = -0.9$), less monotonic in the average tree-ring record $\tau = -0.6$ for annual data and -0.7 for decadal mean data) and almost negligible in the LPX-Bern output ($\tau = -0.2$).

The Mann-Kendall test provides information to which extent a trend is monotonic, but does not provide information on the magnitude of the trend. We do not plan to add the results from this test to the manuscript. It is clear from Figure 7 that the trends in discrimination and iWUE are larger in CLM4.5 than in LPX and the tree-ring data (see also Figure R6 below).

P15L10 These aren't really 'spatial' correlations

Done. Word spatial deleted.

P15L15to21 This paragraph seems to fit better in the methods

Done. Paragraph moved to method section. This implied a slight rearrangement of the subsections in the method section. Spin-up and transient simulations for both CLM4.5 and LPX are now described in the same subsection.

P16L24 Model simulations with increased CO2 and constant climate change could be compared at least quantitatively to FACE data.

This is beyond the scope of this study. For a comparison between carbon-nitrogen land models and FACE data we refer to the literature, e.g., Zaehle et al. (2014)

P16L34 'Recall, however, that …represent annual or multi-annual averages that have been weighted by C assimilation or alloclation'

Done. Text adjusted as proposed.

P17L28 This paragraph is rather short think about combining.

We prefer to keep the paragraph as is to give the same weight to the conclusions drawn at the end of this and the previous paragraph.

Eqn. 14 Would also be interesting to compare how conductance is simulated in LPX. While assimilation in both models is clearly responding to increasing atmospheric CO2, I suspect that transpiration may be responding differentially in the models due to different stomatal response to atmospheric water demand.

Noted. The methods section will be adjusted to better explain how conductance is modelled in LPX. See response to a similar comment by reviewer 1.

P18L27 For the CLM response you should look at the relative changes in CO2 and relative humidity over time (this should be a prognostic variable in the model). Also see work by Isaac Held on the response of the hydrologic cycle to atmospheric warming. Essentially, at the global scale RH does not change in response to warming; however, this might not be true over land. So it would be interesting to see in CLM how RH has changed at the tree ring sites.

This point is rather subtle and follows from Eq. 14. We will  delete the text from line 25 to 31 on page 18 of the submitted manuscript to avoid misunderstanding and for brevity. Please see also reply to reviewer 1, concerning this point.

P20L5 Similar work by Penuelas et al (2011) has shown an increase in water use efficiency but not necessarily an increase in annual ring width. However, a true test would be the relationship between WUE and Biomass- not sure if Klein looked at biomass in this study.

Noted. We will mention the work by (Peñuelas et al., 2011).

P20L9 It seems that both of these FACE studies report a consistent increase in WUE, but of slightly different magnitudes. It is interesting that the responses are so different between European forests and the N. American forests. Unfortunately, most of the FACE studies have been conducted in the Eastern US, where there are no tree ring isotope records.

Thank you. We do not plan to extend the discussion on these FACE studies.

Figure 3. not so sure that the Carvalhais estimates are 'observations', maybe 'derived from' or 'constrained by' observations.

Done. Text modified to read "derived from observations".

Figures 5 and 6. I am not sure that you need both of these figures as they illustrate the same data. Perhaps move one to supplemental.

We will keep both figures as part of the manuscript. This decision is also in response to the comment of reviewer 1 on the visibility of the tree-ring data dots. Showing only regional maps as in Figure 6 would provide incomplete information of the global picture, whereas showing only the global map as in Figure 5 would affect clarity and visibility of regional results.

Figure 7. Can you include all of the tree ring records as thin grey traces in this figure? Would be nice to see some distribution of the observations to see if all the obs are bound by the model simulations.

[Figure]

Figure R6: Changes in discrimination and iWUE. Similar as figure 7 in the original manuscript, but with individual tree ring series added (thin gray lines). All model and tree-ring series represent decadally-smoothed values.

[Figure]

Figure R7: Same as Figure R6, but tree-ring series represent annual values.

We will replace Figure 7 with Figure R6 as requested.

Figure 8. Not sure that you need the discrimination equation, which should be defined in the text.

Done. Equation removed from figure caption. Equation 5, showing this definition, is now mentioned.

Figure 9. The right hand panels where certain variables have been kept constant is not explained in the caption

Done. Explanation added to the caption.

Figures 11 and 12. Once again these figures are both great but they illustrate redundant information maybe move one to the supplemental information.

We will keep both figures as part of the manuscript for the same reason as discussed above for figures 5 and 6

In summary, with tree ring isotope data we are only able to approximate iWUE and cannot partition this response between assimilation and transpiration. However, in the models you can partition these processes, so it would be interesting to see how transpiration and assimilation are responding in the models, which may help identify processes that can reconcile these model simulations.

Figure on transpiration will be added as requested. See response to main comment above.

References:

Ball, J. T., Woodrow, I. E., and J.A. Berry, J. A.: A model predicting stomatal conductance and its contribution to the control of photosynthesis under different environmental conditions., in: Progress in Photosynthesis Research. Vol. 4. Proceedings of the 7th International Congress on Photosynthesis, edited by: Biggins, J., Martinsus, Nijhoff Publishers, Dordrecht, The Netherlands, 221–224, 1987.

Duarte, H. F., Raczka, B. M., Ricciuto, D. M., Lin, J. C., Koven, C. D., Thornton, P. E., Bowling, D. R., Lai, C. T., Bible, K. J., and Ehleringer, J. R.: Evaluating the Community Land Model (CLM 4.5) at a Coniferous Forest Site in Northwestern United States Using Flux and Carbon-Isotope Measurements, Biogeosciences Discuss., 2016, 1-35, 10.5194/bg-2016-441, 2016.

Keel, S. G., Joos, F., Spahni, R., Saurer, M., Weigt, R. B., and Klesse, S.: Simulating oxygen isotope ratios in tree ring cellulose using a dynamic global vegetation model, Biogeosciences, 13, 3869-3886, 10.5194/bg-13-3869-2016, 2016.

Miner, G. L., Bauerle, W. L., and Baldocchi, D. D.: Estimating the sensitivity of stomatal conductance to photosynthesis: A review, Plant, Cell & Environment, n/a-n/a, 10.1111/pce.12871, 2016.

Oleson, K., Lawrence, D. M., Bonan, G. B., B. Drewniak, Huang, M., C.D. Koven, C. D., Levis, S., Li, F., Riley, W. J., Subin, Z. M., Swenson, S., Thornton, P. E., Bozbiyik, A., Fisher, R., C.L. Heald, C. L., Kluzek, E., Lamarque, J. F., Lawrence, P. J., Leung, L. R., Lipscomb, W., Muszala, S. P., Ricciuto, D. M., Sacks, W. J., Sun, Y., Tang, J., and Yang, Z.-L.: Technical description of version 4.5 of the Community Land Model (CLM), Boulder, CO, 420, 2013.

Peñuelas, J., Canadell, J. G., and Ogaya, R.: Increased water-use efficiency during the 20th century did not translate into enhanced tree growth, Global Ecology and Biogeography, 20, 597-608, 10.1111/j.1466-8238.2010.00608.x, 2011.

Sato, H., Kumagai, T. o., Takahashi, A., and Katul, G. G.: Effects of different representations of stomatal conductance response to humidity across the African continent under warmer $CO_2$-enriched climate conditions, Journal of Geophysical Research: Biogeosciences, 120, 979-988, 10.1002/2014jg002838, 2015.

Sitch, S., Smith, B., Prentice, I. C., Arneth, A., Bondeau, A., Cramer, W., Kaplan, J. O., Levis, S., Lucht, W., Sykes, M. T., Thonicke, K., and Venevsky, S.: Evaluation of ecosystem dynamics, plant geography and terrestrial carbon cycling in the LPJ dynamic global vegetation model, Global Change Biology, 9, 161-185, 2003.

Zaehle, S., Medlyn, B. E., De Kauwe, M. G., Walker, A. P., Dietze, M. C., Hickler, T., Luo, Y., Wang, Y.-P., El-Masri, B., Thornton, P., Jain, A., Wang, S., Warlind, D., Weng, E., Parton, W., Iversen, C. M., Gallet-Budynek, A., McCarthy, H., Finzi, A., Hanson, P. J., Prentice, I. C., Oren, R., and Norby, R. J.: Evaluation of 11 terrestrial carbon–nitrogen cycle models against observations from two temperate Free-Air $CO_2$ Enrichment studies, New Phytologist, 202, 803-822, 10.1111/nph.12697, 2014.

---

## Author Response (AR1)

**Reply to Review Comments**

We thank the anonymous referee and Ashley Ballantyne for their thoughtful comments and for their time and effort to review this manuscript. Original review comments are given in black, our answer in red, and new or revised text added to the manuscript in blue fonts. A revised version in track change mode is attached to this reply.

**Anonymous Referee #1**

In the manuscript "20th – century changes in carbon isotopes and water-use efficiency: Tree-ring based evaluation of the CLM4.5 and LPX-Bern models" Keller et al. present the implementation of a carbon isotope scheme in two global models as well as their performance with respect to simulated spatial patterns and decadal trends. The model results are compared to two different datasets, tree-ring records and bulk leaf delta 13C data. This study is a valuable contribution to ongoing efforts on the implementation of carbon isotopes in global vegetation models. The overall approach as well as the results are presented in an adequate and clear manner.

Thank you.

My main criticism is on the conclusions that are drawn from these results. Some parts of the discussion will be subject to revision. More detailed comments are listed below.

We will address your points below.

Abstract The abstract is a nice summary and contains all important aspects of the paper. However, I disagree with the last sentence. Suggesting "fundamental problems associated with the prescribed relationship between conductance and assimilation" is rather provocative and not supported by the results of this study. This relationship is strongly supported by observations (see e.g. Wong 1979 and also De Kauwe et al. 2013, Global Change Biology; papers cited in this study) and consequently used in most global models. I.e. it would indeed be a fundamental problem in our understanding of plant physiology. If this thought is brought up at such a prominent position in the paper, it needs to be better discussed and corroborated later in the manuscript, see below.

The sentence is revised to read: "The results suggest that the down-regulation of $c_i/c_a$ and of photosynthesis by nitrogen limitation is possibly too strong in the standard setup of CLM4.5 or there may be problems associated with the implementation of conductance, assimilation, and related adjustment processes to long-term environmental changes."

We agree that the relationship is experimentally well-established. Yet, it remains controversial whether and how the relationship between conductance and assimilation changes under changing environmental conditions, particularly those now addressed in the model experiments such as the monotonic increase in atmospheric $CO_2$ over the past 170 years (Miner et al., 2016).

Introduction The introduction starts with a nice overview on the application of isotopes in the Earth System, which is a good motivation for this study. Following this part, the connection between carbon isotopes and plant physiological behavior is pointed out. The introduction closes with a very clear outline of the goals of this study.

Thank you.

Methods Page 4: the listing of all PFTs seems a bit unnecessary to me. It is enough to mention the classifiers (phenology, photosynthetic type, etc.). Alternatively, one could provide a table showing the different PFTs and their main attributes in the appendix, but I don't think this is necessary for this manuscript.

Done. PFT listing is deleted.

I would appreciate some more information on the carbon and nitrogen pools mentioned on the same page, in brief. How do they communicate? On what time scales?

Done. The following text is added: "Vegetation comprises fifteen different PFTs, which are classified into three different phenological groups: evergreen, seasonal-deciduous, and stress-deciduous. Fourteen of these PFTs follow the C3 photosynthetic pathway (11 tree, 2 grasses and crops) and one the C4 path (warm grasses). Altogether, 20 carbon and 19 nitrogen pools per PFT represent carbon (C) and nitrogen (N) in vegetation.  C and N are tracked for leaf, live stem, dead stem, live coarse root, dead coarse root, and fine root pools and corresponding storage pools representing, respectively, short-term and long-term storage of non-structural carbohydrates and labile nitrogen.

Decomposition of fresh litter material (including C and N) into progressively more recalcitrant forms of soil organic matter is represented as a cascade of transformations between decomposing coarse woody debris, litter, and soil organic matter pools. Depending on the C:N ratios of involved pools and the amount of carbon lost by respiration, each transformation can generate either a source or a sink of new mineral nitrogen.

Steps that result in an uptake of mineral nitrogen (e.g., immobilization fluxes) are subject to rate limitation, depending on the availability of mineral nitrogen and the sum of nitrogen demands from immobilization, photosynthesis, nitrification, and denitrification. If mineral N is less than the sum of these demands, fluxes are downregulated in order to match N supply. We note that the "Relative Demand" downregulation in CLM4.5 has recently been shown to be inaccurate in several tropical (Zhu et al., 2016b), tundra (Zhu et al., 2016a), and grassland (Zhu et al., 2017) systems. In addition to the cycling of nitrogen within the plant - litter - soil organic matter system, CLM represents external sources, including atmospheric deposition and biological nitrogen fixation. CLM also represents other N sinks not included in this budgeting, including leaching and losses in fire."

There is an abrupt jump from leaf level photosynthesis (the models by Farquhar and Collatz) to the canopy level (GPP). Please add a short sentence explaining how photosynthesis is scaled to the canopy level. However, I think it would make more sense to explain this (p.4, lines 14-19) after equation 6, and not in the general description of the model.

Done. The following text is added:

"The maximum rate of carboxylation at 25°C varies with an assumed static foliage nitrogen concentration and specific leaf area and is a PFT specific parameter. It is assumed that leaf nitrogen and sunlight decrease exponentially with cumulative leaf area index from the canopy top to bottom. Accordingly, the carboxylation rate and other photosynthesis parameter decrease exponentially within the canopy. Leaf level photosynthesis is scaled to the canopy level by integration over the total leaf area. This is done separately for sunlit and shaded leaves and by considering the exponential scaling."

In addition, rather than describing the stomatal model in words on page 6 and showing the Equation in the discussion (Eq. 14), I would show the equation at this point. Please make sure that its original source is cited and that the notation is consistent: here you use "ca" for atmospheric CO2, later in Equation 14 "CO2" is used.

Done. Equation 14, now Eq. 7, moved to method section as requested. Reference to (Ball et al., 1987) is added.

Equation 6: Could this equation be double-checked? In my understanding the last term of this equation should be the overall resistance, i.e. 1/(1.6*gs) + 1/(1.4*gb), which differs from the term here.

Done. Equation is correct and given as implemented in CLM.
Conductance, $g$, and resistance $r$, are inversely related. The boundary layer, $g_b$, and stomatal, $g_s$, conductance of $H_2O$ is related to the boundary layer, $r_b$, and stomatal, $r_s$, resistance of $H_2O$ by:

$$g_b = 1/r_b \text{ and } g_s = 1/r_s.$$

Total resistance is the sum of individual resistances. For $H_2O$ the total resistance is:

$$r = r_b + r_s$$

For $CO_2$, diffusive transport is slower than for $H_2O$ as the molecular mass of $CO_2$ is higher than of $H_2O$. Thus, resistance is larger for $CO_2$ than for $H_2O$ (by a factor of 1.6 for stomatal, molecular diffusion and by a factor 1.4 assumed for the boundary layer). It holds for the total resistance of $CO_2$:

$$r_{CO2} = 1.6\, r_s + 1.4\, r_b = 1.6/g_s + 1.4/\,g_b$$

*Rearranging yields for the conductance:*

$$g_{CO2} = 1/\,r_{CO2} \;=\; 1\,/\,(1.6/g_s + 1.4/\,g_b) = g_s\, g_b\,/\,(1.6\,g_b + 1.4\,g_s\,)$$

Please also check the equation on page 6 l.13, including the unit for conductance.

Done. Equation and units correct. Conductance is usually related to the mixing ratio and given in units of mol m$^{-2}$ s$^{-1}$. In the submitted manuscript, conductance is related to the partial pressure of the gas and given in units mol m$^{-2}$ s$^{-1}$ Pa$^{-1}$). To avoid confusion and for consistency with the notations used to describe LPX, we redefined $c_i$ and $c_a$ to represent $CO_2$ concentrations instead of $CO_2$ partial pressure. This required some small adjustments at various places in the method section.

The information on the LPX-Bern model is quite detailed and in some parts unnecessary. Again, the information on the PFTs can be shortened. E.g. for this paper it is not relevant what PFTs grow on peatland. Descriptive text elements such as "The CO2 flux from the atmosphere to the stomatal cavity is proportional to the CO2 difference between the atmosphere and the stomatal cavity (ca - ci)" are not needed and can be seen from the Equations (e.g. Eq.8) or are physical principles.

Done. The list of PFTs is deleted for natural non-peatland areas. The PFTs for peatland are still mentioned as discrimination is set constant for sphagnum moss to 30 permil and is following the C3 path for C3 graminoids. This is of some relevance for the simulated d13C signature of vegetation and GPP as shown in Figure 4 and the global terrestrial isotopic budget as given in Figure 1 and Table 2.

The sentence on the CO2 flux is deleted as requested. In addition the following sentence is deleted: "Tree PFTs and fires are excluded from the agricultural cell fraction."

The stomatal control as simulated in LPX is poorly described. It is stated that ci/ca is set to 0.8 for non-water stressed conditions. That reads as if ci/ca is constant whenever there is enough water, even under low light, high VPD etc. Further, it is not really clear how this optimization works. Is it an optimization in the sense of Cowan & Farquhar 1977? If yes, the original reference should be cited. If not, it would be good to either elaborate this aspect or cite another study at this point where it is explained in more detail.

Done. Yes, $c_i/c_a$ is set to 0.8 whenever there is enough water supply from soils. Note that the photosynthesis-conductance routines are solved on a daily time step in LPX.

The model description of LPX has been revised and mirrors now in its structure the description of CLM4.5. The description of photosynthesis and conductance is expanded and the nitrogen cycle is briefly described. The previous text is replaced by: "Photosynthesis and stomatal control in LPX is described by (Haxeltine and Prentice, 1996a) as summarized elsewhere (Keel et al., 2016;Sitch et al., 2003). The equations for water supply from soil and transpiration, assimilation, and canopy conductance are solved simultaneously by varying the ratio $c_i/c_a$, also termed $\lambda$, to yield self-consistent values for these properties.

Total day time net photosynthesis, $A_{dt}$, is modeled following (Collatz et al., 1991;Collatz et al., 1992) which is a Farquhar model (Farquhar et al., 1980) generalized for global modeling purposes (for details, see (Haxeltine and Prentice, 1996b) ). $A_{dt}$ is a function of incoming radiation, temperature, day length, and atmospheric $CO_2$ partial pressure and $\lambda$. $A_{dt}$ is computed from a formulation which gives a gradual transition between light-limited and Rubisco-limited rate of photosynthesis. The amount of photosynthetically active radiation absorbed by the entire canopy increases with the modelled leaf area index following Beer's law and is used to compute the light-limited photosynthesis rate. The N content and Rubisco activity of leaves are assumed to vary seasonally and with canopy position in a way to maximize $A_{dt}$.

Canopy conductance, $g_c$, is linked to $A_{dt}$ through

$$g_c = g_{min} + 1.6 \ A_{dt} / (c_a (1-\lambda)) \qquad (9)$$

where $g_{min}$ is a PFT specific minimum canopy conductance.

Daily transpiration of water is calculated for each PFT as the minimum of a plant- and soil limited supply function ( $E_{supply}$) and the demand for transpiration $E_{demand}$. $E_{supply}$ is the product of root-weighted soil moisture availability and a maximum water supply rate that is equal for all PFTs. $E_{demand}$ is calculated following the empirical relation between evaporation efficiency and surface conductance of (Monteith, 1995):

$$E_{demand} = E_{eq} \ \alpha_m \ (1\text{-}\exp(-g_c \ \phi/g_m)) \qquad (10)$$

where $E_{eq}$ is the equilibrium evaporation rate, dependent on temperature and radiation, $g_m$ and $\alpha_m$ are empirical parameters, $g_c$ is the canopy conductance, and $\phi$ the fraction of present foliage area to ground area (i.e., projected leaf area). Eq. 10 is solved for $E_{demand}$ using the non-water-stressed potential canopy conductance which is calculated using Eq. 9 and a fixed ratio $\lambda$ to compute $A_{dt}$. $\lambda$ is

set equal to 0.8 following Sitch et al. (2003) to approximate non-water-stressed conditions and as a starting value for the iterative computation of carbon assimilation and transpiration under water shortage. In case of water-stressed conditions when $E_{demand}$ exceeds $E_{supply}$, canopy conductance and photosynthesis are down-regulated; $E_{demand}$ is set to $E_{supply}$ and Eq. 10 is solved for $g_c$. Knowing $g_c$ and $c_a$, $\lambda$ is varied in the photosynthesis module until the following relationship is satisfied:

$$A_{dt}(\lambda) = ((g_c - g_{min})/1.6) \ c_a \ (1-\lambda) \qquad\qquad (11)$$

NPP is downregulated on the daily model time step if N-demand exceeds N-availability from the inorganic soil nitrogen pools. Daily NPP is integrated over a year and allocated annually to vegetation. Thus, in contrast to CESM there is no immediate feedback of nitrogen limitation on isotopic discrimination on a daily time scale, but there is a long-term feedback by annual changes in vegetation structure and, in turn, photosynthesis and carbon assimilation."

Are the two models forced with two different meteorological datasets? CRUNCEP and CRU TS3.23? Is there a reason for this? And could that affect the results in some way?

Yes, the two models are forced by two different, though closely related, products for technical reasons. CLM4.5 is run with a sub-hourly time step, while a daily time step is used in LPX.

This does not affect our conclusions and differences in these data products cannot explain differences in simulated discrimination and iWUE, as we explain below.

CRU NCEP (https://www.earthsystemgrid.org/dataset/ucar.cgd.ccsm4.CRUNCEP.v4.html) is used to force CLM4.5. CRU NCEP is a combination of two existing datasets (ftp://nacp.ornl.gov/synthesis/2009/frescati/model_driver/cru_ncep/analysis/readme.htm): The CRU TS.3.2 0.5°x0.5° monthly climatology covering the period 1901 to 2009 and the NCEP reanalysis 2.5°x2.5° 6 hours time step beginning in 1948.

The CRU TS3.2 climatology offers a good spatial resolution but only monthly mean fields are available which is a too low resolution for CLM4.5. On the other hand NCEP reanalysis has a temporal resolution of 6 hours and is compatible with the CLM4.5 time step. But the spatial resolution is low and precipitation of such reanalysis is known to be less reliable than CRU data based on station data.

As evident from Figure 9 and 10 of the manuscript, the influence on changes in climate on the change in discrimination and iWUE is small, except in semi-arid regions. But to more comprehensively address the reviewers' question, we further investigated the potential impact of using two different data products. The CRU NCEP data used to force CLM4.5 are interpolated on the 1°x1° LPX grid and integrated to monthly values. This re-gridded CRU NCEP data are then used in LPX. LPX results for both climate input data, CRU NCEP and CRU-TS3.22, are compared. We investigated two cases, the standard model setup where climate is changing transiently and the factorial setup with "constant climate". Difference in simulated changes in discrimination and iWUE are small for both cases (Figure R1).

[Figure]

Figure R1: Century-scale changes in iWUE (%) of C3 trees as simulated by LPX forced by (a) the CRU-TS3.22 (top panels) and the CRU-NCEP climatology (bottom panel). The left panels show results obtained with the standard model setup including transient climate and $CO_2$ forcing.  The right panels show results obtained by keeping climate constant. In both setups, the influence of the different climate input data on results is very small. Changes are based on decadal means (1990 minus 1900). Changes in iWUE as calculated from d13C tree-ring data are shown by colored circles.  The upper left panel is shown as the bottom panel of Figure 11 in the originally submitted manuscript.

The following text is added to the section describing the climate forcing data: "The two models are forced with two slightly different climate data sets as CLM4.5 is run on a sub-hourly time step, while a daily time step is used in LXP. CRU NCEP, used to force CLM4.5, is a combination of the CRU TS3.2 monthly climatology (resolution of 0.5°x0.5°) and the NCEP reanalysis product with a time step of six hours (2.5° x2.5°). The results of the factorial runs, presented later, suggest generally a small influence of this difference in forcings on simulated discrimination and iWUE. We further investigated the potential impact of using two different climate data products. The CRU NCEP data are interpolated on the 1° x1° LPX grid and integrated to monthly values for use in LPX. LPX is run with the modified CRU NCEP and, as usual, with CRU-TS3.22 in the standard model setup where climate is changing transiently and the factorial setup with "constant climate" (cCLIM). Difference in simulated changes in discrimination and iWUE are small for both cases. In conclusion, the large model differences, presented in the result section, are not caused by differences in climate input data."

Just for clarification: when referring to delta 13C forcing (e.g. p.8, l.21) it would be clearer to write atmospheric delta 13C.

Done. The term forcing is not used anymore for prescribed atmospheric $CO_2$ and [13]C in the manuscript.

The sentence "An empirical convective boundary layer parameterization (Monteith, 1995) couples the carbon and water cycle" does not make sense to me. Please explain why the convective boundary layer couples the water and carbon cycle. To me it would be more helpful to read how the leaf boundary layer is treated in the model, as it directly affects your calculations (see Eq. 6). In general, when describing the models I recommend putting more emphasis on the calculation of

variables that are directly used for later calculations or referred to in the Results section (e.g. calculation of the leaf boundary layer, are there differences in how soil water stress affects gs or An?). This will certainly be of greater interest to the reader than a list of PFTs that occurs in every land surface model in a similar form.

Done. Please see our reply to the comments above on stomatal controls.

Page 8: not everyone is familiar with the discrimination model by Lloyd & Farquhar 1994. Please mention the key differences between the two models here (Lloyd & Farquhar 1994 and Farquhar 1989). The fact that the two formulations give similar trends is an interesting aspect but it is a little bit hidden in the Methods section. Lines 14 – 20 are better moved to the discussion and can be extended. I think it would be good to be more precise here: what processes are not considered in the discrimination formulation and what does that change or not change? For instance, why is the agreement with leaf delta 13C worse when the more complex model is used? Why does it not change the trend? Discussing such aspects may not be the focus of this study but it would be a valuable contribution to the discussion on how isotopes are (or should be) considered in global models.

Done. Text moved to discussion section and expanded as requested:

"In applications with the predecessor of LPX (Joos et al., 2004;Scholze et al., 2008) $\delta^{13}C$ was implemented following (Scholze et al., 2003) with discrimination modeled following (Lloyd and Farquhar, 1994). Here, we adjusted the discrimination formulations and use instead the simpler formulations of (Farquhar et al., 1989) for consistency with CLM4.5 and with the computation of iWUE from tree-ring $\delta^{13}C$ measurements. We note that our conclusions do not depend on this choice. LPX yields similar 20$^{th}$-century changes in discrimination and iWUE for both formulations. However, simulated $\delta^{13}C$ of carbon assimilated by C3 trees is generally less negative (by about 2 permil) when applying the (Lloyd and Farquhar, 1994) formulation and agreement with leaf $\delta^{13}C$ is less favorable.

The difference between the two implementations arises from additional processes considered in the approach of (Lloyd and Farquhar 1994) and different parameter choice. Here, fractionations associated with the diffusion of $CO_2$ from the stomatal cavity to the cell wall, entrance of $CO_2$ in solution at the cell wall, and transport within the cell, as well as photorespiration are neglected. The discriminations by the first three processes are set to be constant in the earlier implementation following (Scholze et al., 2003), neglecting a minor temperature dependency associated with the carboxylation by PEP-c (Lloyd and Farquhar 1994). The discrimination associated with respiration varies with atmospheric $CO_2$ and the photocompensation point, but is small. In addition, b, the discrimination during photosynthetic fixation, is set to 27 in this study and to 27.5 in the earlier implementation. Overall, the consideration of the additional processes and the difference in b result in an approximately constant offset between the two implementations. This highlights that absolute values of discrimination depend on uncertain model parameters. Agreement or disagreement between leaf and model data (or tree ring and model data) concerning absolute levels of $\delta^{13}C$ (Figure 5 and 6) may not be interpreted as an indication of the performance of the conductance/photosynthesis module."

Results Overall, this section is nicely written and clearly structured.

Thank you.

Model results are compared to a study by Carvalhais et al. 2014. It would be good to provide a bit more detail here. Do you mean aboveground and belowground vegetation carbon? How was vegetation carbon estimated in the study by Carvalhais et al. 2014? Done. Estimates of total vegetation carbon, including below and aboveground biomass, were derived by Carvalhais et al. 2014 from a collection of estimates for pan-tropical regions and for northern and temperate forests based on radar remote-sensing retrievals. Above- and below-ground biomass uncertainty for the tropical regions was propagated from errors in measurements, allometric relations, sampling and predictions. In the Northern Hemisphere, estimates accounted for uncertainties in the BIOMASAR GSVdata, wood density data and biomass compartment data. Herbaceous carbon is estimated from GPP data products. See the original publication for a more detailed description and references.

The following text is added to the manuscript: "Estimates of total vegetation carbon, including below and aboveground biomass, were derived by Carvalhais et al., 2014 from a collection of estimates for pan-tropical regions and for northern and temperate forests based on radar remote-sensing retrievals."

Section 3.3: Results and Discussion are mixed here. It would be better to focus on the Results and discuss uncertainties in section 4. Just a thought: Why not taking PFT-specific model output? One could only take the corresponding PFT of the simulations that matches the PFT of the measured species. Up to the authors.

We prefer to keep the text on the caveats and uncertainties related to the tree ring data-model comparison at the beginning of section 3.3. This is important information for the reader and we feel allows the reader to better put the results into perspective. In response to the comment, the sentence "Differences in the rooting depth, water transport systems, root-to-leaf distance, leaf morphology, or in irradiance (sunlit versus shadow) affect discrimination." is deleted and we added the following text in the discussion section:

"The influence of rooting depth, water transport systems, root-to-leaf distance, leaf morphology, and irradiance (sunlit versus shadow) on discrimination is neglected. Although, these factors may affect discrimination and reported isotopic differences within a species, even growing at the same site, can be as large or even larger as those between species (McCarroll and Loader, 2004; Leuenberger, 2007)."

We prefer to use grid-cell average model output instead results for individual PFTs. As explained in section 3.3., differences in $\delta^{13}$C are reported for different species. In addition, differences within a species, even growing at the same site, can be as large or even larger as those between species. Differences in the rooting depth, water transport systems, root-to-leaf distance, leaf morphology, or in irradiance (sunlit versus shadow) affect discrimination. Further, the models as applied here use generic PFTs and not individual species.

The authors suggest that there is a stronger downregulation of stomatal conductance by water stress in LPX than in CLM4.5 in some regions. Here, it would be helpful to provide some possible explanations. Is it because water stress in LPX is stronger due to the climate forcing, the way soil moisture is simulated, or due to a stronger stomatal response to water stress?

Differences in the applied climate forcing are not responsible for the stronger downregulation of stomatal conductance in CLM compared to LPX (see Figure R1 and related discussion above).

Soil moisture and stomatal conductance are coupled and influence each other in both models in a non-linear way hampering somewhat a firm attribution of signals to these two processes. The available results suggest that the primary reason for the difference in modeled 20th century changes in discrimination and iWUE is rooted in the different parameterizations of the photosynthesis-conductance coupling in the two models.

[Figure]

Figure R2: Soil moisture (top) and 20th century changes (bottom) as simulated for CLM4.5 (left) and LPX (right) with the standard model setup. Changes are based on decadal means (1990 minus 1900). For CLM, soil liquid water content in kg m$^{-2}$ and its change (1990 minus 1900) is shown. For LPX, the water-filled fraction of the available water holding capacity in the top soil layer and its change is displayed.

[Figure]

Figure R3: As figure R2, but with a model setup where climate is kept constant.

A primary input to soil moisture is precipitation, which is similar in both models. Patterns of soil moisture are similar in both models (Figure R2, top).

20[th] century changes in soil moisture are small in the factorial simulations where climate is kept constant (Figure R3, bottom). For the standard model setup, changes in soil moisture simulated by CLM and LPX are also small in large parts of Europe and Asia, where most of our tree-ring data are located. Despite these small changes in soil moisture, large differences in discrimination and iWUE changes are found between the two models in these simulations and regions (see Figure 9, 10, 11).

This is in line with the conclusion stated in the manuscript at the end of section 3.4 (p15, l33-35): "This suggests that the simulated decrease in discrimination and in $c_i/c_a$ in the CLM4.5 runs is mainly linked to increasing $CO_2$ and a corresponding downregulation of $c_i/c_a$ (Eq. 6). The downregulation in CLM4.5 is larger than suggested by the tree-ring records."

The following text is added to the manuscript at the end of section 3.4: "Soil moisture and stomatal conductance are coupled and influence each other and in turn discrimination. 20[th] century changes in soil moisture are small in the factorial simulations where climate is kept constant. For the standard model setup, changes in soil moisture simulated by CLM and LPX are also small in large parts of Europe and Asia, where most of our tree-ring data are located. Despite these small changes in soil moisture, large differences in discrimination changes are found between the two models in these simulations and regions (Figure 9). This suggests that the primary reason for the model-model difference in simulated 20[th] century changes in discrimination is rooted in the different parameterizations of the photosynthesis-conductance coupling."

Discussion This section contains many interesting thoughts, but its structure is not very clear. If it was divided in several subsections as it is the case for the results section, it would be easier to find certain aspects the reader is interested in.

Done. Subsections are introduced in the discussion section. These mirror the presentation of results. The subsection titles are:
1. "Isotopic signatures, pools and fluxes of the global land biosphere"
2. "$\delta^{13}$C in leaves of C3 plants: modern distribution"
3. "20[th] century changes in carbon isotopes and water use efficiency of C3 plants"
4. "Consistency of a constant ci/ca with previous studies"

In addition, Discussion and Conclusions are now separated into two sections.

p. 17, l.14-18: This paragraph can be expanded. As mentioned before, the differences between the discrimination model used here, and a more complex one (e.g. Lloyd & Farquhar, 1994), as well as possible implications for the simulated absolute values of discrimination and its trends can be discussed in more detail.

Done. The paragraph is expanded and the following text is added: "The influence of rooting depth, water transport systems, root-to-leaf distance, leaf morphology, and irradiance (sunlit versus shaded) on discrimination is neglected. However, these factors may affect discrimination. Reported isotopic differences within a species, even growing at the same site, can be as large or even larger as those between species (McCarroll and Loader, 2004; Leuenberger, 2007)."

As mentioned above, the difference between the two isotopic discrimination models and implications are now explained in the discussion (subsection on $\delta^{13}$C in leaves of C3 plants.) See our response to the earlier comment on this point for the new text

p. 18: The question that the reader will have is: why does CLM4.5 simulate such a strong trend in iWUE? The authors provide two possible explanations: 1) the downregulation of photosynthesis by nitrogen, and 2) an inadequate relationship between simulated stomatal conductance and assimilation. The first one is described well and is supported by other recent studies. In this context it would be helpful to know how the fdreg factor in Equation 6 changes over time, and whether it affects the relationship between An and gs.

The downregulation by nitrogen does not directly affect $g_s$ for the carbon assimilation calculation. It affects the amount of GPP that is allocated to the carbon pools. The following text is added to the method section 2.1.2, where discrimination for CLM4.5 is described, to make this point explicitly: "In other words, plants photosynthesize carbon at a potential, nitrogen unlimited rate. Photosynthesis $a_n$ and, in turn, stomatal conductance $g_s$ are not directly affected by nitrogen limitation."

The paragraph in the discussion section on p17, bottom has been reformulated. The original text "In CLM4.5, photosynthesis for isotopic discrimination is downregulated immediately, on the subhourly time step of the model, by limited nitrogen availability in CLM4.5 (Eq. 6). This can lead to a depression of assimilation in the isotope calculation during times of high assimilation." is changed to read:

"Regarding nitrogen limitation, photosynthesis for isotopic discrimination is downregulated immediately, on the subhourly time step of the model, by limited nitrogen availability in CLM4.5 (Eq. 6). This can lead to a depression of assimilation in the isotope calculation during times of high assimilation. As explained in the method section, photosynthesis of carbon and stomatal conductance for water is computed, unlike for discrimination, without nitrogen limitation. Instead, the allocation of gross primary productivity to carbon pools is downregulated under nitrogen limitation). This downregulation is on annual average generally less than 20% in forested regions and 20[th] century changes in downregulation are generally small (within ±2%) in areas with tree ring data."

[Figure]

Figure R4: An index for nitrogen limitation, FPG top) and its 20[th] century change (bottom) as simulated by CLM4.5. FPG is indicative of nitrogen limitation and varies between 1 (no nitrogen limitation) and 0.

Nitrogen demand is the amount of nitrogen required to transfer all available carbon from photosynthesis into plant carbon. The fraction FPG describes by how much nitrogen demand to support plant growth is downregulated to match nitrogen availability.  A value of 1 corresponds to non-nitrogen limited conditions. As shown in Figure R4, 20th century changes in annual and grid-cell mean FPG are relatively small.

The fraction $f_{dreg}$ used to calculate the intercellular $CO_2$ partial pressure and thus $^{13}C$ discrimination (Eq. 4 and 6 in the submitted manuscript) is closely related to FPG. We therefore also expect that annual mean values in $f_{dreg}$ changed little over the 20th century (Unfortunately, output for $f_{dreg}$ is not available).

It remains difficult to draw conclusions from annual or monthly values of FPG and $f_{dreg}$ as these variables are evaluated on the model time step and influence simulated fractionation and iWUE in a non-linear way (see e.g. Eqs. 4, 6, and 9 in the submitted manuscript and Eqs. 4, 6, and 12 in the resubmitted manuscript).

Concerning the second explanation, I don't understand what the key message should be. Is it the general form of the Ball-Berry model and the prescribed relationship between gs and An? In this case it should be mentioned that this model or similar models are used in most land surface models (see e.g. Sato et al. 2015, JGR Biogeosciences). If the reason for the strong iWUE trend is due to an inadequate relationship between gs and An, we should see a similar behavior in other land surface models. A comparison with other models is missing here.

The text "Another possibility for the model-data mismatch is that the photosynthesis formulation in CLM4.5 may not adequately represent the relationship between stomatal conductance and assimilation" is changed to read:

"Another possibility for the model-data mismatch is that the photosynthesis formulation in CLM4.5 may not adequately represent the relationship between stomatal conductance and assimilation and related adjustment processes to long-term environmental changes.  CLM4.5 employs the experimentally well-verified and widely-used Ball-Berry equation (Eq. 7) applying globally uniform, time invariant slope parameters, $m$, for C3 and C4 plants. Thus, any potential adjustment of $m$ to changes in environmental conditions, including the century-scale increase in atmospheric CO2 or changes in water stress are not considered. It is currently unclear whether such adjustment processes occur and the current understanding of the underlying physiological mechanisms of stomatal responses is incomplete (e.g. Miner et al., 2017)."

New text was added to describe the results of Sato and co-workers and resulting conclusions for this work:

"(Sato et al., 2015) investigated the influence of the use of vapor pressure deficit (VPD) versus relative humidity (RH) in Ball-Berry-type stomatal conductance formulations. About half of the investigated models apply RH as a driving variable (Eq. 7), despite the fact that VPD is considered the more relevant controlling factor. The global warming simulations reveal an increase in VPD and little change in RH. Their results suggest that the increase in VPD under global warming leads to a stronger downregulation of stomatal conductance ($g_s$) and $c_i/c_a$ for the VPD formulations compared to the RH formulation. This implies that replacing RH by VPD in the Ball-Berry equation in CLM4.5 may, without further adjustments, even increase the disagreement between modeled and reconstructed changes in discrimination and in iWUE."

It is then argued that the trend may partly be attributed to changes in relative humidity, but no data are shown that would support this statement. What does the CRUNCEP climate forcing dataset suggest? Is there a trend in relative humidity that could explain the strong trend to some extent? Do areas that show a decrease in discrimination also show a decrease in relative humidity? The role of relative humidity (and possibly other climate variables) is an interesting aspect to discuss at this point, but it should be supported by data and discussed in context of the factorial simulations that were made.

This was not our intention. We do not argue that the changes in discrimination are due to long-term changes in relative humidity. We rather suggest "that assimilation is shifted towards times with lower relative humidity (page 18, line 28)".

This point is rather subtle and follows from Eq. 14 (now Eq. 7 in the resubmitted manuscript). The text from lines 25 to 31 on page 18 of the submitted manuscript is deleted to avoid misunderstanding and for brevity.

It also mentioned that the value of the stomatal slope parameter m might be too high. It would be good to provide some more information, here or in the method section. What is the value of m? Is it constant across PFTs? I agree that m is probably too high for coniferous forests, but not necessarily for other vegetation types. If the value of m is to be discussed here, the authors should at least cite Lin et al. 2015, Nature Climate Change, who looked more generally at patterns of m across PFTs. They used a slightly different model, but that shouldn't affect the patterns of m, see also Miner et al. 2016 Plant, Cell & Environment.

Done. The value of $m$ is now mentioned in the methods section. Parameter values are $m$=9 for C3 plants and for $m$=4 C4 plants. These parameters are time invariant.

The review of Miner et al. (2016) yields mean values for $m$ of 9.8, 8.7, and 6.8 for angiosperm evergreen, angiosperm deciduous and for gymnosperm trees, respectively (Their figure 1). Aranibar et al. (2005) also inferred larger values of $m$ using observed foliar 13C values in pine.

The following text is now added in the discussion:

"The value of $m$ is set to nine in CLM4.5 for all C3 plants world-wide. This value is within the observational range, but species differences (e.g., evergreen versus deciduous) are neglected by using a single value. The review of (Miner et al., 2017) yields mean values for $m$ of 9.8, 8.7, and 6.8 for angiosperm evergreen, angiosperm deciduous and for gymnosperm trees, respectively (Their figure 1).  (Aranibar et al., 2006) inferred values of $m$ around  10 to 12 using observed foliar $^{13}$C values in pine."

Changing m would certainly affect the absolute values of iWUE and discrimination, but would it make a difference to the simulated percentage trend in iWUE as shown in Figure 7? If the value of m is taken as a reason for the overestimated trend in iWUE by CLM4.5 this needs to be shown somehow. In my opinion, a change in m would primarily change the spatial patterns of the simulated discrimination.

(Duarte et al., 2016) present in their figure A1 results from sensitivity simulations where only the parameter $m$ was changed in the model setup. The change in leaf $\delta^{13}$C is larger and the change in

discrimination is smaller for the lower value of m (m equal 6 instead of 9). The following sentence is added: "In addition, the 20th century change in isotopic discrimination is reduced."

The formulation implemented in the LPX is better able to capture the observed iWUE trend. But is that really because of the optimization? I would argue that also the Ball-Berry model (Eq. 14) predicts a constant ci/ca and thus a trend in iWUE that is proportional to ca, provided that rH and m do not change over time. In my eyes this is indicative of changes in rH, or more likely, problems with the nitrogen downregulation, as discussed earlier. Why not testing this? The CLM4.5 model could be run with a version that does not include the nitrogen downregulation. The comparison of this alternative version with the version used in this study could be used to answer the question whether the problem lies in the nitrogen downregulation or in the stomatal conductance scheme (Eq. 14 ). If the alternative model version still shows a stronger iWUE trend than expected, this would be a stronger indication that an optimization based approach indeed works better. Maybe new global runs are not necessary, and a simple analysis based on Eq. 6 would suffice. However, without testing this, the statement (p.20, l.27f) remains speculative and should not be mentioned in the conclusion of the paper. In general, this part of the discussion needs to be revised according to the comments above.

The following statement is deleted: "and may be interpreted as supportive of a plant strategy towards optimizing assimilation and minimizing water loss under changing environmental conditions" on p20, l27 as requested. The remaining text reads: "The results suggest constant ci/ca over the 20th century. The results suggest that it is desirable to adjust the implementation of photosynthesis and conductance in CLM4.5 towards a better agreement with observation-derived century-scale trends in $\delta^{13}C$ discrimination and intrinsic water-use efficiency."

Further, we have clarified the discussion on photosynthesis-conductance relationship and optimization. The existing text has been replaced by:

"Alternative conductance-photosynthesis formulations, may be preferable, compared to the Ball-Berry relation as used in CLM4.5. The Ball-Berry relation is viewed as consistent with an optimization (Cowan I. R. and D., 1977) of stomatal conductance towards maintaining a constant water use efficiency by optimizing carbon gain per unit water loss. However, a number of alternative formulations are found in the literature. These differ in potentially important ways. For example, (Medlyn et al., 2011) suggest that the slope parameter increases with increasing temperature; further the inverse of the square root of vapor pressure deficit is used instead of relative humidity in their relationship. (Prentice et al., 2014) and (Wang et al., 2016) rely on an optimality hypothesis considering the costs of maintaining both water flow and photosynthetic capacity. These authors use information on the spatial gradients in $\delta^{13}C$ from stable isotope measurements on leaf material (Cornwell et al., 2016) to develop their photosynthesis-conductance relation. The slope parameter proposed by (Prentice et al., 2014) depends on a number of variables, e.g., the temperature-dependent Michaelis-Menten coefficient for Rubisco-limited photosynthesis, viscosity of water, or the water potential difference between soil and leaf or foliage height. Hence, the slope parameter varies with environmental conditions in this approach. (Bonan et al., 2014) implemented different photosynthesis-conductance modules within a CLM4.5 model version featuring a multi-layer canopy and compared results with leaf analyses and eddy covariance fluxes at six forest sites. The continuous soil-plant-atmosphere (SPA) module, optimizing carbon gain per unit water loss and considering hydraulic safety, performs similar or better than the Ball-Berry formulations used in CLM4.5. A better performance is particularly achieved under soil moisture stressed conditions."

p. 19: The behavior of iWUE and ci/ca as reconstructed from the tree-ring measurements and modeled by LPX-Bern is compared to other studies. The nice thing on this paragraph is that it is very comprehensive. But it could be clearer with respect to the method used in the cited studies. Rather than just listing the studies you could sort them by method, i.e. mention other isotope-based studies first, then other methods. At the moment studies using the same methods (e.g. FACE) are mentioned in different parts of this section (Ainsworth & Long, 2005 and De Kauwe et al. 2013) which seems a bit fuzzy. I think this aspect is important as different methods are associated with different uncertainties (which, however, do not have to be discussed here).

Done. The studies are now grouped according to the underlying method.

With respect to the eddy covariance records it may be interesting to mention that a recent study (Knauer et al. 2016, New Phytologist) found that large-scale carbon and water fluxes are not in agreement with a constant ci, but rather with a constant ci/ca.

Done. The following text is added after the reference to the studies by Battipaglia et al. and Keenan et al.:

"However, these studies may be affected by local conditions not representative for larger scales and be influenced by temporal sampling biases as the record length of FACE and eddy-covariance measurements is limited. Indeed, the individual tree-ring records shown in Figure 7 show considerable site-to-site variability as well as decadal variability. In addition, factors other than ci/ca, e.g. vapor pressure deficit, may influence trends at eddy covariance sites (Knauer et al., 2017). The suggestion of a constant $c_i$ under increasing $CO_2$ is challenged by (Knauer et al., 2017). These authors find that the ecosystem trends reported by (Keenan et al., 2013) and a scenario of a constant $c_i$ as also suggested by (Battipaglia et al., 2013), are in conflict with observed large-scale trends in continental discharge, evapotranspiration, and the seasonal $CO_2$ exchange. Rather, the comparison of observational data and model outcome by (Knauer et al., 2017) support the finding of a physiological regulation towards a constant ci/ca under rising atmospheric $CO_2$."

p. 20: The effects of land use change and representation issues between the datasets/model simulations are adequately addressed. It may be helpful for the reader to mention the Figures again where the described aspects can be seen.

Done. Figure 4 is now mentioned at two places in the paragraph on land use.

Figures In some figures (e.g. Fig.2), the color code and the associated numbers are very small and hard to read. It would be ok to have fewer color classes as they are hard to distinguish.

Done. The number of labels is reduced and their size increased in Figures 2 to 4. Units are changed from gC m$^{-2}$ to kgC m$^{-2}$ to reduce the number of digits.

Fig. 2: For the difference maps, please state what is subtracted from what, at least in the legend.

The differences are based on decadal means (1996-2005 minus 1896-1905). The caption is revised: "The estimates are based on decadal means (1996-2005 and 1996-2005 minus 1896-1905, respectively)."

Fig. 5: representing the differences in mean delta13C as barplots is not appropriate here. I recommend to remove the bars and show the error bars only, also in Fig. 6.

We prefer to keep the bars for a clear distinction between the difference and the error estimates.

From Fig. 5 onwards: Some of the points on the map are hard to see. It would be helpful if their representation could be changed.

Our choice represents a trade-off between the visibility of individual dots and the visibility of all dots.

**Review 2: A. Ballantyne**

Here the authors compare a compilation of tree ring and leaf isotope data from around the world with isotopic simulations from two common land surface models. While several studies have compared isotopic estimates of iWUE with model simulations of iWUE, especially at regional scales. This study is novel in that it is one of the few to actually investigate isotopic tracer simulations within models as a critical diagnostic for how accurate models are at simulating the global C cycle. In principle, this approach allows us to evaluate to what extent the terrestrial biosphere is being fertilized by increased atmospheric CO2; however, I think that the authors could further partition the response of iWUE into its component processes of assimilation and transpiration (at least in the models). This may also help reconcile why the models appear to show differing degrees of iWUE response.

General Comments:

I suspect that the two models investigated here differ considerably in how stomatal conductance is simulated and this is having a big impact ultimately on the isotopic tracers. While these models may be responding similarly to increases in atmospheric CO2 they may be responding to different metrics of atmospheric water vapor. As the authors point out, assimilation in CLM is modeled as a function of RH and CO2, while it is my understanding that in LPJ stomatal conductance is modeled as a function of VPD. While RH and VPD may be inversely related in some environments, this is not always the case and their relationship might vary over the 20th century. It would be nice to see how assimilation and transpiration have responded over the 20th century independently in the two models. This may also help explain why LPX and CLM show different responses of iWUE over the 20th century.

[Figure]

Figure R5: Simulated transpiration (in mm/s) for the decade around year 2000 (top) and 20th century changes (bottom; 1996-2005 minus 1896-1905).

The panels shown in Figure R5 are added to figure 2. The following text is added on page 13:

"Simulated evapotranspiration is similar for the two models." and

"Simulated 20$^{th}$ century changes in annual-mean transpiration are generally small in both models, except in Australia for LPX and in parts of Latin America for CLM4.5. Generally an increase in transpiration is found in boreal and temperate forest regions in both models. Transpiration is slightly decreasing in LPX and slightly increasing in CLM4.5 in most tropical forest regions. "

**Specific Comments:**

P1L12 'water loss by transpiration.'

Done. Text changed as requested.

P2L4 'and water transpiration'

Done. Text changed as requested.

P2L22 Graven article is on 14C not 13C as cited. Check reference as they may have also included 13C in their simulations.

Done. Reference removed.

P2L27 conductance can be of CO2 or H2O, could be more specific here and say 'transpiration' as the process and H2O as the mass.

Stomatal conductance is used here as a general term for $H_2O$ and $CO_2$. Transport of $^{12}CO_2$, $^{13}CO2$, and $H_2O$ through the stomata are by molecular diffusion. This process depends on molecule mass and leads to the "fractionation" between the $^{13}CO_2$ and $^{12}CO_2$ fluxes. Text is not changed.

P3L7 While the authors mention many 13C tree ring records, they fail to mention the pioneering work by Tans et al. which is found in the references.

Done. Reference added.

P3L22 'to complement recent advances in simulating marine carbon isotopes'

Done. Sentence changed to read:  "This is a step towards fully coupled isotope-enabled CESM applications and complements recent advances in simulating marine carbon isotopes".

P4L24 'reactant to product'

Done. Text changed as requested.

P5L10 I believe that diffusion is only relevant for fractionation in non vascular plants such as bryophytes as well.

Done. Text clarified by adding reference to Farquhar et al., 1989 at the beginning of the paragraph. Non-vascular plants are not included in the standard setup of CLM4.5

P6L6 more realistically related to the gradient between internal water pressure and atmospheric water pressure (approximated as vapor pressure deficit or VPD).

Done. Text clarified. The description of the relationship between conductance and assimilation in CLM4.5 is clarified and Eq. 14 (submitted MS) is moved here to page 6 (now Eq. 7).

P7L7 del 13C signature of what? Atmosphere? Please clarify

Done. Text clarified to read: "Note that no $\delta^{13}$C observational data, e.g., from tree-rings or atmospheric samples, was used as a constraint in the assimilation."

Eqn 8 Isn't this the same as Eqn 7? But not quite sure c* is specified in Eqn 7.

Done. Model description is revised. Please see reply to reviewer 1 and revised manuscript.

P9L9 del 13 C is estimated as the 'weighted flux' of component GPP fluxes from PFTs from within grid cell. Omit 'GPP is used as a weight'.

Done. Sentence deleted as requested.

Eqn 11. I don't think that this equation is necessary (especially given the number of equations already included) and this can simply be explained.

Done. Equation 11 deleted and text adjusted.

P10L14 While this approximation of 36.16 holds well within 0.1% isotopic ratio differences are per 1000, so is this enough significant figures?

Done. Text clarified. The approximation is for Eq. 12 (in the submitted MS) and not for 36.16. It holds exactly: q=1.6 (a-b) = 1.6 (27-4.4) = 36.16. Additionally, four significant digits are enough as units here correspond to "permil units" and are not "per 1000".Text clarified to read: "The approximation given in Eq. 14 holds well within 0.1%."

P11L1 Not sure that you need to correct for the offset if you are only focusing on the trends and normalizing them across sites, regardless this should not affect your analysis.

We agree with the reviewer that this correction hardly influences our analyses and does not alter conclusions. Nevertheless, the relative change in iWUE depends weakly on the magnitude of the discrimination ($\Delta_i(t_1)$) as evident from equation 13 (Eq. 15 in resubmission) and as clearly stated in the manuscript.

P12L18 Were the CLM and LPX simulations conducted with or without land use change and does this have any impact on the global isotopic budget.

Yes, anthropogenic land use is considered in the CLM and LPX simulations as stated on p6, line 26 for CLM and on page 8, line 22). Land use maps from Hurtt et al., 2006 are used in both models. Implications for the global isotopic budget are discussed on page 20, line 14 to 21 in the submitted manuscript.

P13L13 'changes in the atmospheric del13C source'

Done. Text adjusted as requested.

P13L15 'globally-averaged' what? Soil, atmosphere?

Done. Text clarified to read: "There is a substantial offset between the models for globally-averaged soil and vegetation pools, with CLM4.5 being around 5 per mil more negative."

P13L34 2.42 and 3.22 per mil these should have units

Done. Per mil units added.

P14L8 Maybe these global mean estimates should be reported first before noting all the regional differences and more nuanced results.

Done. Global mean estimates are now reported before the regional differences.

P14L12 'bias of the models'

Done. Typo corrected.

P14L28 Were any trend statistics (e.g. Mann-Kendall) conducted on the observations or models?

As suggested, we performed a Mann-Kendall analysis on the discrimination time series. The trend in discrimination of C3 trees is most monotonic in CLM-4.5 ($\tau$= -0.9), less monotonic in the average tree-ring record $\tau$ = -0.6 for annual data and -0.7 for decadal mean data) and almost negligible in the LPX-Bern output ($\tau$ = -0.2).

The Mann-Kendall test provides information to which extent a trend is monotonic, but does not provide information on the magnitude of the trend. We do not plan to add the results from this test to the manuscript. It is clear from Figure 7 that the trends in discrimination and iWUE are larger in CLM4.5 than in LPX and the tree-ring data (see also Figure R6 below).

In response to the comment, we added the following text to the caption of figure 7: Standard deviations calculated over all observations of century-scale changes (1980s minus 1900s) are 0.72 for $\Delta i$ and 9.76 for iWUE, rspectively.

P15L10 These aren't really 'spatial' correlations

Done. Word spatial deleted.

P15L15to21 This paragraph seems to fit better in the methods

Done. Paragraph moved to method section. This implied a slight rearrangement of the subsections in the method section. Spin-up and transient simulations for both CLM4.5 and LPX are now described in the same subsection.

P16L24 Model simulations with increased CO2 and constant climate change could be compared at least quantitatively to FACE data.

This is beyond the scope of this study. For a comparison between carbon-nitrogen land models and FACE data we refer to the literature, e.g., (Zaehle et al., 2014).

P16L34 'Recall, however, that …represent annual or multi-annual averages that have been weighted by C assimilation or alloclation'

Done. Text adjusted as proposed.

P17L28 This paragraph is rather short think about combining.

We prefer to keep the paragraph as is to give the same weight to the conclusions drawn at the end of this and the previous paragraph.

Eqn. 14 Would also be interesting to compare how conductance is simulated in LPX. While assimilation in both models is clearly responding to increasing atmospheric CO2, I suspect that transpiration may be responding differentially in the models due to different stomatal response to atmospheric water demand.

The methods section is revised and explains how conductance is modelled in LPX. See response to a similar comment by reviewer 1.

P18L27 For the CLM response you should look at the relative changes in CO2 and relative humidity over time (this should be a prognostic variable in the model). Also see work by Isaac Held on the response of the hydrologic cycle to atmospheric warming. Essentially, at the global scale RH does not change in response to warming; however, this might not be true over land. So it would be interesting to see in CLM how RH has changed at the tree ring sites.

This point is rather subtle and follows from Eq. 14 (Eq. 7 in resubmitted MS). We will delete the text from line 25 to 31 on page 18 of the submitted manuscript to avoid misunderstanding and for brevity. Please see also reply to reviewer 1, concerning this point.

P20L5 Similar work by Penuelas et al (2011) has shown an increase in water use efficiency but not necessarily an increase in annual ring width. However, a true test would be the relationship between WUE and Biomass- not sure if Klein looked at biomass in this study.

Done. The following text was added:

"Similarly, (Peñuelas et al., 2011), analyzing changes in tree ring $\delta^{13}C$ and growth at 47 sites worldwide, inferred little change in discrimination, $c_i/c_a$, and an increase in iWUE of 20.5% from 1960 to 2000. 18 of the 35 studies with growth data show an increase in growth with time, while the others show no or negative growth changes."

In addition, we also discuss work by Streit et al.: "(Streit et al., 2014) measured needle gas exchange and analyzed $\delta$13C of needles and tree rings of *Larix decidua* and *Pinus mugo* after nine years of free air CO2 enrichment. Both species showed an increase in net photosynthesis and, in agreement with (Keel et al., 2006), small or no changes in conductance under elevated
$CO_2$ and a change in iWUE roughly in proportion to the increase in $CO_2$. Elevated $CO_2$ induced increased basal area growth in *L. decidua*, but not in *P. mugo*. Neither nitrogen limitation, nor end-product limitation, and, in agreement with other FACE studies (Bader et al., 2010;Liberloo et al., 2007), no downregulation of maximal photosynthetic rate was found.

P20L9 It seems that both of these FACE studies report a consistent increase in WUE, but of slightly different magnitudes. It is interesting that the responses are so different between European forests and the N. American forests. Unfortunately, most of the FACE studies have been conducted in the Eastern US, where there are no tree ring isotope records.

Thank you. We do not plan to extend the discussion on these FACE studies.

Figure 3. not so sure that the Carvalhais estimates are 'observations', maybe 'derived from' or 'constrained by' observations.

Done. Text modified to read "derived from observations".

Figures 5 and 6. I am not sure that you need both of these figures as they illustrate the same data. Perhaps move one to supplemental.

We will keep both figures as part of the manuscript. This decision is also in response to the comment of reviewer 1 on the visibility of the tree-ring data dots. Showing only regional maps as in Figure 6 would provide incomplete information of the global picture, whereas showing only the global map as in Figure 5 would affect clarity and visibility of regional results.

Figure 7. Can you include all of the tree ring records as thin grey traces in this figure? Would be nice to see some distribution of the observations to see if all the obs are bound by the model simulations.

[Figure]

Figure R6: Changes in discrimination and iWUE. Similar as figure 7 in the original manuscript, but with individual tree ring series added (thin gray lines). All model and tree-ring series represent decadally-smoothed values.

[Figure]

Figure R7: Same as Figure R6, but tree-ring series represent annual values.

We will replace Figure 7 with Figure R6 as requested.

Figure 8. Not sure that you need the discrimination equation, which should be defined in the text.

Done. Equation removed from figure caption. Equation 5, showing this definition, is now mentioned.

Figure 9. The right hand panels where certain variables have been kept constant is not explained in the caption

Done. Explanation added to the caption:

"The bar plots show the regional average change in discrimination of C3 trees as calculated from tree-ring $\delta^{13}$C data (gray) and from results of the standard simulations of CLM4.5 (filled blue) and LPX (filled red) and from factorial runs (pattern). Individual driving factors (climate, $CO_2$, N-deposition, and land use) were kept constant in the factorial runs as explained in the main text and indicated by the legend."

Figures 11 and 12. Once again these figures are both great but they illustrate redundant information maybe move one to the supplemental information.

We will keep both figures as part of the manuscript for the same reason as discussed above for figures 5 and 6

In summary, with tree ring isotope data we are only able to approximate iWUE and cannot partition this response between assimilation and transpiration. However, in the models you can partition these processes, so it would be interesting to see how transpiration and assimilation are responding in the models, which may help identify processes that can reconcile these model simulations.

Figure on transpiration will be added as requested. See response to main comment above.

[revised manuscript text omitted]